# Empowering Decision Trees via Shape Function Branching

**Nakul Upadhya, Eldan Cohen**
Department of Mechanical and Industrial Engineering
University of Toronto, Toronto, Canada
`nakul.upadhya@mail.utoronto.ca, eldan.cohen@utoronto.ca`

## Abstract

Decision trees are prized for their interpretability and strong performance on tabular data. Yet, their reliance on simple axis-aligned linear splits often forces deep, complex structures to capture non-linear feature effects, undermining human comprehension of the constructed tree. To address this limitation, we propose a novel generalization of a decision tree, the Shape Generalized Tree (SGT), in which each internal node applies a learnable axis-aligned shape function to a single feature, enabling rich, non-linear partitioning in one split. As users can easily visualize each node's shape function, SGTs are inherently interpretable and provide intuitive, visual explanations of the model's decision mechanisms. To learn SGTs from data, we propose ShapeCART, an efficient induction algorithm for SGTs. We further extend the SGT framework to bivariate shape functions ($S^2$GT) and multi-way trees ($SGT_K$), and present Shape$^2$CART and ShapeCART$_K$, extensions to ShapeCART for learning $S^2$GTs and $SGT_K$s, respectively. Experiments on various datasets show that SGTs achieve superior performance with reduced model size compared to traditional axis-aligned linear trees.

## 1 Introduction

Tabular datasets are collections of data organized into rows and columns, containing distinct features that can be continuous, categorical, or ordinal and remain among the most widely used dataset types in machine learning [1, 2]. Decisions trees are one of the most widely used approaches on these tabular datasets [3], largely due to their inherent interpretability [4] and robust performance [3]. These characteristics have established them as a favored choice in high-stakes domains such as healthcare [5, 6], finance [7, 8], and manufacturing [9, 10], where the ability to understand the model predictions is often just as important as high predictive accuracy. Structurally, decision trees organize nodes hierarchically, with internal nodes responsible for routing data samples based on defined criteria and terminal (leaf) nodes providing the final predictions. A common type of tree is the *binary axis-aligned linear tree*, where each internal node routes data to a left or right child node based on a simple threshold comparison involving only a single feature: $x_d \leq \theta$ (send to left if feature $d$ is less than or equal to threshold $\theta$).

While axis-aligned linear trees are simple and interpretable, the impact of feature values on the target is often non-linear, requiring repeated splits on the same features at different levels of the tree to represent the feature-target relationship adequately (see Figure 1b). However, repeatedly employing a single feature across multiple nodes in a decision path can hinder end-user comprehension of the model's learned relationship for that feature [11] and necessitate a larger tree with greater depth and node count to attain sufficient performance. This increase in size directly impacts interpretability, with empirical evidence confirming that tree comprehensibility is highly sensitive to the depth of decision paths and the number of leaves [4]. Furthermore, deeper trees inherently generate larger,

39th Conference on Neural Information Processing Systems (NeurIPS 2025).

less interpretable local explanations [12, 13]. Finally, larger trees are also challenging to visualize effectively, often requiring complex, interactive tools to gain insights [12, 14].

Generalized Additive Models (GAMs) [15] are another class of interpretable models where predictions are the result of the sum of individual feature contributions, each defined by a potentially non-linear *shape function*. Researchers have parameterized these shape functions using diverse model classes such as splines [15], trees [16, 17], and neural networks [18–20], enabling powerful modeling capabilities. Shape functions offer a key interpretability benefit as their flexibility enables GAMs to closely capture the true feature–target dynamics in the data [21] while still allowing practitioners to visualize each function independently, faithfully conveying the learned relationship [15, 21]. GAMs have also been extended to incorporate pairwise feature interactions [17, 19, 20, 22] by adding bivariate shape functions, creating $GA^2Ms$. Bivariate shape functions can be visualized via heatmaps, improving modelling capacity as they can capture second-order feature interaction effects.

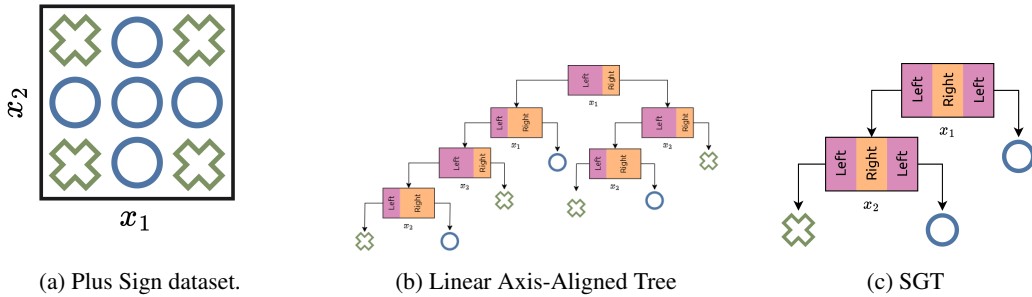

(a) Plus Sign dataset.  (b) Linear Axis-Aligned Tree  (c) SGT

Figure 1: Demonstration of an SGT on a simple toy dataset. As seen here, the SGT can achieve perfect accuracy with a smaller depth and number of nodes compared to the linear axis-aligned tree.

In this work, we introduce the *Shape Generalized Tree (SGT)*, a novel class of decision tree models where each internal node utilizes an axis-aligned shape function to determine the routing of samples to child nodes. As shape functions can be parametrized by a variety of rich models, they provide the capability to capture non-linear decision boundaries with respect to a chosen feature in each node. The benefit of increasing the capabilities of individual nodes can be seen clearly in Figure 1. Figure 1a visualizes a simple two dimensional synthetic dataset. A traditional axis-aligned linear tree requires six splits with a maximum depth of four (Figure 1b) to represent decision boundaries in this dataset, while the SGT can achieve the same result with only two nodes with a maximum depth of two (Figure 1c). To build SGTs, we propose *ShapeCART*, a novel and efficient SGT induction algorithm modeled after the CART framework [23] that constructs shape functions in each node via internal decision trees. Furthermore, we extend the SGT framework by incorporating bivariate shape functions in each node, resulting in $S^2GT$, and by enabling the construction of trees with higher branching factors (i.e., multi-way trees), resulting in $SGT_K$. Correspondingly, we extend the ShapeCART algorithm to accommodate these extensions, resulting in $Shape^2CART$ and $ShapeCART_K$.

Overall our contributions are as follows: *(1) We introduce Shape Generalized Trees (SGTs), a new family of decision trees where internal node splits are governed by interpretable axis-aligned shape functions that can capture non-linear feature effects; (2) We introduce ShapeCART, a novel and efficient top-down induction algorithm for constructing SGTs; (3) We extend the SGT Framework to accommodate multi-way trees ($SGT_K$) and bivariate shape functions in each node ($S^2GT$) and modify ShapeCART accordingly; (4) We demonstrate across a range of datasets that SGTs achieve superior performance with more compact trees compared to traditional linear trees.*

## 2 Background

### 2.1 Decision Trees

Consider a supervised learning setting where we are given a dataset consisting of $N$ data points $\mathcal{X} = \{\mathbf{x}_n\}_{n=1}^N$, where each $\mathbf{x}_n \in \mathbb{R}^D$ is a $D$-dimensional feature vector, and their corresponding

labels $\mathcal{Y} = \{y_n\}_{n=1}^N$. Within a decision tree framework, let $\mathcal{D} \subseteq \mathcal{X} \times \mathcal{Y}$ represent the subset of training data that reaches a specific internal node. The core mechanism of a decision tree involves recursively partitioning the data $\mathcal{D}$ at each internal node based on a defined splitting criterion. Among the most common type of decision trees is the binary axis-aligned linear tree:

**Definition 1 (Binary Axis-Aligned Linear Tree).** *In a binary axis-aligned linear tree, internal nodes partition the data $\mathcal{D}$ into two subsets, $\mathcal{D}^l$ and $\mathcal{D}^r$, based on a threshold $\theta$ applied to a single feature encoded by a one-hot vector $\mathbf{w}$:*

$$\mathcal{D}^l = \{(\mathbf{x}_n, y_n) \in \mathcal{D} \mid \mathbf{w}^\top \mathbf{x}_n \leq \theta; \; \mathbf{w} \in \{0,1\}^D, \|\mathbf{w}\|_0 = 1, \theta \in \mathbb{R}\} \quad \mathcal{D}^r = \mathcal{D}/\mathcal{D}^l \quad (1)$$

A common paradigm for learning binary-axis aligned trees is the Top-Down Induction of Decision Trees (TDIDT) strategy, a greedy procedure that recursively partitions data to minimize a predefined impurity measure, used in algorithms like the widely adopted Classification and Regression Tree (CART) [23]. The appeal of these approaches lies in their empirical performance and efficiency, allowing trees to be trained quickly and scale well. Extensions like SERDT [13] incorporate inductive biases to encourage path-level sparsity and improve interpretability. Other methods refine trees after initial training, such as Tree Alternating Optimization (TAO) [24] which iteratively updates nodes to boost predictive performance and prune the tree, and Hierarchical Shrinkage [25] which modifies the empirical target distributions in each leaf based on the parents of each leaf. Alternatively, global optimization methods such as DL8.5 [26], GOSDT [27], and OMT [28] aim to learn optimal trees but typically scale poorly on large datasets [26, 27] and often require pre-discretization of continuous features [26–28]. Recently, there has been work on developing non-greedy *near-optimal* tree induction methods. Some notable approaches include DPDT [29], which formulates tree construction as an MDP, and SPLIT [30], which reduces the runtime of GOSDT [27] via lookahead methods.

Another important type of decision trees aims to enhance node expressivity by utilizing oblique splits, which allow internal nodes to split on linear combinations of multiple features:

**Definition 2 (Binary $M$-variate Oblique Tree).** *In a Binary $M$-variate oblique tree, nodes use a linear combination of at most $M$ features to partition the data:*

$$\mathcal{D}^l = \{(\mathbf{x}_n, y_n) \in \mathcal{D} \mid \mathbf{w}^\top \mathbf{x}_n \leq \theta; \; \mathbf{w} \in \mathbb{R}^D, \|\mathbf{w}\|_0 \leq M, \theta \in \mathbb{R}\} \quad (2)$$

However, oblique trees are difficult to interpret on most datasets due to the high dimensionality of each cut, making it difficult for users to reason about model decisions. Some notable approaches for learning unconstrained oblique trees ($M = D$) include utilizing the TAO [24] framework and soft decision trees, which use a continuous relaxation of the binary oblique cut to learn a tree using gradient descent. Of particular relevance to our work are pairwise oblique models ($M = 2$) such as BiCART [31], BiTAO [31], and BiDT [32], which restrict each split to consider linear combinations between exactly two input features. This restriction allows the decision boundary in each node to be visualizable, unlike higher-order ($M > 2$) oblique trees, and have shown promise in constructing accurate and compact trees.

## 2.2 Interpretability

Model interpretability, revealing the internal mechanisms driving predictions, is crucial for fostering trust and enabling effective deployment, especially in high-stakes domains. Interpretability is typically understood across two primary scopes: global interpretability, concerning the model's overall logic and the relationships it has learned across the entire dataset, and local interpretability, explaining the model's prediction for a specific instance and the effect of small input perturbations [33]. Approaches to achieving interpretability fall into two main categories: intrinsic model-based interpretability, where the model itself is structured to allow for direct understanding and reasoning about its predictions, and post-hoc interpretability, which uses external methods after training to approximate explanations for models lacking this inherent transparency [12, 34]. Intrinsic model-based interpretability, which allows for understanding at both local and global scopes, is facilitated by specific structural properties of the model, including: *Modularity* (or composability) where different parts of the model can be interpreted independently [34, 35]; *Simulability*, where users can reason about and faithfully trace the model's decision process; [33, 34]; and *Sparsity*, where the complexity of the model is constrained, through both reducing the number of features used and constraining the number of distinct model components a user must analyze [34, 35].

Decision trees are considered modular and simulable models, and provide model-based interpretability when their sparsity is enforced through limiting the total number of features used [13, 34] and restricting the size of the tree, measured by node count and maximum depth [4, 12, 13, 36]. Further discussion on the interpretability of trees in comparison to other models can be found in Appendix A.

## 3   Shape Generalized Tree

While effective in many scenarios, linear splitting criterion may result in a feature being used multiple times in a given decision path, leading to larger trees where the learned relation between the feature and the target is difficult for users to understand. To address this drawback, we propose a novel generalization of linear trees, the **Shape Generalized Tree (SGT)**, that can represent non-linear decisions within a single node. More formally, we define a binary axis-aligned SGT as follows:

**Definition 3 (Binary Axis-Aligned Shape Generalized Tree (SGT)).** *In a Shape Generalized Tree, internal nodes partition the data based on the output of a flexible shape function applied to a single selected feature:*

$$\mathcal{D}^l = \{(\mathbf{x}_n, y_n) \in \mathcal{D} \mid f_\Theta(\mathbf{w}^\top \mathbf{x}_n) \leq 0.0; \mathbf{w} \in \{0,1\}^D; \|\mathbf{w}\|_0 = 1\} \tag{3}$$

*Here, $\mathbf{w}$ is a one-hot feature selection vector where $d = \arg\max(\mathbf{w})$ is the selected feature, and $f_\Theta : dom(\mathcal{X}_d) \to \mathbb{R}$; is a flexible shape function parameterized by $\Theta$. $f$ can be drawn from richer function classes, enabling the capture of more intricate, non-linear decision boundaries. As $f$ operates only on one feature, the shape function is easily visualized and interpreted.*

Additionally, we propose extending this generalization to accommodate bivariate shape functions in each node, leading to the Binary Bivariate Shape Generalized Tree ($S^2GT$) model:

**Definition 4 (Binary Bivariate Shape Generalized Tree ($S^2GT$)).** *In an $S^2GT$, internal nodes partition the data based on a shape function that utilizes at most two features:*

$$\mathcal{D}^l = \{(\mathbf{x}_n, y_n) \in \mathcal{D} \mid f_\Theta^2(\mathbf{w}_0^\top \mathbf{x}_n, \mathbf{w}_2^\top \mathbf{x}_n, \Theta) \leq 0.0;$$
$$\mathbf{w}_1, \mathbf{w}_2 \in \{0,1\}^D; \|\mathbf{w}_1\|_0, \|\mathbf{w}_2\|_0 = 1\} \tag{4}$$

*Here, $\mathbf{w}_1, \mathbf{w}_2$ are one-hot feature selection vectors where $d_1 = \arg\max \mathbf{w}_1, d_2 = \arg\max \mathbf{w}_2$ are the selected features, and $f_\Theta^2 : dom(\mathcal{X}_{d_1}) \times dom(\mathcal{X}_{d_2}) \to \mathbb{R}$ is a shape function that maps the pair of features to an output branch. The Binary Bivariate Oblique Cut (Definition 2) can be seen as a special case where $f_\Theta(z_1, z_2) = \Theta_1 z_1 + \Theta_2 z_2 - \Theta_0$.*

One significant advantage of employing shape functions is that they can be extended to accommodate multi-way branching (i.e. routing data into $K > 2$ child nodes simultaneously). This facilitates the creation of the $K$-Way Axis-Aligned Shape Generalized Tree ($SGT_K$) model:

**Definition 5 (Multi-Way Axis-Aligned Shape Generalized Tree ($SGT_K$)).** *In an $SGT_K$, internal nodes partition the data into up to $K$ distinct subsets based on the output of a vector-valued shape function applied to a single feature:*

$$\mathcal{D}^k = \{(\mathbf{x}_n, y_n) \in \mathcal{D} \mid \arg\max_{k \in \{1, \ldots, K\}} \left( f_\Theta^{(K)}(\mathbf{w}^\top \mathbf{x}_n) \right) = k; \mathbf{w} \in \{0,1\}^D, \|\mathbf{w}\|_0 \leq 1\} \tag{5}$$

*Here, the shape function $f^{(K)} : dom(\mathcal{X}_d) \to \mathbb{R}^K$; $d = \arg\max \mathbf{w}$ maps the selected feature value to a $K$-dimensional vector, and the argmax operation determines the branch assignment.*

Although our model naturally extends to any $K$, we limit our exploration of $SGT_K$ to $K = 3$ to maintain sparsity and, consequently, a high degree of interpretability. By integrating the bivariate splitting and $K$-way branching, we obtain $S^2GT_K$ trees, which support multi-way splits based on non-linear interactions between two features (formal definition in Appendix B).

**Expressiveness Guarantees for SGTs**   We formally establish that SGTs are more expressive than binary axis-aligned linear trees by showing that: (1) SGTs are at least as expressive as binary axis-aligned linear trees with a similar number of decision nodes; (2) SGTs can be strictly more expressive than axis-aligned linear trees with a similar number of decision nodes. More formally:

**Theorem 1.** *Every function that can be represented by a binary axis-aligned linear tree (Definition 1), can be represented by an SGT (Definition 3) with the same number of decision nodes.*

**Algorithm 1:** Selecting Shape Function (Node Level Outer Problem)

---

**Input** : Features $\mathcal{X}$, Target $\mathcal{Y}$, Branching Factor $K$, Pairwise Candidate Limit $P$, Coordinate Descent
Iterations $R$, Pairwise Reg. $\gamma$, Branching Reg. $\lambda$, Impurity $\mathcal{H}(\cdot)$, Weighted Impurity $\mathcal{L}(\cdot)$
**Output**: Set of branching partitions $\boldsymbol{D}^*$, Best Shape Function $f^*(\cdot)$, Best Weighted Impurity $\boldsymbol{L}^*$

---

$\boldsymbol{L}^* \leftarrow \infty; f^*(\cdot) \leftarrow \texttt{None}; \boldsymbol{D}^* \leftarrow \texttt{None}$                                    `// Initialize`
**for** $d \leftarrow 1$ **to** $D$ **do**                                          `// Iterate through all features`
$\quad \boldsymbol{L}_d, f_d(\cdot), \boldsymbol{D}_d \leftarrow \text{FitShapeFunction}(\mathcal{X}, \mathcal{Y}, \mathcal{H}(\cdot), K, \lambda, R, d)$
$\quad$ **if** $\boldsymbol{L}_d < \boldsymbol{L}^*$ **then**
$\quad\quad \mid \boldsymbol{L}^* \leftarrow \boldsymbol{L}_d; f^*(\cdot) \leftarrow f_d(\cdot); \boldsymbol{D}^* \leftarrow \boldsymbol{D}_d$
**end**
**if** *P > 0* **then**
$\quad$ **for** $(d_1, d_2) \in$ *PairwiseCombinations(D)* **do**          `// Filter Interactions (Section 4.2)`

$\quad\quad \boldsymbol{D}_{\text{Intersect}} \leftarrow \{\mathcal{D}^i \cap \mathcal{D}^j \ \forall (\mathcal{D}^i, \mathcal{D}^j) \in \boldsymbol{D}_{d_1} \times \boldsymbol{D}_{d_2}\}$
$\quad\quad \delta_{(d_1, d_2)} \leftarrow \mathcal{L}(\boldsymbol{D}_{\text{Intersect}});$
$\quad$ **end**
$\quad \Delta_P \leftarrow P$ Best $\delta$ values
$\quad$ **for** $(d_1, d_2) \in \Delta_P$ **do**                    `// Iterate through candidate pairs of features`
$\quad\quad \boldsymbol{L}_{(d_1,d_2)}, f_{(d_1,d_2)}(\cdot), \boldsymbol{D}_{(d_1,d_2)} \leftarrow \text{FitShapeFunction}(\mathcal{X}, \mathcal{Y}, \mathcal{H}(\cdot), K, \lambda, R, (d_1, d_2))$
$\quad\quad \boldsymbol{L}_{(d_1,d_2)} \leftarrow \boldsymbol{L}_{(d_1,d_2)} + \gamma$                         `// Add Pairwise Penalty`
$\quad\quad$ **if** $\boldsymbol{L}_{(d_1,d_2)} < \boldsymbol{L}^*$ **then**
$\quad\quad\quad \mid \boldsymbol{L}^* \leftarrow \boldsymbol{L}_d; f^*(\cdot) \leftarrow f_{(d_1,d_2)}(\cdot); \boldsymbol{D}^* \leftarrow \boldsymbol{D}_{(d_1,d_2)}$
$\quad$ **end**
**return** $\boldsymbol{D}^*, \boldsymbol{L}^*, f^*(\cdot)$

---

**Theorem 2.** *For every $B \in \mathbb{N}$, there exists a function for which a binary axis-aligned tree requires at least $B$ additional decision nodes to represent, compared to an SGT.*

To prove Theorem 1, we demonstrate that axis-aligned linear trees are a special case of SGTs. To prove Theorem 2, we construct a family of one-dimensional labeling functions with periodic boundaries and show that an SGT can represent any function in this family using a fixed number of nodes, while an axis-aligned linear tree requires a number of nodes that grows linearly with the number of intervals. Complete proofs for both theorems can be found in Appendix C.

## 4   ShapeCART

To learn SGTs from data, we propose *ShapeCART*, a novel and efficient tree-induction algorithm inspired by the classic Classification and Regression Tree (CART) framework [23]. Furthermore, we extend the core ShapeCART algorithm to learn SGT$_K$s and S$^2$GTs, creating ShapeCART$_K$ and Shape$^2$CART respectively. For simplicity, we present our methods in the classification setting, where each label $y_n \in \mathcal{Y}$ takes values in a finite class set $\mathcal{C}$, but provide a detailed description of the regression variant in Appendix J. ShapeCART constructs a decision tree via the TDIDT paradigm (Pseudocode in Appendix K), decomposing the global optimization into greedy, recursive node-level learning tasks similar to CART. At each node, we aim to learn feature selection parameters $\mathbf{w}$ and shape-function parameters $\Theta$ that minimize the weighted impurity of the empirical class distributions in the resulting child nodes. We pose this node-level problem as a bi-level optimization: (1) An inner problem to learn the optimal shape-function parameters for each candidate feature and (2) an outer problem to select the shape function that yields the lowest impurity:

$$\min_{\mathbf{w} \in \{0,1\}^D, \|\mathbf{w}\| \leq 1} \ \sum_{d=1}^{D} w_d \min_{\Theta_d} \left[ \mathcal{L}\left(\{\mathcal{D}_d^{\text{l}}, \mathcal{D}_d^{\text{r}}\}\right) \right] \tag{6}$$

$$\text{s.t.} \quad \mathcal{D}_d^{\text{l}} = \{(\mathbf{x}_n, y_n) \in \mathcal{D} \mid f_d(x_{n,d}) \leq 0.0\}$$

Here, $f_d(\cdot) \equiv f_{\Theta_d}(\cdot)$ and $\mathcal{L}(\cdot)$ is the weighted impurity of the resulting branches, defined for any superadditive impurity measure $\mathcal{H}$ (e.g. Gini [23] or entropy [37]) as:

$$\mathcal{L}(\boldsymbol{D}) = \sum_{\mathcal{D} \in \boldsymbol{D}} |\mathcal{D}| \mathcal{H}(\Pi(\mathcal{D})); \quad \Pi(\mathcal{D})_c = \frac{1}{|\mathcal{D}|} \sum_{(x_n, y_n) \in \mathcal{D}} \mathbf{1}(y_n = c) \ \forall c \in \mathcal{C} \tag{7}$$

To solve this problem, we construct shape functions for each feature and select the one with the lowest weighted impurity (node level pseudocode in Algorithm 1). Additionally, this formulation naturally extends to implementing $SGT_K$ (Equation 5) by adopting a multi-way shape function in lieu of the binary one, creating $ShapeCART_K$.

**Extending to Shape$^2$CART** To extend ShapeCART to Shape$^2$CART, we extend the above formulation to consider both individual feature shape functions and bivariate shape functions. The resulting bi-level optimization problem is as follows:

$$\min_{\mathbf{w},\mathbf{W}} \quad \sum_{d=1}^{D} w_d \min_{\Theta_d} \left[ \mathcal{L}\left(\{\mathcal{D}_d^{\mathtt{l}}, \mathcal{D}_d^{\mathtt{r}}\}\right) \right] + \tag{8}$$

$$\sum_{(d_1,d_2)\in\Delta} W_{(d_1,d_2)} \min_{\Theta_{(d_1,d_2)}} \left[ \mathcal{L}\left(\{\mathcal{D}_{(d_1,d_2)}^{\mathtt{l}}, \mathcal{D}_{(d_1,d_2)}^{\mathtt{r}}\}\right) \right] + \gamma\|\mathbf{W}\|_0$$

$$\text{s.t.} \quad \mathcal{D}_d^{\mathtt{l}} = \{(\mathbf{x}_n, y_n) \in \mathcal{D} \mid f_d(x_{n,d}) \leq 0.0\} \tag{9}$$

$$\mathcal{D}_{(d_1,d_2)}^{\mathtt{l}} = \{(\mathbf{x}_n, y_n) \in \mathcal{D} \mid f_{(d_1,d_2)}^2(x_{n,d_1}, x_{n,d_2}) \leq 0.0\}$$

$$\|\mathbf{w}\|_0 + \|\mathbf{W}\|_0 = 1, \mathbf{w} \in \{0,1\}^D, \mathbf{W} \in \{0,1\}^{D\times D}$$

Here, $\mathbf{w}$ is a one-hot univariate feature selection vector, $\mathbf{W}$ is a one-hot bivariate interaction selection matrix, $\Delta$ is a set representing all pairs of features, and $\gamma$ is a tuneable regularization hyperparameter added to penalize selection of a bivariate shape function to encourage choosing axis-aligned splits unless the bivariate function leads to a significantly lower loss.

## 4.1 Learning Shape Functions

Recall that for a given feature or pair of features, we would like to learn a shape function $f_d(\cdot)$ (or $f_{(d_1,d_2)}$ for bivariate trees) that assigns samples to a branch (e.g., `left`/`right` or $1,\ldots,K$ in the multi-way case) such that we minimize the impurity of the empirical class distributions of the resultant branches. To learn this shape function efficiently, we propose a two-stage approximation strategy:

1. We first learn a function $\mathbf{T}(\cdot): \mathbb{R} \to \{1,\ldots,L\}$ that maps samples to $L$ mutually exclusive bins based on the value of the $d$-th feature (or $(d_1, d_2)$ for bivariate shape functions).

2. We then compute an assignment of bins to branches (e.g., `left`/`right`) that minimizes the impurity over the induced partitioning of the data.

### 4.1.1 Binning the samples

We choose to represent the binning function $\mathbf{T}(\cdot)$ via an internal decision tree. For univariate shape functions, we use CART to train a binary axis-aligned linear tree only on the relevant feature and the target to construct $L$ mutually exclusive bins that approximately minimize the chosen impurity criteria $\mathcal{H}$. For bivariate shape functions, we instead train a binary bivariate oblique tree using BiCART [31] on the pair of features under consideration and the target. The use of CART and BiCART affords us control over model complexity; by constraining parameters such as the number of leaves and the minimum of samples in each leaf tree depth, one can ensure that the resulting piecewise-constant shape function comprises a bounded number of meaningful segments. Each bin $\ell = 1,\ldots,L$ stores its empirical class distribution $\pi_\ell \in \mathbb{R}^{|\mathcal{C}|}$ and weight $W_\ell \in \mathbb{R}$, representing the number of samples that fall into the bin. Note that while we utilize CART, this step can be performed with a variety of approaches, and we provide results using an alternative tree induction algorithm (DPDT [29]) in Appendix I.

### 4.1.2 Mapping Bins to Branches

Given the bins from the first stage, our goal is to assign each bin $\ell$ to an output branch (e.g. `l,r`) such that we minimize the weighted impurity of the resultant branches. Formally, let $\mathbf{a} \in \{\mathtt{l},\mathtt{r}\}^L$ denote

the branch assignment vector. We aim to solve the following discrete optimization problem:

$$\min_{\mathbf{a}} \quad W_{\mathbf{l}}\mathcal{H}(\Pi_{\mathbf{l}}) + W_{\mathbf{r}} \cdot \mathcal{H}(\Pi_{\mathbf{r}}) \tag{10}$$

$$\text{s.t.} \quad W_k = \sum_{\ell=1}^{L} \mathbf{1}(\mathbf{a}_\ell = k)W^\ell, \quad \Pi^k = \frac{\sum_{\ell=1}^{L} \mathbf{1}(\mathbf{a}_\ell = k)(W^\ell \pi^\ell)}{W_k} \tag{11}$$

To solve this optimization problem in a scalable manner, we adopt a coordinate descent procedure. At each step, we determine the assignment of a single bin $\ell$ while keeping all other assignments fixed. This reduces the optimization problem to iteratively evaluating the impurity resulting from assigning $\ell$ to each possible branch and selecting the assignment that minimizes the objective. The process is repeated for $R$ iterations, with the order of leaf updates randomly shuffled in each pass to mitigate ordering bias (Pseudocode in Appendix K). Coordinate descent guarantees monotonic objective decrease and finite convergence given the discrete search space [38], with time complexity scaling linearly in the number of leaves (Appendix D). However, its sensitivity to initialization can lead to suboptimal local minima [38]. To obtain a strong initialization, we choose the better of the following two strategies for setting $\mathbf{a}$: (1) cluster assignments obtained by applying Weighted K-Means to the empirical distributions of the bins $\pi_\ell$, weighted by $W_\ell$; or (2) bin assignments based on whether each bin falls to the left or right of the root node in the inner CART tree.

Based on the procedure detailed above, we can derive a lower-bound on the information gain at each decision node with respect to the information gain at each node in CART:

**Lemma 1.** *Let $t$ be an individual node in $T$, let $\mathcal{IG}(t)$ denote the information gain obtained by an axis-aligned (threshold) split on the samples in $t$ from CART, and let $\mathcal{IG}_f(t)$ denote the information gain obtained by a shape function split from ShapeCART on the samples in $t$. We show that $\mathcal{IG}_f(t) \geq \mathcal{IG}(t)$.*

Given this lemma, ShapeCART inherits the bound on information gain for any terminal node of the tree established for CART by Klusowski and Tian [39]. Proof of Lemma 1 can be found in Appendix C.2.

**Extending to Multi-Way Branching** ShapeCART can be adapted to the multi-way setting, ShapeCART$_K$, by increasing the number of clusters in the Weighted K-Means initialization

---

**Algorithm 2:** Fit Shape Function

**Input** : $\mathcal{X}$, $\mathcal{Y}$, $R$, $K$, $\lambda$, $\mathcal{H}(\cdot)$, $d$ or $(d_1, d_2)$,
**Output** : $\boldsymbol{L}_{k^*}$, $f_d(\cdot)$

**if** *$d$ was provided* **then**
  $\quad \mathbf{T} \leftarrow \text{FitCART}(\mathcal{X}_d, \mathcal{Y})$ ;
**else**
  $\quad \mathbf{T} \leftarrow \text{FitBiCART}(\mathcal{X}_{d_1}, \mathcal{X}_{d_2}, \mathcal{Y})$
**end**
$(W_1, \ldots, W_L), (\pi_1, \ldots, \pi_L) \leftarrow \text{ExtractBinStats}(\mathbf{T})$
$\mathbf{a}_{(0),\text{root}}, \boldsymbol{L}_{\text{root}}^{(0)} \leftarrow \text{ExtractRootAssignments}(\mathbf{T})$
**for** $k \leftarrow 2$ **to** $K$ **do**
  $\quad \mathbf{a}_{(0),\text{WKM}}^k, \boldsymbol{L}_{k,\text{WKM}}^{(0)} \leftarrow \text{WeightedKMeans}\big([\pi_1, \ldots, \pi_L];$
  $\qquad\qquad\qquad\qquad\qquad [W_1, \ldots, W_L]; k\big)$
  $\quad$ **if** $\boldsymbol{L}_{\text{root}}^{(0)} < \boldsymbol{L}_{k,\text{WKM}}^{(0)}$ **then**
  $\quad\quad \mathbf{a}_{(0)}^k \leftarrow \mathbf{a}_{(0),\text{root}}$
  $\quad$ **else**
  $\quad\quad \mathbf{a}_{(0)}^k \leftarrow \mathbf{a}_{(0),\text{WKM}}^k$
  $\quad$ **end**
  $\quad (\mathbf{a}^k, \boldsymbol{L}_k) \leftarrow \text{CoordDescent}\big(\mathbf{a}_{(0)}^k; [\pi_1, \ldots, \pi_L];$
  $\qquad\qquad\qquad\qquad [W_1, \ldots, W_L]; k; \mathcal{H}(\cdot); R\big)$
**end**
$k^* \leftarrow \arg\min_{k=2,\ldots,K} (\boldsymbol{L}_k + \lambda(k-2))$
$\{\mathcal{D}^1, \ldots, \mathcal{D}^{k^*}\} \leftarrow \text{RetrieveBranchAssignments}(\mathcal{X}, \mathbf{T}, \mathbf{a}^{k^*})$
$f(\cdot) \leftarrow \mathbf{a}_{\mathbf{T}(\cdot)}^{k^*}$
**return** $\boldsymbol{L}_{k^*}$, $f(\cdot)$, $\{\mathcal{D}^1, \ldots, \mathcal{D}^{k^*}\}$

---

and considering more options for $\mathbf{a}_\ell$ in each step of the coordinate descent procedure. To regularize ShapeCART$_K$ and prevent the creation of branches that do not provide a significant impurity decrease, we run the Weighted K-Means initialization and Coordinate Descent procedure for each $k \in \{2, \ldots, K\}$ and add a penalty term $\lambda(k-2)$ to the weighted impurity of the assignments for that $k$, where $\lambda$ is a hyperparameter. We then select the $k$ with the minimum penalized impurity.

In summary, the shape function for a given feature $f_d(\cdot)$ is represented jointly by an internal tree $\mathbf{T}_d(\cdot)$ which maps samples to bins, and a lookup vector $\mathbf{a}$ which maps bins to branches (Pseudocode in Algorithm 2). The time complexity of constructing a shape function is $\mathcal{O}\left(NC \log N + K^2 LC\right)$ where $C$ is the number of classes (see Appendix D for detailed analysis).

## 4.2 Limiting Bivariate Shape Function Construction

Recall that to solve the optimization problem presented in Equations 8, we need to construct all univariate and bivariate shape functions to select the one that minimizes the weighted impurity of the resultant branches. A naive approach of constructing a shape function for each pair of features will require constructing $\mathcal{O}(D^2)$ shape functions, incurring significant computational cost. Instead, we propose a heuristic that utilizes the computation of univariate shape functions to identify a few promising pairs for which a shape function will be constructed. We first construct univariate shape functions $f_1, \ldots, f_D$ and denote the set of branches resultant from applying each feature's shape functions on the input data as $\boldsymbol{D}_d = \{\mathcal{D}_d^1, \mathcal{D}_d^r\} \; \forall d = 1, \ldots, D$ (or $\boldsymbol{D}_d = \{\mathcal{D}_d^1, \ldots, \mathcal{D}_d^K\}$ in the multi-way case). Our goal is to identify pairs of features $(d_1, d_2)$ that have the potential to have an impurity that is significantly lower than the best impurity of the individual features. We achieve this by calculating the weighted impurity of the sets induced by the Cartesian product of the branching sets of a pair of features and subtracting this value from the best impurity obtained from the univariate shape functions in the pair:

$$\delta_{(d_1, d_2)} = \min(\mathcal{L}(\boldsymbol{D}_{d_1}), \mathcal{L}(\boldsymbol{D}_{d_2})) - \mathcal{L}(\{\mathcal{D}^i \cap \mathcal{D}^j | (\mathcal{D}^i, \mathcal{D}^j) \in \boldsymbol{D}_{d_1} \times \boldsymbol{D}_{d_2}\}) \tag{12}$$

We retain the $P$ pairs with the highest $\delta$ values and denote this set as $\Delta_P$, which we use in lieu of the full set of pairs ($\Delta$) in Equation 8. With our heuristic, we only require $\mathcal{O}((N + CK^2)D^2)$ operations to construct the intersection of sets followed by the construction of $P$ shape functions ($\mathcal{O}(P(NC \log N + K^2 LC))$ vs. $\mathcal{O}(D^2(NC \log N + K^2 LC))$). Assuming $P \ll D^2$, this heuristic can significantly reduce runtime (evaluated in Appendix G.2.3). The choice of $P$ is guided by the user's computational preferences; higher $P$ increases runtime in return for increased performance.

## 4.3 Post-Processing

TDIDT approaches are efficient and allow trees to be trained quickly. However, one downside of TDIDT approaches is their greedy behavior, potentially resulting in globally-suboptimal trees [24, 29]. To overcome this, we globally refine the tree constructed via ShapeCART using Tree Alternating Optimization (TAO) [24, 31]. This post-processing procedure refits the shape function at each internal node of the constructed tree using the predictions of the subtrees rooted at the node's children, while also pruning nodes that do not significantly reduce error. While the original TAO algorithm is restricted to binary trees, we extend it to support $K$-way branching. Full implementation details and our modifications to TAO are provided in Appendix F.

# 5 Experimental Evaluation

In this section, we evaluate the performance of trees induced by ShapeCART (SGT-C), ShapeCART$_3$ (SGT$_3$-C), Shape$^2$CART (S$^2$GT-C) and Shape$^2$CART$_3$ (S$^2$GT$_3$-C) against various benchmark approaches on a range of 26 real-world classification datasets (details in Appendix E). We also present results on the TAO refined ShapeCART models - denoted as SGT-T, SGT$_3$-T, S$^2$GT-T, and S$^2$GT$_3$-T. We benchmark our axis-aligned approaches against CART [23], SERDT [13], HSTree [25], Axis-Aligned TAO (AxTAO) [24], DPDT [29], and SPLIT [30]. We evaluate our bivariate approaches against the newly proposed BiCART and BiTAO algorithms [31]. CART, SERDT, HSTree, and BiCART are greedy TDIDT induction approaches, while AxTAO, DPDT, SPLIT and BiTAO are non-greedy induction approaches. Implementation details for all models evaluated can be found in Appendix F.

**Hyperparameter Search and Evaluation** In our experiments, we report results for all approaches across reasonable tree depths (2–6), following Souza et al. [13] and for the overall best model. Each dataset is split into three folds using a 70/30 train/validation–test split, with the non-training data further divided in the same ratio. Hyperparameters are optimized via Bayesian search with Optuna [40] using 50 trials per model and depth to ensure fairness across differing search spaces. Each configuration is scored by mean validation accuracy, and we retain the best per depth and overall. Training time is limited to 15 minutes for univariate and 30 minutes for bivariate models. All runs are executed on GCP N2 instances (8 vCPUs, 32 GB RAM). Hyperparameter search spaces for all evaluated approaches and hyperparameter importance analyses for all ShapeCART variants appear in Appendix F.1. We present average test accuracy results across datasets below, and additional evaluations and ablations are presented in Appendix G.

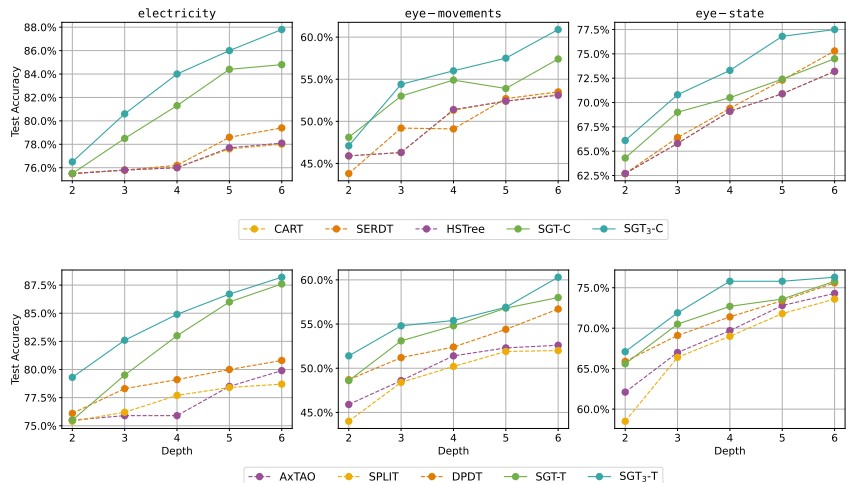

Figure 2: Test accuracy per depth for axis-aligned approaches on `eye-movements`, `electricity`, and `eye-state`. Top: TDIDT-Approaches. Bottom: Non-Greedy Approaches.

Table 1: Average Test Accuracy (%) across the datasets tested. Top: TDIDT approaches, Bottom: Non-Greedy Approaches. Best approach per block is **bolded** and second best is *italicized*.

(a) Axis-Aligned Approaches

| Model | 2 | 3 | 4 | 5 | 6 | Best |
|---|---|---|---|---|---|---|
| CART | 83.9 | 81.6 | 82.3 | 84.0 | 85.1 | 85.2 |
| SERDT | 83.7 | 80.9 | 81.7 | 83.7 | 85.1 | 85.1 |
| HSTree | 83.9 | 81.6 | 82.4 | 84.0 | 85.1 | 85.1 |
| SGT-C | *84.5* | *82.5* | *83.7* | *84.9* | *86.2* | *86.3* |
| SGT$_3$-C | **85.7** | **84.6** | **85.9** | **87.3** | **87.8** | **87.8** |
| AxTAO | 83.9 | 82.1 | 82.8 | 84.5 | 85.5 | 85.4 |
| DPDT | 84.9 | 83.4 | 83.8 | 85.2 | 86.2 | 86.2 |
| SPLIT | 81.7 | 76.6 | 77.9 | 79.5 | 80.2 | 80.3 |
| SGT-T | *85.0* | *83.5* | *84.6* | *85.9* | *86.8* | *86.8* |
| SGT$_3$-T | **86.4** | **85.1** | **86.2** | **87.5** | **88.0** | **88.0** |

(b) Bivariate Approaches

| Model | 2 | 3 | 4 | 5 | 6 | Best |
|---|---|---|---|---|---|---|
| BiCART | 87.3 | 87.6 | 87.9 | 88.7 | 89.6 | 89.6 |
| S$^2$GT-C | *89.3* | *89.1* | *89.5* | *90.5* | *91.3* | *91.4* |
| S$^2$GT$_3$-C | **90.0** | **90.2** | **91.1** | **91.8** | **91.5** | **91.7** |
| BiTAO | 87.9 | 88.1 | 88.4 | 89.3 | 90.1 | 90.0 |
| S$^2$GT-T | *89.7* | *90.0* | *90.2* | *91.2* | *91.7* | *91.6* |
| S$^2$GT$_3$-T | **90.2** | **90.6** | **91.2** | **91.7** | **92.4** | **91.9** |

**Results on Axis-Aligned Trees** Table 1a reports the average test accuracy of axis-aligned approaches across all datasets. We observe that SGT-C consistently outperforms all baseline TDIDT methods at every evaluated depth. Moreover, SGT$_3$-C achieves superior performance both overall and at each depth, highlighting the benefit of higher branching factors. It is important to note that despite the increase in the number of nodes, ternary trees have the same size of local explanation as binary trees of the same depth [13]. Among non-greedy methods, we observe that the TAO-refined SGTs attain the highest performance across all axis-aligned approaches. At the dataset level (Figure 2), SGT-C often matches or exceeds the maximum-depth accuracy of other TDIDT methods with substantially shallower trees. A similar, though less pronounced, pattern is observed for SGT-T when compared to deeper non-greedy methods.

**Results on Bivariate Trees** In our evaluation of bivariate approaches, we exclude Covertype, Eucalyptus, and Higgs as BiCART and BiTAO require more than the 32GB of available memory on these datasets. When looking at the average performance of the bivariate models on the remaining datasets (Table 1b), we observe that S$^2$GT-C and S$^2$GT-T consistently outperform BiCART and BiTAO, respectively, across all depths as well as overall. Like in the univariate case, we observe that a higher branching factor helps improve expressivity as both S$^2$GT$_3$-C and S$^2$GT$_3$-T outperform S$^2$GT-C and S$^2$GT-T respectively. We also observe that bivariate shape function branching can significantly compress needed tree size in many cases. More specifically, we observe that on the `eye-movements` and `electricity` datasets (See Figure G.4 in Appendix), S$^2$GT-C with a depth of two achieves similar performance to both BiTAO and BiCART with depths of six.

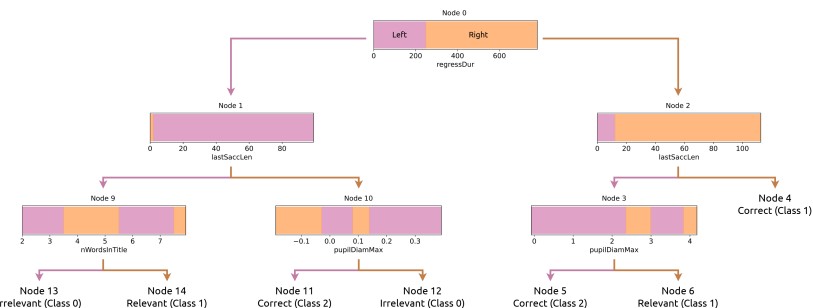

Figure 3: Visualization of a SGT induced via ShapeCART (max depth 3) trained on the `eye-movements` dataset. Purple indicates regions where samples will be directed to the left child; Orange directed to the right.

**Visualization**    To illustrate the interpretability of SGTs, we visualize a trained ShapeCART model on the `eye-movements` dataset [41] in Figure 3. We observe that multiple nodes benefit from non-linear shape functions. For example, the decisions represented in Nodes 3, 9, and 10 would each require at least three additional splits in a linear axis-aligned tree. Additional examples, including visualized Shape$^2$CART, ShapeCART$_3$, and Shape$^2$CART$_3$ trees, are provided in Appendix H.

## 6    Conclusion

This paper introduces the Shape Generalized Tree (SGT), a novel generalization of traditional linear decision trees that empowers each node with shape-function branching. Shape Generalized Trees retain the inherent interpretability that makes decision trees a leading approach as the shape functions learned at each node can be easily visualized, allowing users to understand an SGT's internal decision mechanisms. We also introduce ShapeCART, an efficient top-down induction algorithm to construct these SGT models. Additionally, we extend our framework to support bivariate SGTs (S$^2$GT) and multi-way SGTs (SGT$_K$), and present the corresponding ShapeCART variants (Shape$^2$CART and ShapeCART$_K$). Overall, we show that SGTs outperform linear trees on various datasets across various model sizes.

**Limitations**    As tree-based models, Shape Generalized Trees (SGTs) inherit intrinsic constraints of traditional decision trees, most notably that they are designed primarily for tabular data. Furthermore, interpreting the shape functions at each node can be more challenging than understanding axis-aligned linear splits, potentially increasing the cognitive effort required for analysis. Nevertheless, SGTs preserve essential interpretability properties, including simulability and modularity, while their enhanced expressiveness enables the construction of smaller trees without compromising predictive performance, thereby improving the sparsity of local explanations. Despite these advantages, further work is needed to systematically evaluate the interpretability of SGTs, and human-subject studies, similar to those conducted for GAMs [42], present a valuable direction for future research.

**Broader Impact**    SGTs constitute an inherently interpretable class of models, and our promising results inducing them via ShapeCART may inspire other future inherently interpretable approaches, thereby enhancing transparency and accountability in machine learning systems. Exploring alternative shape function representations representations, such as neural networks used in GAMs [18–20], and developing clustering SGTs [43–45] are promising directions for future work. Furthermore, our work opens interesting directions for future work to further theoretically characterize SGTs and ShapeCART, including extending theoretical properties of CART such as consistency rate [39, 46], the impact of sparsity [47], and tighter empirical risk bounds [39].

## Acknowledgments and Disclosure of Funding

This research was supported by grant number DSI-CGY3R1P18 from the Data Sciences Institute at the University of Toronto. The authors also gratefully acknowledge funding from the Natural Sciences and Engineering Research Council of Canada (NSERC).

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

# A Interpretability Discussion

Decision trees are widely regarded as modular and simulable models, offering model-based inter­pretability when their complexity is controlled through constraints on the number of features used [13, 34] and through structural restrictions such as limiting the total node count and maximum depth [4, 12, 13, 36]. As a decision tree variant, the SGT inherits this modularity, while the restriction that each internal node operates on only one or two features preserves simulability. In particular, each node's shape function can be directly visualized and queried to support transparent predictions. For example, consider the SGT trained on the eye-movements dataset (Figure 3). For a sample with `regressDur = 100`, `lastSaccLen = 0`, and `nWordsInTitle = 4`, the tree yields a local explanation of the form: "The model predicted (Relevant, Class 1) because regressDur $< 220$, lastSaccLen $< 5$, and nWordsInTitle $\in [4, 5]$." Moreover, the enhanced expressivity of SGTs enables them to achieve comparable or superior performance with smaller trees, thereby improving sparsity in the resulting explanations and aligning with the desiderata for interpretability proposed by Murdoch et al. [34]. Note that higher-order oblique trees (more than 2 features per node) are not naturally visualizable and are considered less interpretable [31]. We similarly excluded SGTs with higher-order shape functions beyond bivariate for the same reason.

One benefit of decision trees that the SGT inherits is that the local explanation length of an SGT is naturally bounded: a user needs to examine at most one piecewise-constant shape function per decision node along a path, bounding the explanation's size to the tree's depth. Consequently, interpretability is preserved only if the maximum depth is sufficiently small. For this reason, and following prior work on compact decision trees [13, 29, 30], we cap the depth of our models at six.

**SGTs and GAMs:** Despite being inspired by the feature specific transformations of GAMs, SGTs and GAMs are not directly comparable. In GAMs, a separate shape function is learned for each feature (and for each class in multi-class settings, yielding $|\mathcal{C}|$ shape functions per feature). As a result, the size of a local explanation in a GAM is identical to its global explanation, since all shape functions must be considered simultaneously. In contrast, decision trees bound the size of a local explanation by the depth of the tree, and the global explanation by the number of nodes (at most $2^d - 1$ for depth $d$). Due to these reasons, GAMs are not directly comparable to trees and we do not include them in our evaluation.

# B S²GT_K Formulation

**Definition 6 (Binary Multi-Way Shape Generalized Tree (S²GT_K)).** *In an $S^2GT_K$, internal nodes partition the data into at most $K$ distinct subsets based on the output of a vector-valued shape function that utilizes at most two features:*

$$\mathcal{D}^k = \{(\mathbf{x}_n, y_n) \in \mathcal{D} \mid \underset{k \in \{1,...,K\}}{\arg\max} \left( f_\Theta^{2,(K)}(\mathbf{w}_0^\top \mathbf{x}_n, \mathbf{w}_2^\top \mathbf{x}_n, \Theta) \right) = k;$$

$$\mathbf{w}_1, \mathbf{w}_2 \in \{0,1\}^D; \|\mathbf{w}_1\|_0, \|\mathbf{w}_2\|_0 = 1\} \tag{13}$$

*Here, $\mathbf{w}_1, \mathbf{w}_2$ are one-hot feature selection vectors where $d_1 = \arg\max \mathbf{w}_1, d_2 = \arg\max \mathbf{w}_2$ are the selected features, and $f_\Theta^{2,(K)} : dom(\mathcal{X}_{d_1}) \times dom(\mathcal{X}_{d_2}) \to \mathbb{R}^K$ is a shape function that maps the selected features to a $K$-dimensional vector, and the argmax operation determines the branch assignment.*

# C Proofs

## C.1 Expressiveness Guarantees

**Theorem 1.** *Every function that can be represented by a binary axis-aligned linear tree (Definition 1), can be represented by an SGT (Definition 3) with the same number of decision nodes.*

*Proof.* Given a binary axis-aligned linear tree, we construct an SGT with similar structure where the shape function in each decision node takes the form $f_\Theta(\mathbf{w}^\top \mathbf{x}) = \mathbf{w}^\top \mathbf{x}_n - \Theta$ where $\mathbf{w}$ and $\Theta$ match the values in the corresponding decision node in the axis-aligned linear tree. □

**Definition 7 ($\omega$-Bars Labeling Function).** *Let $\omega \in \mathbb{N}$ be a frequency parameter. We define the Bars Labeling function $g : [0, 1] \to \{0, 1\}$ as follows:*

$$g(x) = \begin{cases} 1 & \text{if } \cos(2\pi\omega x_1) \leq 0, \\ 0 & \text{otherwise}. \end{cases} \tag{14}$$

*The $\omega$-Bars Labeling Function partitions the unit interval $[0, 1]$ into alternating subintervals according to the sign of $\cos(2\pi\omega x)$. Within $[0, 1]$, the cosine function completes $\omega$ full oscillations, producing $\omega + 1$ zero-crossings and hence $\omega + 2$ disjoint subintervals of equal width. These subintervals alternate between label $1$ and label $0$.*

We show that an SGT can directly utilize nonlinear shape functions to represent all functions in this family in a fixed number of nodes, while the number of nodes required by an axis-aligned linear tree scales linearly with the number of rectangles utilized by each function.

**Lemma 2.** *The $\omega$-Bars labelling function can be represented by an SGT with one internal node for any value of $\omega$.*

*Proof.* The $\omega$-Bars Labelling function can be represented by an SGT in single node by utilizing a shape function of the form $f_\Theta(z) = \cos(2\pi\Theta z)$, where $\Theta$ is a learnable parameter. Recall that samples are assigned to the left partition if $f_\Theta(\mathbf{w}^\top \mathbf{x}_n) \leq 0.0$. As this is a univariate dataset, we select the only feature by setting $\mathbf{w} = [1]$. Additionally, we set $\Theta = \omega$, therefore samples satisfying $f_\Theta(x_1) = \cos(2\pi\omega x_1) \leq 0$ form the left subset, which is assigned to the positive class. The remaining points form the right subset and are assigned to the negative class. The resulting tree matches the definition of the Bars labelling function. $\square$

**Lemma 3.** *A binary axis-aligned linear tree requires at least $\omega + 1$ internal nodes to represent the Bars labelling function.*

*Proof.* A binary axis-aligned decision tree partitions the input space into disjoint intervals, each corresponding to a leaf [23]. Since each subinterval of $[0, 1]$ under $g$ is class-constant, and adjacent subintervals differ in label, every subinterval must correspond to a distinct leaf. Thus, a tree that represents $g$ requires at least $\omega + 2$ leaves. In a binary tree, the number of internal nodes is always one less than the number of leaves. Therefore, the exact representation of the $\omega$-Bars Labeling Function requires at least $\omega + 1$ internal nodes. $\square$

**Theorem 2.** *For every $B \in \mathbb{N}$, there exists a function for which a binary axis-aligned tree requires at least $B$ additional decision nodes to represent, compared to an SGT.*

*Proof.* For a given $\omega \in \mathbb{N}$, the binary axis-aligned linear tree requires $\omega + 1$ decision nodes while the SGT requires only one to perfectly represent the $\omega$-Bars labelling function. To obtain a function in which the binary axis-aligned linear tree requires at least $B$ additional decision nodes to represent, we can simply set $\omega = B$. $\square$

## C.2 Information Gain Guarantees

**Lemma 1.** *Let $t$ be an individual node in $T$, let $\mathcal{IG}(t)$ denote the information gain obtained by an axis-aligned (threshold) split on the samples in $t$ from CART, and let $\mathcal{IG}_f(t)$ denote the information gain obtained by a shape function split from ShapeCART on the samples in $t$. We show that $\mathcal{IG}_f(t) \geq \mathcal{IG}(t)$*

To construct the shape function $f_j$ for each feature $j$, we begin by fitting a univariate CART tree that partitions the data along $x_j$. The root node of this tree provides the best axis-aligned split, with associated information gain $\mathcal{IG}(t, j)$. For the initial bin-to-branch assignment, we consider two alternatives: (1) the left/right partition induced by the root split, and (2) a clustering-based assignment obtained via Weighted K-Means. By selecting the better of these two, we guarantee that the initialization of coordinate descent achieves at least $\mathcal{IG}(t, j)$. Since coordinate descent monotonically improves the objective, the resulting shape function satisfies $\mathcal{IG}_f(f_j, t) \geq \mathcal{IG}(t, j), \quad \forall j$. Because ShapeCART employs a line search to select the feature shape function, it follows that

$$\mathcal{IG}_f(t) = \max_j \mathcal{IG}_f(f_j, t) \geq \max_j \mathcal{IG}(t, j) = \mathcal{IG}(t).$$

# D  Time Complexity Analysis

In this section, we provide the time-complexity of splitting a node using a shape function with respect to the number of samples $N$, number of classes $C$, branching factor $K$, and number of bins (internal tree leaves) $L$.

**Internal Tree Construction**  To assign samples to $L$ bins, an internal tree is constructed using the CART algorithm [23], trained on a single feature and the target variable. We can break down the analysis of tree construction into four components: sorting the samples, tabulating class counts, calculating impurity, and the recursive relation.

1. **Sorting**: Sorting the samples takes $\mathcal{O}(N \log N)$ time. This only needs to be done once for the whole tree as we operate only on one feature and samples retain their relative ordering when assigned to children nodes. [48–50].

2. **Class Count Tabulation** Once the samples are sorted, CART calculates a cumulative class count at each split point. This involves a scan of all possible splits ($N$ at most).

3. **Impurity Calculation:** Once class counts are tabulated, we need to calculate the impurity for the class counts of each candidate split. Calculating the impurity of a single distribution scales linearly with $C$, which we repeat $N$ times, resulting in a complexity of $\mathcal{O}(NC)$.

4. **Recursive Relation**: To simplify our runtime analysis, we assume the induced decision tree remains approximately balanced (similar to [51]), such that its maximum depth is on the order of $\log L$, where $L$ denotes the number of leaves. Once a node is split, the cumulative class tabulations (Step 2) and impurity evaluations (Step 3) must be recomputed for the resulting child nodes. However, the total number of samples processed across all nodes at any given depth remains constant at $N$, since each split simply redistributes the same data among child nodes. Recall that Steps 2 and 3 have per-node complexities of $\mathcal{O}(N)$ and $\mathcal{O}(NC)$, respectively. Consequently, the total cost of class tabulation and impurity computation across all nodes at a given depth remains $\mathcal{O}(N+NC) \approx \mathcal{O}(NC)$. Aggregating over all levels of the tree yields a total complexity of $\mathcal{O}(NC \log L)$, which we can further simplify to $\mathcal{O}(NC \log N)$ as $L \leq N$ in all cases.

As result, the total complexity of fitting CART on a single feature is $\mathcal{O}(N \log N + NC \log N) \approx \mathcal{O}(NC \log N)$.

**Bin-to-Branch Optimization**  Mapping bins (tree leaves) to branches involves running Weighted K-Means on the empirical distributions of the $L$ bins to discover an initial assignment solution, then running coordinate descent to optimize the the assignments to minimize the weighted impurity of the resultant branches.

- For a given branching factor $k$, we run Weighted K-Means on the empirical class distributions of the $L$ leaves/bins to group the bins into $k$ groups. This is equivalent to running Weighted K-Means on $L$, $C$-dimensional samples, which has a time complexity of $\mathcal{O}(kLTC)$ where $T$ is the number of iterations the algorithm runs for [49, 52].

- Our coordinate descent procedure optimizes the partition assignment for each bin independently, iterating over all leaves $R$ times and evaluates the impurity for $k - 1$ values for each leaf assignment. The cost of testing a new branch assignment for a bin does not scale with the number of branches as we only need to recalculate the impurity and weights of the current and new branch while the impurity of the other branches are frozen and can be treated as one unit. Therefore, the number of impurity evaluations in our coordinate descent is $\approx kLR$. The cost of impurity evaluation scales linearly with the number of classes ($\mathcal{O}(C)$) for both Gini impurity and entropy, therefore the time complexity of our coordinate descent procedure is $\mathcal{O}(kLRC)$.

As we repeat K-Means and coordinate descent for $k = 2, \ldots, K$, the time complexity of the bin-to-branch optimization is $\mathcal{O}(K^2 LTC + K^2 LRC)$. As $T$ and $R$ are algorithmic constants, we remove them from consideration for simplicity resulting in a time complexity of $\mathcal{O}(K^2 LC)$.

The total time complexity of fitting a univariate shape function can then be obtained by adding up the two steps together, resulting in a complexity of $\mathcal{O}(NC \log N + K^2 LC)$. As we construct shape

functions for each of the $D$ features, the total time complexity of fitting an internal node is:

$$\mathcal{O}\left(D(NC\log N + K^2 LC)\right) \tag{15}$$

## D.1 Shape$^2$CART

For Shape$^2$CART and its variants, we utilize BiCART to construct the internal tree for each pair of features. This algorithm creates $H$ augmented features for each pair of features, then constructs a tree using this augmented feature and passes them into CART [31]. We can analyze the cost of internal tree construction using BiCART as a CART tree with $H$ features. The runtime of CART scales linearly with the number of features [48–51] as each step (sorting, tabulating class counts, and evaluating impurity) needs to be done once for each feature. As such, the complexity of fitting bicart is simply $\mathcal{O}(HNC\log N)$, which we can further simplify back to $\mathcal{O}(NC\log N)$ as we only consider small, fixed values of $H$. In the naive approach of constructing shape functions for each pair of features, the time-complexity of splitting a node is $\mathcal{O}\left(D^2(NC\log N + K^2 LC)\right)$.

When using our pairwise selection heuristic, the complexity can be broken down into the time to score each pair of features and the time to construct shape functions of the top $P$ pairs. To evaluate pairs of features, we take the Cartesian product of the branching decisions provided from fitting the univariate shape functions, resulting in $K^2$ sets. We then calculate the empirical class distribution of each partition, which can be done efficiently in a single scan of the data. We then evaluate the impurity of each of the $K^2$ partitions' empirical class distributions, resulting in a complexity of $\mathcal{O}(N + CK^2)$ for each pair of features, and $\mathcal{O}((N + CK^2)D^2)$ when aggregated across all pairs. We then take the top $P$ pairs of features and fit bivariate shape functions for these terms, resulting in a time complexity of:

$$\mathcal{O}\left(\underbrace{D(NC\log N + K^2 LC)}_{\text{Univariate Shape Function Construction}} + \underbrace{(N + CK^2)D^2}_{\text{Pair Tests}} + \underbrace{P\left(NC\log N + K^2 LC\right)}_{\text{Bivariate Shape Function Construction}}\right)$$

$$= \mathcal{O}\left((D + P)(NC\log N + K^2 LC) + (N + CK^2)D^2\right)$$

Empirical evaluation of the runtime benefits of limiting pairs is provided in Section G.2.3.

## E Dataset Information

Table 2: Dataset Information. Dimensionality refers to the number of columns after one-hot encoding.

| Dataset | $N$ | $|\mathcal{C}|$ | Dimensionality | $D$ | # Cat. | # Num./Binary |
|---|---|---|---|---|---|---|
| adult | 45222 | 2 | 108 | 13 | 11 | 2 |
| avila | 20867 | 12 | 10 | 10 | 0 | 10 |
| bank | 1372 | 2 | 4 | 4 | 0 | 4 |
| bean | 13611 | 7 | 16 | 16 | 0 | 16 |
| bidding | 6321 | 2 | 9 | 9 | 0 | 9 |
| covtype | 581012 | 7 | 52 | 12 | 2 | 10 |
| electricity | 45312 | 2 | 13 | 8 | 1 | 7 |
| eucalyptus | 736 | 5 | 1487 | 19 | 14 | 5 |
| eye-movements | 10936 | 3 | 27 | 27 | 0 | 27 |
| eye-state | 14980 | 2 | 14 | 14 | 0 | 14 |
| fault | 1941 | 7 | 27 | 27 | 0 | 27 |
| gas-drift | 13910 | 6 | 128 | 128 | 0 | 128 |
| higgs | 11,000,000 | 2 | 29 | 29 | 0 | 29 |
| htru | 17898 | 2 | 8 | 8 | 0 | 8 |
| magic | 13376 | 2 | 10 | 10 | 0 | 10 |
| mini-boone | 130064 | 2 | 50 | 50 | 0 | 50 |
| mushroom | 8124 | 2 | 94 | 21 | 16 | 5 |
| occupancy | 20560 | 2 | 5 | 5 | 0 | 5 |
| page | 5473 | 5 | 10 | 10 | 0 | 10 |
| pendigits | 10992 | 10 | 16 | 16 | 0 | 16 |

| Dataset | $N$ | $|\mathcal{C}|$ | Dimensionality | $D$ | # Cat. | # Num./Binary |
|---------|-----|------|----------------|-----|--------|---------------|
| raisin | 900 | 2 | 7 | 7 | 0 | 7 |
| rice | 3810 | 2 | 7 | 7 | 0 | 7 |
| room | 10129 | 4 | 16 | 16 | 0 | 16 |
| segment | 2310 | 7 | 18 | 18 | 0 | 18 |
| skin | 245057 | 2 | 3 | 3 | 0 | 3 |
| wilt | 4839 | 2 | 5 | 5 | 0 | 5 |

# F   Model Implementation Details

All models are implemented in Python. For CART, we utilize the implementation found in the Scikit-Learn [49]. For Axis-Aligned TAO and HSTree, we utilize the implementation found in the imodels [53] package. For SERDT, we utilize the implementation provided by the original authors [13] found at `https://github.com/user-anonymous-researcher/interpretable-dts`. For DPDT, we utilize the official implementation provided by the original authors [29] found at `https://github.com/KohlerHECTOR/DPDTreeEstimator`. For SPLIT, we utilize the official implementation provided by the original authors found at `https://github.com/VarunBabbar/SPLIT-ICML`. As noted in Section 2.1, some optimal methods, including GOSDT [27] and SPLIT [30] (which builds on GOSDT), require pre-binarization of continuous features. Similar to the setting from the original work [30], we set `binarize=True` when running SPLIT, which applies Threshold Guessing to produce a sparse set of binary features. While full binarization over all thresholds may improve performance, it is orders of magnitude more computationally expensive [30].

**Categorical Variables**   Since all baseline methods require numerical inputs, we one-hot encode categorical variables when relevant. Unlike the baselines that are limited to split on one (axis-aligned) or two (bivariate) levels at a time, we note that ShapeCART and its variants support superset branching on categorical variables owing to the flexibility of their shape functions. This is achieved by passing all relevant columns of the one-hot encoded categorical variable into the internal tree.

**BiCART and BiTAO**   As there are no available pre-existing implementations of the BiCART and BiTAO models, we do our best to replicate these approaches following the methods outlined by Kairgeldin and Carreira-Perpiñán [31] in their original paper. For each pair of features $(d_1, d_2)$, BiCART and BiTAO create $H$ pre-computed linear combination features. They do this by creating a small, fixed subset of line orientation $\mathbf{W} \in \mathbb{R}^{2 \times H}$, sampled uniformly in two directions by rotating it around the origin $H$ times, within a range of 0-180 degrees. The augmented feature set for $d_1, d_2$ is then computed as $\mathbf{X}^{\text{aug}}_{(d_1, d_2)} = \mathbf{X}_{(d_1, d_2)} \mathbf{W}$ where $\mathbf{X}_{(d_1, d_2)} \in \mathbb{R}^{N \times 2}$ is the matrix containing the relevant pair of features. In BiCART, we utilize this augmented feature along with the original features to induce a tree via the CART algorithm [31]. For BiTAO, we utilize TAO to refine a bivariate tree constructed by BiCART, finding the best linear split for nodes in a reverse depth order. For pseudocode of this procedure, we refer reads to Kairgeldin and Carreira-Perpiñán [31] Figure 3.

**ShapeTAO**   To implement ShapeTAO (and Shape$^2$TAO), we train a ShapeCART (and Shape$^2$CART) tree and refine it using TAO. Unlike Axis-Aligned TAO and BiTAO, our `FindBestSplit` function trains a decision tree via CART (BiCART for Shape$^2$TAO) on the "care" set of samples when refining each node. More formally, we follow the pseudocode provided by Kairgeldin and Carreira-Perpiñán [31] Figure 3 and modify the solving procedure of the reduced problem (Kairgeldin and Carreira-Perpiñán [31] Equation 3) from considering linear splits, to instead training a decision tree via CART/BiCART. For Shape$^2$TAO, we only consider pairs of features that were considered in the node under refinement during the initial induction process. To extend TAO to multi-way branching, we make two key changes:

1. In the standard TAO procedure, each sample is assigned to a single "correct" branch, which is then used as its pseudolabel. In the multi-way setting, however, a sample may correspond to multiple valid branches. To accommodate this, we modify the pseudolabeling procedure by duplicating each sample across its valid branches and passing the new upsampled dataset to CART.

2. To evaluate whether an updated decision function improves upon the previous one, we adapt the accuracy metric to reflect the multi-way setting. Specifically, a prediction is deemed correct if the decision function assigns the sample to any branch that is among the set of valid branches for that sample.

Source code for for ShapeCART, ShapeTAO, and its variants can be found at https://github.com/optimal-uoft/Empowering-DTs-via-Shape-Functions.

### F.1 Hyperparameter Information

For all models, we vary the maximum depth between 2-6. Below are the other parameters we tune for each model. Note that when models share hyperparameters, we utilize the same values across all modesl.

- **CART**: `criterion`: {gini, entropy}, `min_samples_split`: {2,4,8,16,32}, `min_samples_leaf`: [1,32], `min_impurity_decrease`: {0.0, 1e-4, 5e-4, 1e-3, 5e-3, 0.01}, `ccp_alpha`: {0.0, 1e-4, 5e-4, 1e-3, 5e-3, 0.01}

- **SERDT**: `min_samples_stop`: {2,4,8,16,32}, `gini_factor`: {0.9, 0.91, ..., 0.99}, `gamma_factor`: {0.5, 0.65, 0.8}

- **HSTree**: `criterion`: {gini, entropy}, `min_samples_split`: {2,4,8,16,32}, `min_samples_leaf`: [1,32], `min_impurity_decrease`: {0.0, 1e-4, 5e-4, 1e-3, 5e-3, 0.01}, `ccp_alpha`: {0.0, 1e-4, 5e-4, 1e-3, 5e-3, 0.01}, `reg_param` (different from TAO): {1e-2, 5e-2, 1e-1, 5e-1, 1.0, 5.0, 10, 50}

- **AxTAO**: `criterion`: {gini, entropy}, `min_samples_split`: {2,4,8,16,32}, `min_samples_leaf`: [1,32], `min_impurity_decrease`: {0.0, 1e-4, 5e-4, 1e-3, 5e-3, 0.01}, `reg_param/lambda_`: {0.0, 1e-4, 5e-4, 1e-3, 5e-3, 0.01}

- **DPDT**: `criterion`: {gini, entropy}, `min_samples_split`: {2,4,8,16,32}, `min_samples_leaf`: [1,32], `min_impurity_decrease`: {0.0, 1e-4, 5e-4, 1e-3, 5e-3, 0.01}, `cart_nodes_list`: {(8,3,), (32,), (4,3,), (16,6,), (16,4,), (4,4), (3,3), (6,6)}

- **SPLIT**: `reg`: {1e-3, 1e-2, .1}, `lookahead_depth`: {2,..., maximum depth of tree}, `gbdt_n_est`: {10, 20, ..., 100}

- **BiCART**: `criterion`: {gini, entropy}, `min_samples_split`: {2,4,8,16,32}, `min_samples_leaf`: [1,32], `min_impurity_decrease`: {0.0, 1e-4, 5e-4, 1e-3, 5e-3, 0.01}, `ccp_alpha`: {0.0, 1e-4, 5e-4, 1e-3, 5e-3, 0.01}, $H$: {5,6,7,8}

- **BiTAO**: `criterion`: {gini, entropy}, `min_samples_split`: {2,4,8,16,32}, `min_samples_leaf`: [1,32], `min_impurity_decrease`: {0.0, 1e-4, 5e-4, 1e-3, 5e-3, 0.01}, `reg_param/lambda_`: {0.0, 1e-4, 5e-4, 1e-3, 5e-3, 0.01}, $H$: {5,6,7,8}

- **ShapeCART/ ShapeCART$_3$**: `criterion`: {gini, entropy}, `min_samples_split`: {2,4,8,16,32}, `min_samples_leaf`: [1,32], `min_impurity_decrease`: {0.0, 1e-4, 5e-4, 1e-3, 5e-3, 0.01}, `inner_max_leaf_nodes` (aka $L$): {4,8,12,...,64}, `inner_min_samples_leaf`: {1, 1e-4, 5e-4, 1e-3, 5e-3, 1e-2}, `branching_penalty` (aka $\lambda$): {0.0, 1e-4, 5e-4, 1e-3, 5e-3, 0.01}

- **ShapeTAO / ShapeTAO$_3$**: `criterion`: {gini, entropy}, `min_samples_split`: {2,4,8,16,32}, `min_samples_leaf`: [1,32], `min_impurity_decrease`: {0.0, 1e-4, 5e-4, 1e-3, 5e-3, 0.01}, `reg_param/lambda_`: {0.0, 1e-4, 5e-4, 1e-3, 5e-3, 0.01}, `inner_max_leaf_nodes` (aka $L$): {4,8,12,...,64}, `inner_min_samples_leaf`: {1, 1e-4, 5e-4, 1e-3, 5e-3, 1e-2}

- **Shape$^2$CART / Shape$^2$CART$_3$**: `criterion`: {gini, entropy}, `min_samples_split`: {2,4,8,16,32}, `min_samples_leaf`: [1,32], `min_impurity_decrease`: {0.0, 1e-4, 5e-4, 1e-3, 5e-3, 0.01}, `inner_max_leaf_nodes` (aka $L$): {4,8,12,...,64}, `inner_min_samples_leaf`: {1, 1e-4, 5e-4, 1e-3, 5e-3, 1e-2}, `branching_penalty` (aka $\lambda$), $H$: {5,6,7,8}

- **Shape$^2$TAO / Shape$^2$TAO$_3$**: `criterion`: {gini, entropy}, `min_samples_split`: {2,4,8,16,32}, `min_samples_leaf`: [1,32], `min_impurity_decrease`: {0.0, 1e-4, 5e-4, 1e-3, 5e-3, 0.01}, `reg_param/lambda_`: {0.0, 1e-4, 5e-4, 1e-3, 5e-3, 0.01},

`inner_max_leaf_nodes` (aka $L$): {4,8,12,...,64}, `inner_min_samples_leaf`: {1, 1e-4, 5e-4, 1e-3, 5e-3, 1e-2}, $H$: {5,6,7,8}

Note on $H$: The choice of H was predicated on ensuring computational tractability. While prior work by [31] examined H values from 30 to 90, our evaluation revealed that these higher values imposed unmanageable memory burdens, preventing successful execution on any dataset. Accordingly, a more computationally conservative value for H was selected. Despite this restriction, three datasets (Eucalyptus, Higgs, and Covertype) still were not able to run on BiCART and BiTAO given our 32GB RAM limit.

### F.1.1 Hyperparameter Importance

In this section, we analyze the important hyperparameters for our approaches. Following [2] and [3], we normalize the test accuracy within each model and dataset and train a Random Forest regressor to predict test Z-scores from the model hyperparameters and provide the feature importance in Table 3

Table 3: Hyperparameter Importance of ShapeCART and its variants.

| | ShapeCART | ShapeCART$_3$ | Shape$^2$CART | Shape$^2$CART$_3$ | ShapeTAO | ShapeTAO$_3$ | Shape$^2$TAO | Shape$^2$TAO$_3$ |
|---|---|---|---|---|---|---|---|---|
| max_depth | 0.35 | 0.24 | 0.179 | 0.095 | 0.359 | 0.221 | 0.142 | 0.085 |
| criterion | 0.038 | 0.035 | 0.033 | 0.034 | 0.035 | 0.025 | 0.038 | 0.035 |
| min_samples_split | 0.079 | 0.082 | 0.08 | 0.081 | 0.075 | 0.075 | 0.078 | 0.069 |
| min_samples_leaf | 0.182 | 0.178 | 0.188 | 0.189 | 0.154 | 0.179 | 0.17 | 0.159 |
| min_impurity_decrease | 0.094 | 0.08 | 0.094 | 0.08 | 0.074 | 0.079 | 0.082 | 0.078 |
| inner_max_leaf_nodes | 0.169 | 0.192 | 0.167 | 0.19 | 0.137 | 0.172 | 0.154 | 0.195 |
| inner_min_samples_leaf | 0.088 | 0.098 | 0.094 | 0.088 | 0.082 | 0.084 | 0.096 | 0.089 |
| branching_penalty | — | 0.095 | — | 0.088 | — | 0.079 | — | 0.082 |
| H | — | — | 0.066 | 0.062 | — | — | 0.069 | 0.054 |
| pairwise_penalty | — | — | 0.098 | 0.094 | — | — | 0.088 | 0.084 |

# G Extra results

In this section, we provide results for all approaches across all datasets as well as results from more empirical evaluations.

## G.1 Full Results

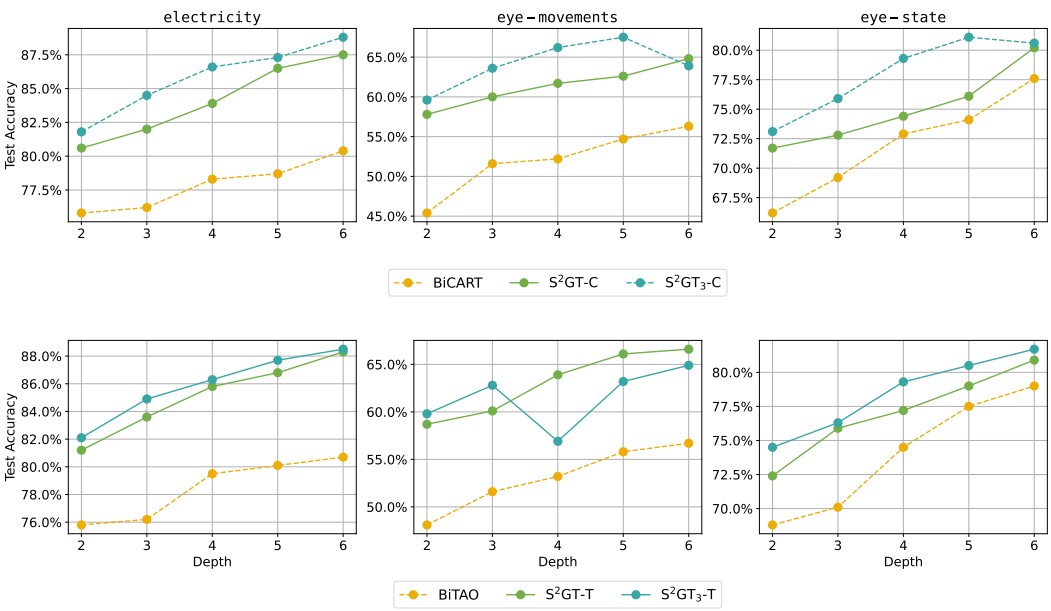

Figure G.4: Test accuracy per depth for Bivariate models on `eye-movements`, `electricity`, and `eye-state`. Top: TDIDT-Approaches. Bottom: Non-Greedy Approaches.

Table 4: Test Accuracy (% ± std.) for all axis-aligned approaches separated into TDIDT and Non-Greedy approaches by dashed lines per dataset. Best approach is **bolded**, second-best is *italicized*.

| dataset | model | 2 | 3 | 4 | 5 | 6 | Best |
|---------|-------|---|---|---|---|---|------|
| adult | CART | *82.60 ± 0.20* | 82.60 ± 0.30 | 83.20 ± 0.30 | 83.60 ± 0.30 | 83.60 ± 0.30 | 83.60 ± 0.30 |
| | HSTree | *82.60 ± 0.20* | 82.60 ± 0.30 | 83.20 ± 0.30 | 83.60 ± 0.30 | 83.70 ± 0.50 | 83.70 ± 0.50 |
| | SERDT | *82.60 ± 0.20* | 82.60 ± 0.20 | 82.60 ± 0.20 | 83.00 ± 0.20 | 83.10 ± 0.30 | 83.10 ± 0.30 |
| | SGT-C | **82.80 ± 0.20** | **84.10 ± 0.30** | *84.00 ± 0.30* | *84.30 ± 0.40* | *84.60 ± 0.60* | *84.60 ± 0.60* |
| | SGT$_3$-C | **82.80 ± 0.20** | **84.10 ± 0.30** | **84.10 ± 0.40** | **84.50 ± 0.60** | **84.80 ± 0.40** | **84.80 ± 0.40** |
| | AxTAO | 82.60 ± 0.20 | 82.60 ± 0.30 | 83.30 ± 0.30 | 83.50 ± 0.50 | 83.90 ± 0.20 | 83.90 ± 0.20 |
| | DPDT | 82.60 ± 0.20 | 83.20 ± 0.20 | 83.50 ± 0.40 | 83.50 ± 0.30 | 83.90 ± 0.40 | 83.90 ± 0.40 |
| | SPLIT | 78.00 ± 0.30 | 82.60 ± 0.20 | 83.50 ± 0.30 | 84.00 ± 0.30 | 84.20 ± 0.30 | 84.20 ± 0.30 |
| | SGT-T | **82.90 ± 0.20** | **84.10 ± 0.30** | *84.10 ± 0.30* | *84.40 ± 0.20* | *84.80 ± 0.60* | *84.80 ± 0.60* |
| | SGT$_3$-T | **82.90 ± 0.20** | **84.10 ± 0.30** | **84.50 ± 0.40** | **84.80 ± 0.50** | **85.00 ± 0.50** | **85.00 ± 0.50** |
| avila | CART | — | — | 57.60 ± 1.20 | 62.30 ± 0.90 | 65.20 ± 0.90 | 65.20 ± 0.90 |
| | HSTree | — | — | 57.60 ± 1.20 | 62.30 ± 0.90 | 65.20 ± 0.90 | 65.20 ± 0.90 |
| | SERDT | — | — | 58.10 ± 0.30 | 61.90 ± 0.70 | 66.60 ± 1.90 | 66.60 ± 1.90 |
| | SGT-C | — | — | *64.50 ± 2.90* | *71.70 ± 1.80* | *82.00 ± 2.10* | *82.00 ± 2.10* |
| | SGT$_3$-C | — | — | **87.00 ± 1.50** | **98.00 ± 0.50** | **99.70 ± 0.10** | **99.70 ± 0.10** |
| | AxTAO | — | — | 58.00 ± 0.60 | 64.10 ± 1.10 | 67.00 ± 0.80 | 67.00 ± 0.80 |
| | DPDT | — | — | 61.00 ± 0.60 | 64.70 ± 1.00 | 68.70 ± 0.00 | 68.70 ± 0.00 |
| | SPLIT | — | — | 52.90 ± 0.30 | 56.50 ± 0.40 | 58.30 ± 1.10 | 58.30 ± 1.10 |
| | SGT-T | — | — | *69.60 ± 1.40* | *77.00 ± 0.60* | *84.40 ± 1.70* | *84.40 ± 1.70* |
| | SGT$_3$-T | — | — | **87.50 ± 1.80** | **98.10 ± 0.60** | **99.60 ± 0.30** | **99.60 ± 0.30** |
| bank | CART | *89.10 ± 1.30* | *94.40 ± 0.40* | 95.80 ± 0.60 | 96.00 ± 2.00 | **97.90 ± 0.90** | **97.90 ± 0.90** |
| | HSTree | *89.10 ± 1.30* | *94.40 ± 0.40* | 94.80 ± 2.00 | 96.20 ± 1.80 | **97.90 ± 0.90** | 96.20 ± 1.80 |
| | SERDT | 88.80 ± 1.30 | 93.20 ± 2.60 | 94.70 ± 2.00 | 96.10 ± 2.10 | *97.30 ± 1.30* | *97.30 ± 1.30* |

| Dataset | Method | | | | | | |
|---|---|---|---|---|---|---|---|
| | SGT-C | 88.50 ± 2.30 | 89.50 ± 3.20 | **97.00 ± 0.90** | *96.70 ± 0.00* | 96.70 ± 0.00 | 97.00 ± 0.90 |
| | SGT$_3$-C | **90.90 ± 0.70** | **97.00 ± 0.80** | *96.80 ± 1.90* | **96.80 ± 1.90** | 96.80 ± 1.90 | 96.80 ± 1.90 |
| | AxTAO | *88.80 ± 1.30* | 95.40 ± 0.60 | 96.50 ± 1.40 | *97.70 ± 1.80* | **98.80 ± 0.20** | 97.70 ± 1.80 |
| | DPDT | 88.50 ± 1.50 | *95.60 ± 1.00* | **97.80 ± 1.00** | 97.90 ± 1.20 | *98.40 ± 0.90* | **98.40 ± 0.90** |
| | SPLIT | 82.70 ± 0.60 | 90.10 ± 1.40 | 96.50 ± 0.60 | 97.20 ± 0.80 | 97.10 ± 1.00 | 97.20 ± 0.80 |
| | SGT-T | *88.80 ± 1.30* | 92.00 ± 4.20 | **97.80 ± 1.00** | *97.70 ± 0.80* | 97.90 ± 0.60 | *97.70 ± 0.80* |
| | SGT$_3$-T | **91.60 ± 1.30** | **96.40 ± 1.10** | *97.30 ± 1.40* | 97.30 ± 1.40 | 97.30 ± 1.40 | 97.30 ± 1.40 |
| bean | CART | — | 79.90 ± 0.40 | 85.60 ± 0.10 | 89.60 ± 0.70 | *90.30 ± 0.50* | 90.30 ± 0.50 |
| | HSTree | — | 79.90 ± 0.40 | 85.60 ± 0.10 | 89.60 ± 0.60 | *90.30 ± 0.40* | 90.30 ± 0.40 |
| | SERDT | — | 77.10 ± 0.60 | 83.00 ± 0.70 | 88.20 ± 0.60 | **90.70 ± 0.50** | **90.70 ± 0.50** |
| | SGT-C | — | *85.10 ± 1.20* | 88.80 ± 0.40 | 89.70 ± 0.40 | 89.80 ± 0.10 | 89.80 ± 0.10 |
| | SGT$_3$-C | — | **87.90 ± 0.50** | 90.40 ± 0.60 | 90.50 ± 0.20 | *90.30 ± 0.50* | 90.30 ± 0.50 |
| | AxTAO | — | 81.90 ± 0.40 | 87.20 ± 0.10 | 90.00 ± 0.50 | 90.40 ± 0.40 | 90.40 ± 0.40 |
| | DPDT | — | 85.10 ± 0.30 | *89.80 ± 0.30* | *90.80 ± 0.80* | **90.90 ± 0.20** | **90.90 ± 0.20** |
| | SPLIT | — | 55.80 ± 0.40 | 66.80 ± 1.80 | 71.20 ± 1.00 | 72.80 ± 1.10 | 72.80 ± 1.10 |
| | SGT-T | — | 85.70 ± 1.10 | 88.80 ± 0.60 | 90.10 ± 0.80 | *90.80 ± 0.70* | 90.80 ± 0.70 |
| | SGT$_3$-T | — | **88.20 ± 0.60** | 90.40 ± 0.70 | 90.90 ± 0.40 | 90.00 ± 0.30 | **90.90 ± 0.40** |
| bidding | CART | *98.10 ± 0.60* | 98.10 ± 0.60 | 98.20 ± 0.50 | **99.70 ± 0.30** | 99.60 ± 0.20 | **99.70 ± 0.30** |
| | HSTree | *98.10 ± 0.60* | 98.10 ± 0.60 | 98.20 ± 0.50 | **99.70 ± 0.30** | 99.60 ± 0.30 | **99.70 ± 0.30** |
| | SERDT | *98.10 ± 0.60* | 98.10 ± 0.60 | 99.60 ± 0.10 | **99.70 ± 0.30** | 99.70 ± 0.10 | 99.70 ± 0.10 |
| | SGT-C | 98.10 ± 0.70 | 98.40 ± 0.60 | **99.70 ± 0.20** | *99.60 ± 0.30* | 99.60 ± 0.10 | 99.60 ± 0.10 |
| | SGT$_3$-C | **99.10 ± 0.10** | **99.60 ± 0.20** | 99.60 ± 0.20 | 99.60 ± 0.20 | 99.60 ± 0.20 | 99.60 ± 0.20 |
| | AxTAO | *98.10 ± 0.60* | 98.20 ± 0.60 | 98.30 ± 0.40 | 99.50 ± 0.00 | 99.50 ± 0.20 | 99.50 ± 0.00 |
| | DPDT | *98.10 ± 0.60* | **99.50 ± 0.10** | **99.70 ± 0.30** | 99.60 ± 0.10 | **99.80 ± 0.10** | **99.70 ± 0.30** |
| | SPLIT | 97.30 ± 0.60 | 98.10 ± 0.60 | 99.40 ± 0.10 | **99.70 ± 0.30** | 99.70 ± 0.30 | **99.70 ± 0.30** |
| | SGT-T | *98.10 ± 0.60* | 98.50 ± 0.70 | *99.40 ± 0.20* | 99.60 ± 0.10 | 99.60 ± 0.20 | 99.40 ± 0.20 |
| | SGT$_3$-T | **99.10 ± 0.10** | 99.30 ± 0.20 | 99.40 ± 0.20 | 99.40 ± 0.20 | 99.40 ± 0.20 | 99.40 ± 0.20 |
| covtype | CART | — | 67.70 ± 0.00 | 70.10 ± 0.10 | 70.20 ± 0.10 | 71.50 ± 0.00 | 71.50 ± 0.00 |
| | HSTree | — | 67.70 ± 0.00 | 70.10 ± 0.10 | 70.20 ± 0.10 | 71.50 ± 0.00 | 71.50 ± 0.00 |
| | SERDT | — | 67.40 ± 0.00 | 69.00 ± 0.10 | 69.90 ± 0.30 | 71.80 ± 0.20 | 71.80 ± 0.20 |
| | SGT-C | — | *68.90 ± 0.20* | *70.40 ± 0.10* | *72.10 ± 0.50* | *73.70 ± 0.20* | *73.70 ± 0.20* |
| | SGT$_3$-C | — | **71.00 ± 0.20** | **72.60 ± 0.20** | **75.40 ± 0.20** | **78.10 ± 0.30** | **78.10 ± 0.30** |
| | AxTAO | — | 67.90 ± 0.10 | 70.10 ± 0.10 | 70.30 ± 0.10 | 72.00 ± 0.20 | 72.00 ± 0.20 |
| | DPDT | — | 68.60 ± 0.00 | 70.80 ± 0.10 | 71.60 ± 0.20 | 73.20 ± 0.20 | 73.20 ± 0.20 |
| | SPLIT | — | 63.60 ± 0.10 | 63.60 ± 0.10 | 63.60 ± 0.10 | 63.60 ± 0.10 | 63.60 ± 0.10 |
| | SGT-T | — | *69.50 ± 0.20* | *70.90 ± 0.50* | *72.50 ± 0.40* | *74.40 ± 0.20* | *74.40 ± 0.20* |
| | SGT$_3$-T | — | **71.20 ± 0.10** | **73.30 ± 0.30** | **76.20 ± 0.00** | **79.30 ± 0.40** | **79.30 ± 0.40** |
| electricity | CART | *75.50 ± 0.10* | 75.80 ± 0.20 | 76.00 ± 0.20 | 77.60 ± 0.20 | 78.00 ± 0.10 | 78.00 ± 0.10 |
| | HSTree | *75.50 ± 0.10* | 75.80 ± 0.20 | 76.00 ± 0.20 | 77.70 ± 0.10 | 78.10 ± 0.10 | 78.10 ± 0.10 |
| | SERDT | *75.50 ± 0.10* | 75.80 ± 0.20 | 76.20 ± 0.10 | 78.60 ± 0.80 | 79.40 ± 0.30 | 79.40 ± 0.30 |
| | SGT-C | *75.50 ± 0.10* | *78.50 ± 0.40* | *81.30 ± 0.80* | *84.40 ± 0.10* | *84.80 ± 0.60* | *84.80 ± 0.60* |
| | SGT$_3$-C | **76.50 ± 0.30** | **80.60 ± 0.60** | **84.00 ± 1.20** | **86.00 ± 0.40** | **87.80 ± 1.00** | **87.80 ± 1.00** |
| | AxTAO | 75.50 ± 0.10 | 75.90 ± 0.10 | 75.90 ± 0.30 | 78.50 ± 0.80 | 79.90 ± 0.20 | 79.90 ± 0.20 |
| | DPDT | *76.10 ± 0.20* | 78.30 ± 0.20 | 79.10 ± 0.20 | 80.00 ± 0.30 | 80.80 ± 0.50 | 80.80 ± 0.50 |
| | SPLIT | 75.40 ± 0.10 | 76.20 ± 0.10 | 77.70 ± 0.50 | 78.40 ± 0.20 | 78.70 ± 0.30 | 78.70 ± 0.30 |
| | SGT-T | 75.50 ± 0.10 | *79.50 ± 0.40* | *83.00 ± 0.40* | *86.00 ± 0.40* | *87.60 ± 0.50* | *87.60 ± 0.50* |
| | SGT$_3$-T | **79.30 ± 0.30** | **82.60 ± 0.60** | **84.90 ± 0.60** | **86.70 ± 0.60** | **88.20 ± 0.30** | **88.20 ± 0.30** |
| eucalyptus | CART | — | 50.90 ± 4.50 | 52.50 ± 3.80 | 52.00 ± 5.40 | *55.20 ± 5.00* | *55.20 ± 5.00* |
| | HSTree | — | 50.90 ± 4.50 | 52.50 ± 3.80 | 52.00 ± 5.40 | 54.10 ± 5.10 | 54.10 ± 5.10 |
| | SERDT | — | 46.80 ± 2.70 | 44.60 ± 1.80 | 47.10 ± 3.70 | 51.40 ± 1.80 | 51.40 ± 1.80 |
| | SGT-C | — | *53.60 ± 3.30* | **55.90 ± 4.00** | **56.30 ± 4.60** | **58.80 ± 7.20** | **58.80 ± 7.20** |
| | SGT$_3$-C | — | **55.00 ± 2.70** | *53.40 ± 4.10* | *55.40 ± 6.80* | 54.70 ± 4.90 | 53.40 ± 4.10 |
| | AxTAO | — | *50.90 ± 4.50* | 51.80 ± 5.50 | 52.00 ± 5.40 | 54.70 ± 3.80 | 54.70 ± 3.80 |
| | DPDT | — | 49.80 ± 4.90 | *52.50 ± 3.80* | *53.80 ± 3.20* | 55.20 ± 3.00 | 55.20 ± 3.00 |
| | SPLIT | — | 29.30 ± 2.60 | 36.70 ± 2.10 | 36.30 ± 2.20 | 34.20 ± 4.50 | 34.20 ± 4.50 |
| | SGT-T | — | **55.40 ± 4.10** | **57.20 ± 2.00** | **57.00 ± 3.00** | *55.20 ± 4.80* | *55.20 ± 4.80* |
| | SGT$_3$-T | — | **55.40 ± 1.80** | 50.00 ± 2.90 | **57.00 ± 2.80** | 57.00 ± 2.80 | 55.40 ± 1.80 |
| | CART | 45.90 ± 0.30 | 46.30 ± 2.60 | 51.30 ± 0.20 | 52.40 ± 1.40 | 53.20 ± 0.70 | 53.20 ± 0.70 |
| | HSTree | 45.90 ± 0.30 | 46.30 ± 2.60 | 51.40 ± 0.20 | 52.40 ± 1.40 | 53.10 ± 0.70 | 53.10 ± 0.70 |

| Dataset | Method | | | | | | |
|---|---|---|---|---|---|---|---|
| | SERDT | 43.80 ± 0.30 | 49.20 ± 0.20 | 49.10 ± 0.40 | 52.70 ± 1.00 | 53.50 ± 0.60 | 53.50 ± 0.60 |
| | SGT-C | 48.10 ± 2.10 | 53.00 ± 0.70 | 54.90 ± 1.20 | 53.90 ± 0.70 | 57.40 ± 1.20 | 57.40 ± 1.20 |
| | SGT$_3$-C | 47.10 ± 0.50 | 54.40 ± 0.80 | 56.00 ± 0.30 | 57.50 ± 1.60 | 60.90 ± 2.60 | 60.90 ± 2.60 |
| | AxTAO | 45.90 ± 0.30 | 48.60 ± 0.80 | 51.40 ± 0.30 | 52.30 ± 1.20 | 52.60 ± 1.60 | 52.60 ± 1.60 |
| | DPDT | 48.70 ± 0.40 | 51.20 ± 0.30 | 52.40 ± 0.80 | 54.40 ± 1.20 | 56.70 ± 1.20 | 56.70 ± 1.20 |
| | SPLIT | 44.00 ± 0.70 | 48.40 ± 0.30 | 50.20 ± 0.60 | 51.90 ± 0.20 | 52.00 ± 0.50 | 51.90 ± 0.20 |
| | SGT-T | 48.60 ± 2.40 | 53.10 ± 1.50 | 54.80 ± 0.50 | 56.80 ± 2.80 | 58.00 ± 1.40 | 58.00 ± 1.40 |
| | SGT$_3$-T | 51.40 ± 0.90 | 54.80 ± 0.30 | 55.40 ± 0.80 | 56.90 ± 2.50 | 60.30 ± 2.90 | 60.30 ± 2.90 |
| eye-state | CART | 62.70 ± 0.60 | 65.80 ± 1.30 | 69.10 ± 1.60 | 70.90 ± 0.30 | 73.20 ± 1.40 | 73.20 ± 1.40 |
| | HSTree | 62.70 ± 0.60 | 65.80 ± 1.30 | 69.10 ± 1.60 | 70.90 ± 0.30 | 73.20 ± 1.40 | 73.20 ± 1.40 |
| | SERDT | 62.70 ± 0.60 | 66.40 ± 1.50 | 69.40 ± 0.90 | 72.30 ± 0.70 | 75.30 ± 0.50 | 75.30 ± 0.50 |
| | SGT-C | 64.30 ± 0.90 | 69.00 ± 0.20 | 70.50 ± 1.20 | 72.40 ± 1.30 | 74.50 ± 1.10 | 74.50 ± 1.10 |
| | SGT$_3$-C | 66.10 ± 0.70 | 70.80 ± 1.40 | 73.30 ± 1.00 | 76.80 ± 1.30 | 77.50 ± 0.50 | 77.50 ± 0.50 |
| | AxTAO | 62.10 ± 1.40 | 67.00 ± 1.00 | 69.70 ± 1.20 | 72.80 ± 0.70 | 74.30 ± 1.60 | 74.30 ± 1.60 |
| | DPDT | 65.90 ± 0.50 | 69.10 ± 1.90 | 71.40 ± 1.30 | 73.40 ± 0.50 | 75.60 ± 0.90 | 75.60 ± 0.90 |
| | SPLIT | 58.50 ± 0.40 | 66.40 ± 0.30 | 69.00 ± 1.30 | 71.80 ± 1.10 | 73.60 ± 0.30 | 73.60 ± 0.30 |
| | SGT-T | 65.60 ± 1.20 | 70.50 ± 0.60 | 72.70 ± 0.40 | 73.60 ± 1.20 | 75.80 ± 0.90 | 75.80 ± 0.90 |
| | SGT$_3$-T | 67.10 ± 0.50 | 71.90 ± 0.60 | 75.80 ± 0.60 | 75.80 ± 0.90 | 76.30 ± 1.20 | 76.30 ± 1.20 |
| fault | CART | — | 61.50 ± 4.50 | 66.30 ± 1.30 | 67.10 ± 1.60 | 70.30 ± 2.30 | 70.30 ± 2.30 |
| | HSTree | — | 61.50 ± 4.50 | 66.30 ± 1.30 | 67.60 ± 2.30 | 70.30 ± 2.30 | 70.30 ± 2.30 |
| | SERDT | — | 63.60 ± 0.90 | 64.80 ± 0.80 | 67.00 ± 1.20 | 70.50 ± 1.00 | 70.50 ± 1.00 |
| | SGT-C | — | 59.70 ± 3.90 | 65.30 ± 1.80 | 68.00 ± 0.40 | 69.70 ± 3.60 | 69.70 ± 3.60 |
| | SGT$_3$-C | — | 65.20 ± 5.10 | 69.60 ± 2.30 | 69.70 ± 1.30 | 69.10 ± 3.40 | 69.10 ± 3.40 |
| | AxTAO | — | 61.50 ± 4.50 | 66.40 ± 1.70 | 68.40 ± 2.00 | 69.80 ± 1.30 | 69.80 ± 1.30 |
| | DPDT | — | 67.40 ± 2.10 | 66.90 ± 2.20 | 71.00 ± 2.10 | 70.90 ± 2.80 | 70.90 ± 2.80 |
| | SPLIT | — | 47.00 ± 2.30 | 59.50 ± 1.00 | 61.90 ± 2.10 | 67.10 ± 3.50 | 67.10 ± 3.50 |
| | SGT-T | — | 65.70 ± 1.40 | 68.00 ± 3.30 | 70.10 ± 3.10 | 70.70 ± 5.20 | 70.70 ± 5.20 |
| | SGT$_3$-T | — | 69.30 ± 1.60 | 70.40 ± 2.20 | 69.20 ± 2.50 | 69.20 ± 3.40 | 69.20 ± 3.40 |
| gas-drift | CART | — | 65.40 ± 0.80 | 72.30 ± 2.10 | 81.00 ± 0.10 | 86.90 ± 1.70 | 86.90 ± 1.70 |
| | HSTree | — | 65.40 ± 0.80 | 72.30 ± 2.10 | 81.00 ± 0.10 | 86.80 ± 1.60 | 86.80 ± 1.60 |
| | SERDT | — | 65.50 ± 0.30 | 74.70 ± 1.00 | 81.70 ± 0.10 | 87.60 ± 0.90 | 87.60 ± 0.90 |
| | SGT-C | — | 66.30 ± 1.30 | 76.00 ± 1.00 | 83.10 ± 2.00 | 89.00 ± 0.90 | 89.00 ± 0.90 |
| | SGT$_3$-C | — | 81.00 ± 1.00 | 87.90 ± 1.90 | 93.00 ± 1.60 | 94.70 ± 1.20 | 94.70 ± 1.20 |
| | AxTAO | — | 66.90 ± 1.00 | 76.10 ± 1.80 | 83.50 ± 2.90 | 89.50 ± 1.30 | 89.50 ± 1.30 |
| | DPDT | — | 70.20 ± 0.80 | 78.50 ± 1.30 | 86.60 ± 0.80 | 91.90 ± 0.90 | 91.90 ± 0.90 |
| | SPLIT | — | 51.20 ± 0.90 | 59.00 ± 1.00 | 64.50 ± 1.80 | 69.00 ± 1.50 | 69.00 ± 1.50 |
| | SGT-T | — | 70.90 ± 3.30 | 80.00 ± 1.60 | 86.40 ± 0.70 | 91.10 ± 1.30 | 91.10 ± 1.30 |
| | SGT$_3$-T | — | 83.00 ± 0.60 | 89.20 ± 0.80 | 92.80 ± 1.60 | 94.80 ± 0.30 | 94.80 ± 0.30 |
| higgs | CART | 64.70 ± 0.30 | 65.60 ± 0.60 | 66.00 ± 0.80 | 66.30 ± 1.20 | 67.60 ± 0.30 | 67.60 ± 0.30 |
| | HSTree | 64.70 ± 0.20 | 65.60 ± 0.70 | 66.00 ± 0.80 | 66.50 ± 1.20 | 67.10 ± 0.20 | 67.10 ± 0.20 |
| | SERDT | 64.70 ± 0.20 | 65.60 ± 0.60 | 65.70 ± 1.30 | 66.20 ± 1.10 | 66.80 ± 0.40 | 66.80 ± 0.40 |
| | SGT-C | 64.40 ± 0.20 | 63.50 ± 0.50 | 64.10 ± 1.70 | 64.90 ± 1.30 | 63.70 ± 1.80 | 64.90 ± 1.70 |
| | SGT$_3$-C | 63.50 ± 0.70 | 64.70 ± 0.70 | 63.30 ± 1.20 | 63.00 ± 0.90 | 64.30 ± 0.60 | 64.30 ± 0.90 |
| | AxTAO | 62.60 ± 0.20 | 64.90 ± 0.30 | 65.00 ± 0.70 | 67.40 ± 0.40 | 65.40 ± 0.40 | 65.40 ± 0.40 |
| | DPDT | 64.70 ± 0.60 | 66.80 ± 0.90 | 65.50 ± 1.80 | 65.50 ± 0.60 | 66.40 ± 1.40 | 66.40 ± 1.40 |
| | SPLIT | 61.90 ± 0.60 | 65.10 ± 0.30 | 65.50 ± 1.10 | 66.80 ± 1.50 | 66.00 ± 2.00 | 66.00 ± 2.00 |
| | SGT-T | 63.80 ± 0.10 | 64.60 ± 1.00 | 65.90 ± 1.30 | 65.00 ± 2.10 | 63.90 ± 2.60 | 63.90 ± 2.60 |
| | SGT$_3$-T | 62.90 ± 0.10 | 65.10 ± 0.60 | 64.70 ± 0.90 | 63.70 ± 1.00 | 64.30 ± 1.00 | 64.30 ± 1.00 |
| htru | CART | 97.80 ± 0.10 | 97.80 ± 0.20 | 97.90 ± 0.10 | 97.90 ± 0.10 | 97.80 ± 0.00 | 97.90 ± 0.10 |
| | HSTree | 97.80 ± 0.10 | 97.80 ± 0.10 | 98.00 ± 0.10 | 97.90 ± 0.10 | 97.80 ± 0.00 | 97.90 ± 0.10 |
| | SERDT | 97.80 ± 0.10 | 97.80 ± 0.10 | 98.00 ± 0.10 | 98.00 ± 0.10 | 98.00 ± 0.10 | 98.00 ± 0.10 |
| | SGT-C | 97.80 ± 0.10 | 97.80 ± 0.10 | 97.80 ± 0.20 | 97.90 ± 0.20 | 97.60 ± 0.10 | 97.80 ± 0.10 |
| | SGT$_3$-C | 97.70 ± 0.10 | 97.70 ± 0.10 | 97.80 ± 0.10 | 97.90 ± 0.20 | 97.70 ± 0.10 | 97.80 ± 0.10 |
| | AxTAO | 97.80 ± 0.10 | 97.80 ± 0.10 | 98.00 ± 0.20 | 97.90 ± 0.10 | 97.90 ± 0.10 | 97.90 ± 0.10 |
| | DPDT | 98.00 ± 0.10 | 97.80 ± 0.10 | 98.00 ± 0.10 | 97.90 ± 0.10 | 97.90 ± 0.10 | 98.00 ± 0.10 |
| | SPLIT | 97.80 ± 0.10 | 97.80 ± 0.10 | 97.90 ± 0.20 | 98.00 ± 0.10 | 98.00 ± 0.10 | 98.00 ± 0.10 |
| | SGT-T | 97.70 ± 0.10 | 97.80 ± 0.10 | 97.80 ± 0.10 | 97.80 ± 0.10 | 97.70 ± 0.10 | 97.70 ± 0.10 |
| | SGT$_3$-T | 97.70 ± 0.10 | 97.70 ± 0.10 | 97.70 ± 0.10 | 97.70 ± 0.10 | 97.70 ± 0.10 | 97.70 ± 0.10 |
| magic | CART | 73.70 ± 1.20 | 76.80 ± 0.90 | 78.90 ± 0.70 | 80.10 ± 0.90 | 81.30 ± 0.90 | 81.30 ± 0.90 |

|  | | | | | | |
|---|---|---|---|---|---|---|
| HSTree | $73.70 \pm 1.20$ | $76.80 \pm 0.90$ | $78.90 \pm 0.70$ | *$80.10 \pm 0.90$* | *$81.30 \pm 0.90$* | **$81.30 \pm 0.90$** |
| SERDT | $73.70 \pm 1.20$ | $76.70 \pm 1.00$ | $78.70 \pm 0.80$ | $80.00 \pm 1.20$ | $81.10 \pm 0.80$ | *$81.10 \pm 0.80$* |
| SGT-C | *$73.90 \pm 1.00$* | *$77.10 \pm 0.10$* | *$79.60 \pm 1.10$* | $80.10 \pm 1.70$ | $81.00 \pm 0.50$ | $81.00 \pm 0.50$ |
| SGT$_3$-C | **$77.90 \pm 1.20$** | **$78.40 \pm 0.80$** | **$79.90 \pm 0.50$** | **$81.10 \pm 0.60$** | **$81.40 \pm 0.40$** | *$81.10 \pm 0.60$* |
| AxTAO | $74.60 \pm 0.90$ | $76.90 \pm 0.90$ | $79.30 \pm 0.80$ | **$81.10 \pm 1.30$** | $82.10 \pm 1.00$ | $82.10 \pm 1.00$ |
| DPDT | *$76.50 \pm 1.20$* | *$78.30 \pm 0.90$* | *$80.30 \pm 0.80$* | **$81.10 \pm 0.30$** | **$82.30 \pm 1.00$** | **$82.30 \pm 1.00$** |
| SPLIT | $72.00 \pm 0.70$ | $77.00 \pm 1.10$ | $78.70 \pm 1.20$ | $80.50 \pm 0.90$ | $81.20 \pm 0.80$ | $81.20 \pm 0.80$ |
| SGT-T | $74.80 \pm 1.00$ | $78.00 \pm 0.80$ | $80.00 \pm 0.70$ | $80.30 \pm 1.10$ | $81.80 \pm 1.30$ | $81.80 \pm 1.30$ |
| SGT$_3$-T | **$78.50 \pm 1.20$** | **$78.70 \pm 0.70$** | **$81.10 \pm 0.60$** | *$81.00 \pm 0.40$* | $81.00 \pm 0.50$ | $81.00 \pm 0.50$ |
| | CART | $83.50 \pm 0.20$ | $86.60 \pm 0.20$ | $87.70 \pm 0.10$ | $89.00 \pm 0.20$ | $89.70 \pm 0.10$ | $89.70 \pm 0.10$ |
| | HSTree | $83.50 \pm 0.20$ | $86.60 \pm 0.20$ | $87.70 \pm 0.10$ | $89.00 \pm 0.20$ | $89.70 \pm 0.10$ | $89.70 \pm 0.10$ |
| | SERDT | $83.30 \pm 0.10$ | $86.40 \pm 0.10$ | $88.10 \pm 0.10$ | $89.00 \pm 0.20$ | *$89.80 \pm 0.10$* | *$89.80 \pm 0.10$* |
| mini-boone | SGT-C | *$84.80 \pm 0.20$* | *$87.10 \pm 0.50$* | $88.30 \pm 0.30$ | $89.20 \pm 0.20$ | $89.70 \pm 0.10$ | $89.70 \pm 0.10$ |
| | SGT$_3$-C | **$85.60 \pm 0.30$** | **$88.10 \pm 0.50$** | **$89.10 \pm 0.20$** | **$89.70 \pm 0.10$** | **$89.90 \pm 0.20$** | **$89.90 \pm 0.20$** |
| | AxTAO | $83.50 \pm 0.20$ | $86.60 \pm 0.10$ | $88.00 \pm 0.20$ | $89.20 \pm 0.30$ | $89.70 \pm 0.30$ | $89.70 \pm 0.30$ |
| | DPDT | *$86.50 \pm 0.20$* | *$88.00 \pm 0.20$* | $88.30 \pm 0.20$ | $89.20 \pm 0.20$ | *$89.90 \pm 0.10$* | *$89.90 \pm 0.10$* |
| | SPLIT | $83.20 \pm 0.20$ | $86.80 \pm 0.20$ | $88.00 \pm 0.30$ | $88.60 \pm 0.30$ | $86.80 \pm 0.20$ | $88.60 \pm 0.30$ |
| | SGT-T | $85.80 \pm 1.60$ | $87.90 \pm 0.20$ | $88.60 \pm 0.10$ | *$89.40 \pm 0.10$* | *$89.90 \pm 0.10$* | *$89.90 \pm 0.10$* |
| | SGT$_3$-T | **$87.10 \pm 0.20$** | **$88.40 \pm 0.10$** | **$89.20 \pm 0.20$** | **$89.70 \pm 0.20$** | **$90.10 \pm 0.20$** | **$90.10 \pm 0.20$** |
| | CART | $95.10 \pm 0.40$ | $98.30 \pm 0.20$ | $99.20 \pm 0.20$ | **$100.00 \pm 0.00$** | $100.00 \pm 0.00$ | $100.00 \pm 0.00$ |
| | HSTree | $95.10 \pm 0.40$ | $98.30 \pm 0.20$ | $99.20 \pm 0.20$ | **$100.00 \pm 0.00$** | $100.00 \pm 0.00$ | $100.00 \pm 0.00$ |
| | SERDT | $95.10 \pm 0.40$ | $98.30 \pm 0.20$ | $99.20 \pm 0.20$ | $99.60 \pm 0.10$ | *$99.90 \pm 0.20$* | *$99.90 \pm 0.20$* |
| mushroom | SGT-C | *$98.60 \pm 0.30$* | *$99.30 \pm 0.10$* | *$99.80 \pm 0.20$* | $99.90 \pm 0.10$ | *$99.90 \pm 0.10$* | *$99.90 \pm 0.10$* |
| | SGT$_3$-C | **$99.40 \pm 0.20$** | **$99.70 \pm 0.20$** | **$100.00 \pm 0.00$** | **$100.00 \pm 0.00$** | **$100.00 \pm 0.00$** | **$100.00 \pm 0.00$** |
| | AxTAO | $95.10 \pm 0.40$ | $98.30 \pm 0.20$ | $99.20 \pm 0.20$ | **$100.00 \pm 0.00$** | $100.00 \pm 0.00$ | $100.00 \pm 0.00$ |
| | DPDT | $96.90 \pm 0.30$ | *$99.00 \pm 0.30$* | **$100.00 \pm 0.00$** | **$100.00 \pm 0.00$** | $100.00 \pm 0.00$ | $100.00 \pm 0.00$ |
| | SPLIT | $88.10 \pm 1.10$ | $96.90 \pm 0.30$ | $98.50 \pm 0.10$ | *$99.70 \pm 0.30$* | *$99.70 \pm 0.30$* | *$99.70 \pm 0.30$* |
| | SGT-T | *$98.60 \pm 0.30$* | **$99.60 \pm 0.20$** | *$99.90 \pm 0.10$* | **$100.00 \pm 0.00$** | $100.00 \pm 0.00$ | $100.00 \pm 0.00$ |
| | SGT$_3$-T | **$99.40 \pm 0.20$** | **$99.60 \pm 0.20$** | **$100.00 \pm 0.00$** | **$100.00 \pm 0.00$** | $100.00 \pm 0.00$ | $100.00 \pm 0.00$ |
| | CART | **$99.00 \pm 0.10$** | **$99.00 \pm 0.10$** | *$99.00 \pm 0.10$* | $99.00 \pm 0.10$ | $99.00 \pm 0.10$ | $99.00 \pm 0.10$ |
| | HSTree | **$99.00 \pm 0.10$** | **$99.00 \pm 0.10$** | *$99.00 \pm 0.10$* | $99.00 \pm 0.10$ | $99.10 \pm 0.10$ | $99.10 \pm 0.10$ |
| | SERDT | **$99.00 \pm 0.10$** | **$99.00 \pm 0.10$** | **$99.10 \pm 0.10$** | *$99.10 \pm 0.10$* | *$99.10 \pm 0.10$* | *$99.10 \pm 0.10$* |
| occupancy | SGT-C | **$99.00 \pm 0.10$** | *$98.90 \pm 0.10$* | *$99.00 \pm 0.10$* | $99.00 \pm 0.20$ | **$99.20 \pm 0.20$** | **$99.20 \pm 0.20$** |
| | SGT$_3$-C | **$99.00 \pm 0.10$** | *$98.90 \pm 0.10$* | *$99.00 \pm 0.10$* | **$99.20 \pm 0.20$** | *$99.10 \pm 0.10$* | $99.00 \pm 0.10$ |
| | AxTAO | **$99.00 \pm 0.10$** | **$99.00 \pm 0.10$** | *$99.00 \pm 0.10$* | $99.00 \pm 0.10$ | $99.10 \pm 0.10$ | $99.10 \pm 0.10$ |
| | DPDT | **$99.00 \pm 0.10$** | **$99.00 \pm 0.10$** | **$99.10 \pm 0.20$** | **$99.20 \pm 0.10$** | *$99.20 \pm 0.20$* | *$99.20 \pm 0.10$* |
| | SPLIT | **$99.00 \pm 0.10$** | **$99.00 \pm 0.10$** | *$99.00 \pm 0.10$* | $99.10 \pm 0.20$ | $99.10 \pm 0.20$ | $99.10 \pm 0.20$ |
| | SGT-T | **$99.00 \pm 0.10$** | **$99.00 \pm 0.20$** | **$99.10 \pm 0.20$** | **$99.20 \pm 0.30$** | **$99.30 \pm 0.20$** | **$99.30 \pm 0.20$** |
| | SGT$_3$-T | *$98.90 \pm 0.20$* | **$99.00 \pm 0.20$** | *$99.00 \pm 0.20$* | $99.10 \pm 0.20$ | $99.10 \pm 0.10$ | $99.10 \pm 0.10$ |
| | CART | — | *$96.00 \pm 0.30$* | $96.50 \pm 0.50$ | **$97.30 \pm 0.60$** | *$97.20 \pm 0.80$* | **$97.30 \pm 0.60$** |
| | HSTree | — | *$96.00 \pm 0.30$* | $96.50 \pm 0.50$ | *$97.20 \pm 0.60$* | **$97.30 \pm 0.50$** | *$97.20 \pm 0.60$* |
| | SERDT | — | **$96.10 \pm 0.20$** | **$96.60 \pm 0.60$** | $96.90 \pm 0.40$ | $96.90 \pm 0.20$ | $96.90 \pm 0.40$ |
| page | SGT-C | — | $95.20 \pm 0.30$ | $96.40 \pm 0.20$ | $96.30 \pm 1.20$ | $96.20 \pm 0.40$ | $96.30 \pm 1.20$ |
| | SGT$_3$-C | — | $95.60 \pm 0.10$ | $95.70 \pm 0.30$ | $96.30 \pm 0.30$ | $96.00 \pm 0.10$ | $96.30 \pm 0.30$ |
| | AxTAO | — | *$96.20 \pm 0.50$* | $96.70 \pm 0.60$ | **$97.20 \pm 0.60$** | *$97.10 \pm 0.60$* | *$97.10 \pm 0.60$* |
| | DPDT | — | **$96.70 \pm 0.40$** | **$97.00 \pm 0.30$** | *$97.20 \pm 0.50$* | **$97.40 \pm 0.60$** | **$97.40 \pm 0.60$** |
| | SPLIT | — | $95.20 \pm 0.20$ | $96.10 \pm 0.20$ | $96.30 \pm 1.00$ | $96.20 \pm 0.60$ | $96.30 \pm 1.00$ |
| | SGT-T | — | $95.50 \pm 0.00$ | $96.30 \pm 0.70$ | *$96.50 \pm 0.50$* | $96.30 \pm 0.90$ | $96.30 \pm 0.90$ |
| | SGT$_3$-T | — | $96.10 \pm 0.10$ | $96.50 \pm 0.70$ | $96.20 \pm 0.40$ | $95.70 \pm 1.10$ | $96.50 \pm 0.70$ |
| | CART | — | — | $77.20 \pm 2.10$ | $85.70 \pm 1.50$ | *$90.00 \pm 0.60$* | *$90.00 \pm 0.60$* |
| | HSTree | — | — | $77.20 \pm 2.10$ | $85.70 \pm 1.50$ | *$90.00 \pm 0.60$* | *$90.00 \pm 0.60$* |
| | SERDT | — | — | $77.60 \pm 3.90$ | $85.30 \pm 3.00$ | $89.30 \pm 1.40$ | $89.30 \pm 1.40$ |
| pendigits | SGT-C | — | — | *$79.30 \pm 3.90$* | *$86.40 \pm 0.70$* | $88.30 \pm 1.20$ | $88.30 \pm 1.20$ |
| | SGT$_3$-C | — | — | **$87.30 \pm 0.50$** | **$90.80 \pm 0.60$** | **$93.50 \pm 0.40$** | **$93.50 \pm 0.40$** |
| | AxTAO | — | — | $78.70 \pm 2.40$ | $86.00 \pm 1.10$ | $87.50 \pm 0.60$ | $87.50 \pm 0.60$ |
| | DPDT | — | — | *$82.70 \pm 0.50$* | *$89.00 \pm 0.50$* | *$93.10 \pm 0.30$* | *$93.10 \pm 0.30$* |
| | SPLIT | — | — | $57.90 \pm 0.50$ | $66.20 \pm 0.60$ | $70.70 \pm 2.20$ | $70.70 \pm 2.20$ |
| | SGT-T | — | — | $81.50 \pm 3.90$ | $87.50 \pm 0.90$ | $91.50 \pm 0.60$ | $91.50 \pm 0.60$ |
| | SGT$_3$-T | — | — | **$87.70 \pm 0.40$** | **$92.50 \pm 0.50$** | **$93.90 \pm 0.50$** | **$93.90 \pm 0.50$** |

| | | | | | | | |
|---|---|---|---|---|---|---|---|
| raisin | CART | **85.90 ± 2.30** | 85.90 ± 2.30 | 85.20 ± 1.20 | **85.90 ± 2.00** | 85.20 ± 1.20 | 85.20 ± 1.20 |
| | HSTree | **85.90 ± 2.30** | *86.30 ± 3.10* | *85.90 ± 3.30* | 85.70 ± 2.20 | **86.10 ± 2.00** | *85.90 ± 3.30* |
| | SERDT | **85.90 ± 2.30** | 86.70 ± 2.00 | **86.30 ± 2.80** | 85.60 ± 2.50 | *85.60 ± 2.50* | **86.30 ± 2.80** |
| | SGT-C | *85.70 ± 1.40* | 85.90 ± 2.30 | 84.30 ± 3.90 | 84.30 ± 2.90 | 84.40 ± 2.90 | 84.30 ± 2.90 |
| | SGT$_3$-C | 84.60 ± 2.30 | 85.20 ± 2.00 | 84.40 ± 3.10 | 84.60 ± 2.30 | 84.60 ± 2.30 | 85.20 ± 2.00 |
| | AxTAO | **86.50 ± 2.70** | **86.50 ± 2.70** | **86.30 ± 2.20** | **86.10 ± 2.20** | **86.30 ± 2.20** | **86.30 ± 2.20** |
| | DPDT | **86.50 ± 2.30** | *86.10 ± 2.50* | *86.10 ± 3.10* | **86.10 ± 3.10** | 85.90 ± 1.70 | *85.90 ± 1.70* |
| | SPLIT | *86.10 ± 2.50* | 85.70 ± 2.80 | 85.70 ± 3.10 | *85.90 ± 2.50* | 85.40 ± 2.20 | 85.70 ± 3.10 |
| | SGT-T | 85.00 ± 3.80 | 85.70 ± 2.20 | 85.40 ± 2.90 | 85.40 ± 2.90 | 85.40 ± 2.90 | 85.40 ± 2.90 |
| | SGT$_3$-T | 85.90 ± 1.30 | 85.60 ± 2.90 | *86.10 ± 3.10* | **86.10 ± 3.10** | *86.10 ± 3.10* | 85.60 ± 2.90 |
| rice | CART | *93.10 ± 0.40* | *93.10 ± 0.40* | 92.70 ± 0.50 | 92.50 ± 0.30 | 92.30 ± 0.30 | 92.50 ± 0.30 |
| | HSTree | *93.10 ± 0.40* | *93.10 ± 0.40* | *93.00 ± 0.60* | *93.00 ± 0.20* | 92.80 ± 0.70 | *93.00 ± 0.60* |
| | SERDT | *93.10 ± 0.40* | *93.10 ± 0.40* | 92.80 ± 0.50 | 92.80 ± 0.60 | 92.50 ± 0.80 | 92.80 ± 0.50 |
| | SGT-C | *93.10 ± 0.40* | 92.70 ± 0.80 | 92.50 ± 0.90 | 92.10 ± 0.90 | 92.50 ± 0.50 | 92.50 ± 0.50 |
| | SGT$_3$-C | **93.20 ± 0.30** | **93.20 ± 0.40** | **93.20 ± 0.50** | **93.20 ± 0.50** | **93.20 ± 0.50** | **93.20 ± 0.30** |
| | AxTAO | *93.10 ± 0.40* | 93.10 ± 0.40 | *92.90 ± 0.50* | 92.90 ± 0.40 | *92.80 ± 0.30* | 92.80 ± 0.30 |
| | DPDT | 93.00 ± 0.20 | *93.20 ± 0.40* | *92.90 ± 0.50* | 92.70 ± 0.40 | 92.70 ± 0.40 | **93.20 ± 0.40** |
| | SPLIT | *93.10 ± 0.40* | **93.30 ± 0.20** | **93.20 ± 0.50** | **93.20 ± 0.50** | **93.20 ± 0.50** | *93.10 ± 0.40* |
| | SGT-T | *93.10 ± 0.40* | 92.30 ± 0.50 | 92.70 ± 0.60 | 92.60 ± 0.60 | 92.30 ± 1.10 | 92.70 ± 0.60 |
| | SGT$_3$-T | **93.20 ± 0.50** | 93.00 ± 0.70 | 92.60 ± 0.20 | 92.60 ± 0.20 | 92.60 ± 0.20 | 92.60 ± 0.20 |
| room | CART | 92.50 ± 0.30 | 97.30 ± 0.50 | 98.70 ± 0.30 | *99.40 ± 0.20* | 99.30 ± 0.20 | 99.40 ± 0.20 |
| | HSTree | 92.50 ± 0.30 | 97.30 ± 0.50 | 98.70 ± 0.30 | *99.40 ± 0.10* | 99.40 ± 0.10 | 99.40 ± 0.10 |
| | SERDT | 92.50 ± 0.30 | 96.20 ± 0.60 | 98.90 ± 0.10 | 99.30 ± 0.20 | *99.50 ± 0.20* | 99.50 ± 0.20 |
| | SGT-C | *93.00 ± 0.40* | 98.30 ± 0.20 | *99.40 ± 0.10* | 99.40 ± 0.00 | **99.60 ± 0.00** | **99.60 ± 0.00** |
| | SGT$_3$-C | **98.70 ± 0.10** | **99.50 ± 0.20** | **99.50 ± 0.20** | **99.50 ± 0.20** | *99.50 ± 0.20* | 99.50 ± 0.20 |
| | AxTAO | 92.50 ± 0.30 | 98.80 ± 0.30 | 99.20 ± 0.00 | *99.50 ± 0.20* | 99.40 ± 0.20 | 99.40 ± 0.20 |
| | DPDT | 93.60 ± 1.10 | 99.00 ± 0.00 | 99.30 ± 0.20 | **99.60 ± 0.10** | *99.40 ± 0.10* | **99.60 ± 0.10** |
| | SPLIT | 88.80 ± 0.30 | 90.90 ± 0.30 | 92.40 ± 0.70 | 94.00 ± 0.70 | 94.20 ± 0.90 | 94.20 ± 0.90 |
| | SGT-T | *96.90 ± 0.20* | **99.50 ± 0.10** | **99.60 ± 0.00** | *99.50 ± 0.20* | **99.50 ± 0.10** | 99.50 ± 0.10 |
| | SGT$_3$-T | **98.60 ± 0.30** | 99.30 ± 0.10 | 99.50 ± 0.20 | 99.20 ± 0.30 | **99.50 ± 0.10** | 99.50 ± 0.20 |
| segment | CART | — | 81.30 ± 0.10 | 88.70 ± 1.10 | 90.90 ± 0.80 | **93.40 ± 1.00** | **93.40 ± 1.00** |
| | HSTree | — | 81.30 ± 0.10 | 88.70 ± 1.10 | *91.30 ± 1.10* | **93.40 ± 1.20** | **93.40 ± 1.20** |
| | SERDT | — | 67.00 ± 3.20 | 81.50 ± 0.20 | 88.70 ± 1.20 | 91.00 ± 1.80 | 91.00 ± 1.80 |
| | SGT-C | — | *83.30 ± 3.60* | 88.90 ± 0.80 | 89.80 ± 2.50 | 91.50 ± 1.90 | 91.50 ± 1.90 |
| | SGT$_3$-C | — | **88.70 ± 0.20** | **91.20 ± 0.50** | **92.40 ± 1.00** | *92.90 ± 1.10* | 92.40 ± 1.00 |
| | AxTAO | — | 81.70 ± 0.70 | 89.40 ± 1.00 | *92.40 ± 0.90* | 93.30 ± 1.00 | *93.30 ± 1.00* |
| | DPDT | — | 87.10 ± 0.50 | 90.50 ± 0.60 | 92.10 ± 0.30 | **94.20 ± 0.70** | **94.20 ± 0.70** |
| | SPLIT | — | 52.30 ± 2.50 | 61.90 ± 3.80 | 67.20 ± 3.50 | 67.70 ± 2.60 | 67.70 ± 2.60 |
| | SGT-T | — | 84.20 ± 3.40 | 89.80 ± 0.60 | 91.60 ± 1.50 | 90.80 ± 1.70 | 90.80 ± 1.70 |
| | SGT$_3$-T | — | **89.30 ± 1.20** | **91.70 ± 0.10** | **93.00 ± 0.90** | 92.90 ± 0.90 | 93.00 ± 0.90 |
| skin | CART | 90.80 ± 0.10 | 96.60 ± 0.10 | 97.40 ± 0.10 | 98.40 ± 0.10 | 98.90 ± 0.10 | 98.90 ± 0.10 |
| | HSTree | 90.80 ± 0.10 | 96.60 ± 0.10 | 97.50 ± 0.10 | 98.50 ± 0.00 | 99.00 ± 0.00 | 99.00 ± 0.00 |
| | SERDT | 90.80 ± 0.10 | 96.60 ± 0.10 | 97.80 ± 0.10 | 98.60 ± 0.10 | *99.30 ± 0.00* | 99.30 ± 0.00 |
| | SGT-C | 93.20 ± 0.20 | 97.70 ± 0.10 | 98.50 ± 0.00 | 98.90 ± 0.10 | 99.20 ± 0.20 | 99.20 ± 0.20 |
| | SGT$_3$-C | **96.80 ± 0.10** | **98.80 ± 0.00** | **99.20 ± 0.10** | **99.60 ± 0.10** | **99.80 ± 0.00** | **99.80 ± 0.00** |
| | AxTAO | 90.80 ± 0.10 | 96.60 ± 0.10 | 97.90 ± 0.00 | 98.70 ± 0.20 | *99.60 ± 0.00* | 99.60 ± 0.00 |
| | DPDT | 92.50 ± 0.10 | 96.60 ± 0.10 | 98.50 ± 0.00 | *99.10 ± 0.00* | 99.40 ± 0.10 | 99.40 ± 0.10 |
| | SPLIT | 88.40 ± 0.10 | 92.70 ± 0.00 | 96.50 ± 0.10 | 98.10 ± 0.10 | 98.50 ± 0.00 | 98.50 ± 0.00 |
| | SGT-T | 93.20 ± 0.10 | 98.20 ± 0.00 | 99.20 ± 0.10 | **99.70 ± 0.10** | **99.80 ± 0.10** | **99.80 ± 0.10** |
| | SGT$_3$-T | **97.10 ± 0.10** | **98.80 ± 0.10** | **99.70 ± 0.10** | **99.70 ± 0.10** | **99.80 ± 0.00** | **99.80 ± 0.00** |
| wilt | CART | *97.30 ± 0.40* | **97.50 ± 0.20** | *97.80 ± 0.60* | *98.20 ± 0.50* | 98.10 ± 0.70 | 98.20 ± 0.50 |
| | HSTree | *97.30 ± 0.40* | **97.50 ± 0.20** | 97.70 ± 0.70 | *98.20 ± 0.50* | **98.40 ± 0.50** | **98.40 ± 0.50** |
| | SERDT | *97.30 ± 0.40* | **97.50 ± 0.20** | **97.90 ± 0.60** | **98.30 ± 0.40** | *98.30 ± 0.60* | 98.30 ± 0.40 |
| | SGT-C | 97.10 ± 0.60 | *97.10 ± 0.10* | **97.90 ± 0.50** | 97.60 ± 0.20 | 97.60 ± 0.20 | 97.90 ± 0.50 |
| | SGT$_3$-C | **97.50 ± 0.50** | **97.50 ± 0.20** | *97.80 ± 0.50* | 97.60 ± 0.30 | 97.60 ± 0.30 | 97.50 ± 0.20 |
| | AxTAO | **97.30 ± 0.40** | 97.50 ± 0.20 | 97.80 ± 0.70 | *98.10 ± 0.70* | 98.00 ± 0.10 | *98.10 ± 0.70* |
| | DPDT | 97.20 ± 0.40 | **97.80 ± 0.30** | 98.20 ± 0.40 | **98.20 ± 0.50** | **98.30 ± 0.60** | **98.30 ± 0.60** |
| | SPLIT | 95.00 ± 0.40 | 97.30 ± 0.40 | 97.70 ± 0.20 | 97.80 ± 0.10 | 97.80 ± 0.10 | 97.80 ± 0.10 |
| | SGT-T | **97.30 ± 0.40** | 96.90 ± 0.40 | 97.40 ± 0.30 | 97.60 ± 0.50 | 97.60 ± 0.40 | 97.60 ± 0.40 |

SGT$_3$-T **97.30 ± 0.60** *97.50 ± 0.30* 97.70 ± 0.30  97.50 ± 0.20  97.50 ± 0.20  97.50 ± 0.20

Table 5: Test Accuracy (% ± std.) for all bivariate approaches separated into TDIDT and Non-Greedy approaches by dashed lines per dataset. Best approach is **bolded**, second-best is *italicized*.

| dataset | model | 2 | 3 | 4 | 5 | 6 | Best |
|---|---|---|---|---|---|---|---|
| adult | BiCART | 83.40 ± 0.30 | 83.60 ± 0.30 | 83.80 ± 0.40 | 84.10 ± 0.50 | *84.10 ± 0.50* | *84.10 ± 0.50* |
| | S$^2$GT-C | *83.60 ± 0.30* | *83.80 ± 0.40* | 84.40 ± 0.50 | 84.90 ± 0.60 | **85.10 ± 0.60** | **85.10 ± 0.60** |
| | S$^2$GT$_3$-C | **84.30 ± 0.20** | **84.10 ± 0.50** | **84.60 ± 0.70** | **85.00 ± 0.40** | **85.10 ± 0.20** | **85.10 ± 0.20** |
| | BiTAO | 83.60 ± 0.40 | 84.00 ± 0.30 | 84.20 ± 0.40 | 84.30 ± 0.50 | 84.30 ± 0.50 | 84.30 ± 0.50 |
| | S$^2$GT-T | *83.90 ± 0.60* | *84.30 ± 0.40* | *84.90 ± 0.40* | *85.00 ± 0.70* | *84.90 ± 0.30* | *85.00 ± 0.70* |
| | S$^2$GT$_3$-T | **84.10 ± 0.40** | **84.50 ± 0.50** | **85.00 ± 0.40** | **85.10 ± 0.60** | **85.00 ± 0.40** | **85.10 ± 0.60** |
| avila | BiCART | — | — | 59.80 ± 0.50 | 62.20 ± 0.60 | 66.90 ± 0.50 | 66.90 ± 0.50 |
| | S$^2$GT-C | — | — | *80.20 ± 2.80* | *88.70 ± 4.30* | *95.30 ± 0.90* | *95.30 ± 0.90* |
| | S$^2$GT$_3$-C | — | — | **97.30 ± 0.90** | **99.90 ± 0.10** | **99.90 ± 0.10** | **99.90 ± 0.10** |
| | BiTAO | — | — | 60.00 ± 0.40 | 66.00 ± 1.20 | 70.90 ± 1.50 | 70.90 ± 1.50 |
| | S$^2$GT-T | — | — | *83.00 ± 0.70* | *90.90 ± 1.80* | *95.10 ± 1.10* | *95.10 ± 1.10* |
| | S$^2$GT$_3$-T | — | — | **97.10 ± 0.20** | **99.90 ± 0.10** | **99.80 ± 0.00** | **99.80 ± 0.00** |
| bank | BiCART | 95.60 ± 1.90 | *97.80 ± 2.00* | **99.40 ± 0.80** | 99.50 ± 0.80 | 99.30 ± 0.70 | **99.40 ± 0.80** |
| | S$^2$GT-C | *98.40 ± 0.80* | **98.40 ± 0.80** | 98.40 ± 0.80 | 98.40 ± 0.80 | 98.40 ± 0.80 | 98.40 ± 0.80 |
| | S$^2$GT$_3$-C | **98.50 ± 0.60** | 98.40 ± 0.60 | 98.40 ± 0.60 | 98.40 ± 0.60 | 98.40 ± 0.60 | 98.50 ± 0.60 |
| | BiTAO | 95.40 ± 1.80 | 98.30 ± 0.80 | **99.50 ± 0.60** | **99.50 ± 0.60** | **99.50 ± 0.60** | **99.50 ± 0.60** |
| | S$^2$GT-T | *97.80 ± 1.00* | **98.90 ± 0.40** | 98.90 ± 0.40 | 98.90 ± 0.40 | 98.90 ± 0.40 | 98.90 ± 0.40 |
| | S$^2$GT$_3$-T | **98.90 ± 1.00** | 98.50 ± 0.70 | 98.50 ± 0.70 | 98.50 ± 0.70 | 98.50 ± 0.70 | 98.90 ± 1.00 |
| bean | BiCART | — | 89.80 ± 0.70 | 90.40 ± 0.30 | **91.50 ± 0.00** | 91.80 ± 0.20 | **91.80 ± 0.20** |
| | S$^2$GT-C | — | 90.10 ± 0.50 | 90.60 ± 0.40 | 91.30 ± 0.30 | 91.30 ± 0.50 | 91.30 ± 0.50 |
| | S$^2$GT$_3$-C | — | **90.30 ± 0.90** | **91.10 ± 0.30** | 91.30 ± 0.50 | 90.70 ± 0.30 | 91.10 ± 0.30 |
| | BiTAO | — | 90.20 ± 0.30 | **91.00 ± 0.40** | *91.50 ± 0.30* | *91.70 ± 0.40* | *91.70 ± 0.40* |
| | S$^2$GT-T | — | **91.00 ± 0.30** | *90.90 ± 0.30* | **91.60 ± 0.40** | **91.80 ± 0.50** | **91.80 ± 0.50** |
| | S$^2$GT$_3$-T | — | 90.60 ± 0.30 | 90.70 ± 0.60 | 91.20 ± 0.70 | 91.40 ± 0.40 | 91.20 ± 0.70 |
| bidding | BiCART | 97.90 ± 0.40 | 99.60 ± 0.20 | **99.80 ± 0.10** | **99.80 ± 0.10** | **99.80 ± 0.10** | **99.80 ± 0.10** |
| | S$^2$GT-C | *99.50 ± 0.60* | **99.70 ± 0.20** | *99.70 ± 0.20* | *99.70 ± 0.20* | *99.70 ± 0.20* | *99.70 ± 0.20* |
| | S$^2$GT$_3$-C | **99.70 ± 0.30** | 99.60 ± 0.10 | 99.60 ± 0.10 | 99.60 ± 0.10 | 99.60 ± 0.10 | *99.70 ± 0.30* |
| | BiTAO | 98.00 ± 0.50 | 99.50 ± 0.20 | **99.80 ± 0.10** | **99.80 ± 0.10** | **99.80 ± 0.10** | **99.80 ± 0.10** |
| | S$^2$GT-T | **99.50 ± 0.10** | **99.80 ± 0.10** | **99.80 ± 0.10** | **99.80 ± 0.10** | **99.80 ± 0.10** | **99.80 ± 0.10** |
| | S$^2$GT$_3$-T | **99.50 ± 0.10** | 99.50 ± 0.20 | 99.50 ± 0.20 | 99.50 ± 0.20 | 99.50 ± 0.20 | 99.50 ± 0.20 |
| electricity | BiCART | 75.80 ± 0.20 | 76.20 ± 0.10 | 78.30 ± 0.70 | 78.70 ± 0.30 | 80.40 ± 0.30 | 80.40 ± 0.30 |
| | S$^2$GT-C | *80.60 ± 0.60* | *82.00 ± 0.60* | *83.90 ± 0.30* | 86.50 ± 1.20 | 87.50 ± 0.50 | 87.50 ± 0.50 |
| | S$^2$GT$_3$-C | **81.80 ± 0.20** | **84.50 ± 0.30** | **86.60 ± 0.80** | **87.30 ± 1.00** | **88.80 ± 0.20** | **88.80 ± 0.20** |
| | BiTAO | 75.80 ± 0.20 | 76.20 ± 0.10 | 79.50 ± 0.50 | 80.10 ± 0.70 | 80.70 ± 0.40 | 80.70 ± 0.40 |
| | S$^2$GT-T | *81.20 ± 0.50* | *83.60 ± 0.50* | *85.80 ± 0.80* | *86.80 ± 0.70* | 88.30 ± 0.80 | 88.30 ± 0.80 |
| | S$^2$GT$_3$-T | **82.10 ± 0.30** | **84.90 ± 0.60** | **86.30 ± 0.30** | **87.70 ± 0.20** | **88.50 ± 0.50** | **88.50 ± 0.50** |
| eye-movements | BiCART | 45.40 ± 1.20 | 51.60 ± 0.70 | 52.20 ± 1.10 | 54.70 ± 1.60 | 56.30 ± 0.50 | 56.30 ± 0.50 |
| | S$^2$GT-C | *57.80 ± 1.40* | *60.00 ± 2.00* | *61.70 ± 3.20* | *62.60 ± 3.30* | **64.80 ± 3.50** | **64.80 ± 3.50** |
| | S$^2$GT$_3$-C | **59.60 ± 2.40** | **63.60 ± 1.40** | **66.20 ± 1.90** | **67.50 ± 3.00** | 63.90 ± 2.40 | 63.90 ± 2.40 |
| | BiTAO | 48.10 ± 2.00 | 51.60 ± 0.10 | 53.20 ± 0.40 | 55.80 ± 0.70 | 56.70 ± 0.40 | 56.70 ± 0.40 |
| | S$^2$GT-T | *58.70 ± 0.60* | *60.10 ± 1.70* | **63.90 ± 1.10** | **66.10 ± 1.20** | **66.60 ± 1.40** | **66.60 ± 1.40** |
| | S$^2$GT$_3$-T | **59.80 ± 1.70** | **62.80 ± 1.10** | *56.90 ± 1.10* | *63.20 ± 1.20* | *64.90 ± 1.30* | *62.80 ± 1.10* |
| eye-state | BiCART | 66.20 ± 2.40 | 69.20 ± 1.10 | 72.90 ± 0.60 | 74.10 ± 2.70 | 77.60 ± 0.80 | 77.60 ± 0.80 |
| | S$^2$GT-C | *71.70 ± 1.10* | *72.80 ± 0.70* | *74.40 ± 2.10* | *76.10 ± 2.00* | 80.20 ± 0.30 | 80.20 ± 0.30 |
| | S$^2$GT$_3$-C | **73.10 ± 1.10** | **75.90 ± 0.40** | **79.30 ± 1.10** | **81.10 ± 0.60** | 80.60 ± 1.90 | **81.10 ± 0.60** |
| | BiTAO | 68.80 ± 1.20 | 70.10 ± 0.70 | 74.50 ± 1.50 | 77.50 ± 0.40 | 79.00 ± 1.40 | 79.00 ± 1.40 |
| | S$^2$GT-T | *72.40 ± 1.60* | *75.90 ± 1.20* | 77.20 ± 0.80 | 79.00 ± 1.70 | 80.90 ± 1.00 | 80.90 ± 1.00 |

| | | | | | | | |
|---|---|---|---|---|---|---|---|
| | S$^2$GT$_3$-T | **74.50 ± 1.20** | **76.30 ± 0.20** | **79.30 ± 1.80** | **80.50 ± 1.70** | **81.70 ± 1.20** | **81.70 ± 1.20** |
| fault | BiCART | — | 61.40 ± 3.00 | 67.40 ± 1.90 | 68.80 ± 1.10 | **73.40 ± 1.50** | **73.40 ± 1.50** |
| | S$^2$GT-C | — | *66.60 ± 5.80* | *69.00 ± 5.80* | *69.20 ± 1.00* | *71.90 ± 2.40* | *71.90 ± 2.40* |
| | S$^2$GT$_3$-C | — | **68.50 ± 2.50** | **70.30 ± 1.70** | **71.60 ± 2.80** | 70.00 ± 1.90 | 71.60 ± 2.80 |
| | BiTAO | — | 62.30 ± 3.30 | 67.90 ± 3.00 | 68.00 ± 1.70 | **73.40 ± 1.20** | **73.40 ± 1.20** |
| | S$^2$GT-T | — | *67.90 ± 1.20* | *70.40 ± 3.60* | *71.80 ± 1.50* | *72.40 ± 2.00* | *71.80 ± 1.50* |
| | S$^2$GT$_3$-T | — | **70.40 ± 3.10** | **73.40 ± 1.90** | **73.20 ± 3.40** | 72.30 ± 1.50 | 73.20 ± 3.40 |
| gas-drift | BiCART | — | *86.70 ± 1.10* | *95.40 ± 0.40* | **97.90 ± 0.10** | **98.60 ± 0.20** | **98.60 ± 0.20** |
| | S$^2$GT-C | — | 83.60 ± 4.00 | 94.90 ± 1.20 | 96.80 ± 0.20 | *97.20 ± 0.30* | *97.20 ± 0.30* |
| | S$^2$GT$_3$-C | — | **96.20 ± 0.50** | **97.00 ± 0.40** | *97.10 ± 0.30* | 97.10 ± 0.40 | 97.10 ± 0.40 |
| | BiTAO | — | 91.50 ± 0.10 | 96.20 ± 0.70 | 97.20 ± 0.30 | **98.20 ± 0.00** | **98.20 ± 0.00** |
| | S$^2$GT-T | — | *92.00 ± 1.10* | 94.80 ± 1.20 | *97.30 ± 0.60* | *98.00 ± 0.10* | *98.00 ± 0.10* |
| | S$^2$GT$_3$-T | — | **96.40 ± 0.50** | **97.90 ± 0.20** | *97.90 ± 0.20* | 97.90 ± 0.20 | 97.90 ± 0.20 |
| htru | BiCART | **97.90 ± 0.10** | 98.00 ± 0.10 | 98.10 ± 0.20 | 98.00 ± 0.20 | 98.00 ± 0.10 | 98.00 ± 0.10 |
| | S$^2$GT-C | *97.80 ± 0.00* | *97.80 ± 0.10* | *97.80 ± 0.20* | *97.80 ± 0.30* | *97.80 ± 0.30* | *97.80 ± 0.20* |
| | S$^2$GT$_3$-C | 97.80 ± 0.00 | **98.00 ± 0.00** | 98.00 ± 0.10 | **98.00 ± 0.10** | **98.00 ± 0.10** | **98.00 ± 0.00** |
| | BiTAO | **97.90 ± 0.10** | **98.10 ± 0.10** | *98.00 ± 0.30* | 98.00 ± 0.10 | 98.00 ± 0.00 | 98.00 ± 0.00 |
| | S$^2$GT-T | *97.90 ± 0.20* | *98.00 ± 0.20* | 97.80 ± 0.20 | **98.10 ± 0.10** | *98.00 ± 0.10* | *98.00 ± 0.20* |
| | S$^2$GT$_3$-T | 97.60 ± 0.10 | 98.00 ± 0.10 | **98.10 ± 0.10** | **98.10 ± 0.00** | **98.10 ± 0.00** | **98.10 ± 0.00** |
| magic | BiCART | 77.50 ± 1.40 | 81.10 ± 0.30 | 81.10 ± 0.30 | 82.90 ± 0.20 | 83.00 ± 0.60 | 82.90 ± 0.20 |
| | S$^2$GT-C | *81.10 ± 1.00* | *81.70 ± 0.90* | *81.20 ± 0.80* | *81.90 ± 0.40* | *82.10 ± 0.10* | *81.90 ± 0.40* |
| | S$^2$GT$_3$-C | **81.50 ± 0.80** | **82.90 ± 0.60** | **82.90 ± 0.40** | **83.50 ± 0.50** | 82.70 ± 0.70 | **82.90 ± 0.40** |
| | BiTAO | 77.80 ± 1.30 | 81.70 ± 0.50 | 82.00 ± 0.40 | 82.70 ± 0.30 | **83.30 ± 0.70** | **83.30 ± 0.70** |
| | S$^2$GT-T | *81.40 ± 0.10* | *82.10 ± 0.20* | *82.60 ± 0.70* | 82.50 ± 0.40 | *82.80 ± 0.60* | 82.50 ± 0.40 |
| | S$^2$GT$_3$-T | **82.20 ± 0.70** | **83.00 ± 0.80** | **83.60 ± 1.00** | **82.80 ± 0.20** | 82.50 ± 0.60 | 82.80 ± 0.20 |
| mini-boone | BiCART | 87.20 ± 0.10 | 88.60 ± 0.20 | 89.50 ± 0.30 | 90.30 ± 0.10 | 90.70 ± 0.20 | 90.70 ± 0.20 |
| | S$^2$GT-C | *87.60 ± 0.30* | *89.50 ± 0.10* | *90.10 ± 0.20* | *90.50 ± 0.20* | 90.60 ± 0.20 | 90.60 ± 0.20 |
| | S$^2$GT$_3$-C | **88.80 ± 0.30** | **90.00 ± 0.20** | *90.50 ± 0.20* | **90.90 ± 0.10** | **90.90 ± 0.20** | **90.90 ± 0.20** |
| | BiTAO | 87.30 ± 0.10 | 88.40 ± 0.30 | 89.70 ± 0.40 | 90.20 ± 0.10 | 90.50 ± 0.20 | 90.50 ± 0.20 |
| | S$^2$GT-T | *88.40 ± 0.50* | *89.70 ± 0.20* | *90.30 ± 0.20* | *90.80 ± 0.10* | *91.10 ± 0.10* | *91.10 ± 0.10* |
| | S$^2$GT$_3$-T | **89.00 ± 0.30** | **90.10 ± 0.20** | **90.60 ± 0.30** | **91.20 ± 0.30** | **91.20 ± 0.10** | **91.20 ± 0.10** |
| mushroom | BiCART | 98.70 ± 0.10 | **100.00 ± 0.00** | **100.00 ± 0.00** | **100.00 ± 0.00** | **100.00 ± 0.00** | **100.00 ± 0.00** |
| | S$^2$GT-C | *99.40 ± 0.20* | **100.00 ± 0.00** | **100.00 ± 0.00** | **100.00 ± 0.00** | **100.00 ± 0.00** | **100.00 ± 0.00** |
| | S$^2$GT$_3$-C | **100.00 ± 0.00** | **100.00 ± 0.00** | **100.00 ± 0.00** | **100.00 ± 0.00** | **100.00 ± 0.00** | **100.00 ± 0.00** |
| | BiTAO | 98.70 ± 0.10 | **100.00 ± 0.00** | **100.00 ± 0.00** | **100.00 ± 0.00** | **100.00 ± 0.00** | **100.00 ± 0.00** |
| | S$^2$GT-T | *99.40 ± 0.10* | **100.00 ± 0.00** | **100.00 ± 0.00** | **100.00 ± 0.00** | **100.00 ± 0.00** | **100.00 ± 0.00** |
| | S$^2$GT$_3$-T | **100.00 ± 0.00** | **100.00 ± 0.00** | **100.00 ± 0.00** | **100.00 ± 0.00** | **100.00 ± 0.00** | **100.00 ± 0.00** |
| occupancy | BiCART | 99.00 ± 0.10 | 99.00 ± 0.10 | 99.10 ± 0.10 | 99.20 ± 0.10 | 99.20 ± 0.10 | **99.20 ± 0.10** |
| | S$^2$GT-C | *99.10 ± 0.20* | *99.10 ± 0.10* | *99.20 ± 0.10* | **99.30 ± 0.10** | **99.30 ± 0.10** | *99.10 ± 0.10* |
| | S$^2$GT$_3$-C | **99.20 ± 0.20** | **99.30 ± 0.20** | **99.30 ± 0.10** | **99.30 ± 0.20** | **99.30 ± 0.20** | **99.20 ± 0.20** |
| | BiTAO | 99.00 ± 0.10 | 99.00 ± 0.10 | *99.10 ± 0.10* | *99.10 ± 0.10* | *99.10 ± 0.20* | *99.10 ± 0.20* |
| | S$^2$GT-T | *99.10 ± 0.20* | *99.10 ± 0.20* | **99.30 ± 0.20** | *99.10 ± 0.20* | **99.20 ± 0.20** | **99.20 ± 0.20** |
| | S$^2$GT$_3$-T | **99.20 ± 0.20** | **99.20 ± 0.20** | *99.10 ± 0.20* | **99.20 ± 0.20** | **99.20 ± 0.20** | **99.20 ± 0.20** |
| page | BiCART | — | *96.40 ± 0.30* | **97.20 ± 0.20** | 96.80 ± 0.50 | *97.10 ± 0.60* | *97.10 ± 0.60* |
| | S$^2$GT-C | — | **96.90 ± 0.10** | **97.20 ± 0.70** | *96.90 ± 1.20* | 96.10 ± 0.90 | **97.20 ± 0.70** |
| | S$^2$GT$_3$-C | — | 96.20 ± 0.40 | 96.80 ± 0.60 | 96.80 ± 0.30 | 96.80 ± 0.40 | 96.80 ± 0.60 |
| | BiTAO | — | *96.60 ± 0.60* | **97.40 ± 0.20** | **97.40 ± 0.40** | 96.80 ± 0.80 | 96.80 ± 0.80 |
| | S$^2$GT-T | — | **97.00 ± 0.20** | *96.90 ± 0.40* | *97.10 ± 0.80* | 96.60 ± 0.10 | **97.00 ± 0.20** |
| | S$^2$GT$_3$-T | — | 96.50 ± 0.50 | 96.40 ± 0.60 | 96.40 ± 0.60 | 96.40 ± 0.60 | 96.40 ± 0.60 |
| pendigits | BiCART | — | — | *87.40 ± 0.30* | 91.90 ± 1.30 | **94.50 ± 1.10** | **94.50 ± 1.10** |
| | S$^2$GT-C | — | — | 87.20 ± 1.10 | 91.00 ± 0.80 | *93.30 ± 0.10* | *93.30 ± 0.10* |
| | S$^2$GT$_3$-C | — | — | **91.80 ± 1.10** | **95.80 ± 0.70** | 94.20 ± 0.60 | **95.80 ± 0.70** |
| | BiTAO | — | — | 88.60 ± 0.50 | 93.80 ± 0.80 | **95.30 ± 0.90** | **95.30 ± 0.90** |
| | S$^2$GT-T | — | — | *89.00 ± 0.90* | 92.10 ± 1.00 | *94.40 ± 0.20* | *94.40 ± 0.20* |

| dataset | model | | | | | | |
|---|---|---|---|---|---|---|---|
| | $S^2GT_3$-T | — | — | **94.10 ± 0.40** | **94.10 ± 0.30** | *94.80 ± 0.20* | *94.80 ± 0.20* |
| raisin | BiCART | **89.40 ± 3.10** | **85.70 ± 1.70** | **86.50 ± 1.30** | 86.50 ± 3.70 | *85.40 ± 1.20* | **85.70 ± 1.70** |
| | $S^2GT$-C | 85.40 ± 2.20 | **85.70 ± 1.70** | *85.00 ± 3.50* | 85.00 ± 1.90 | **86.50 ± 1.30** | **85.70 ± 1.70** |
| | $S^2GT_3$-C | *85.70 ± 1.70* | 83.00 ± 1.20 | 83.00 ± 1.20 | 83.00 ± 1.20 | 83.00 ± 1.20 | **85.70 ± 1.70** |
| | BiTAO | **89.30 ± 3.40** | **88.50 ± 3.10** | **89.40 ± 3.50** | **89.40 ± 3.50** | **89.30 ± 3.40** | **88.50 ± 3.10** |
| | $S^2GT$-T | 85.70 ± 3.70 | *86.70 ± 2.40* | 85.00 ± 3.10 | 85.00 ± 3.10 | 85.00 ± 3.10 | 85.70 ± 3.70 |
| | $S^2GT_3$-T | *85.90 ± 3.70* | *85.90 ± 3.70* | *85.90 ± 3.70* | *85.90 ± 3.70* | *85.90 ± 3.70* | *85.90 ± 3.70* |
| rice | BiCART | **93.10 ± 0.50** | **93.10 ± 0.50** | 92.70 ± 0.40 | 92.80 ± 0.60 | **92.80 ± 0.50** | 92.80 ± 0.60 |
| | $S^2GT$-C | 92.90 ± 0.90 | *93.00 ± 0.60* | 92.00 ± 0.60 | 92.50 ± 0.60 | 91.90 ± 0.60 | *92.90 ± 0.90* |
| | $S^2GT_3$-C | *93.00 ± 0.50* | 92.70 ± 0.40 | 92.50 ± 0.60 | 92.50 ± 0.60 | 92.50 ± 0.60 | **93.00 ± 0.50** |
| | BiTAO | **93.10 ± 0.50** | **92.90 ± 0.50** | *92.90 ± 0.60* | **93.00 ± 0.50** | **93.00 ± 0.50** | **93.00 ± 0.50** |
| | $S^2GT$-T | *92.80 ± 0.20* | 92.30 ± 1.40 | **93.10 ± 0.60** | 92.90 ± 0.90 | 92.70 ± 0.20 | 92.30 ± 1.40 |
| | $S^2GT_3$-T | 92.60 ± 1.40 | *92.50 ± 0.10* | 92.50 ± 0.10 | 92.50 ± 0.10 | 92.50 ± 0.10 | *92.50 ± 0.10* |
| room | BiCART | 93.30 ± 0.30 | 98.80 ± 0.20 | **99.50 ± 0.20** | **99.50 ± 0.20** | *99.40 ± 0.10* | **99.50 ± 0.20** |
| | $S^2GT$-C | *94.60 ± 0.20* | *99.40 ± 0.10* | *99.40 ± 0.20* | *99.40 ± 0.20* | *99.40 ± 0.20* | *99.40 ± 0.20* |
| | $S^2GT_3$-C | **99.50 ± 0.00** | *99.50 ± 0.20* | *99.50 ± 0.20* | *99.50 ± 0.20* | *99.50 ± 0.20* | **99.50 ± 0.20** |
| | BiTAO | 96.60 ± 0.00 | *99.50 ± 0.20* | *99.30 ± 0.10* | **99.60 ± 0.20** | *99.50 ± 0.20* | *99.60 ± 0.20* |
| | $S^2GT$-T | *98.20 ± 0.50* | *99.50 ± 0.20* | **99.50 ± 0.00** | **99.60 ± 0.20** | *99.50 ± 0.20* | *99.50 ± 0.00* |
| | $S^2GT_3$-T | **99.70 ± 0.10** | **99.60 ± 0.10** | *99.50 ± 0.30* | *99.50 ± 0.10* | **99.50 ± 0.10** | **99.70 ± 0.10** |
| segment | BiCART | — | 84.50 ± 6.60 | *93.10 ± 1.20* | 92.90 ± 2.00 | *94.00 ± 1.10* | **94.00 ± 1.10** |
| | $S^2GT$-C | — | **92.90 ± 1.10** | **93.60 ± 0.50** | *93.50 ± 1.30* | 93.90 ± 1.40 | *93.90 ± 1.40* |
| | $S^2GT_3$-C | — | *92.60 ± 1.80* | *93.10 ± 1.20* | **94.50 ± 1.90** | **94.50 ± 1.90** | 93.10 ± 1.20 |
| | BiTAO | — | 85.20 ± 6.80 | 92.70 ± 0.90 | 93.70 ± 2.30 | *94.70 ± 0.50* | *94.70 ± 0.50* |
| | $S^2GT$-T | — | *93.40 ± 2.00* | *93.40 ± 1.50* | *94.30 ± 1.20* | 94.40 ± 1.50 | 94.30 ± 1.20 |
| | $S^2GT_3$-T | — | **94.40 ± 1.10** | **94.90 ± 0.50** | **94.80 ± 1.30** | **94.90 ± 1.20** | **94.80 ± 1.30** |
| skin | BiCART | 98.60 ± 0.10 | *99.40 ± 0.00* | *99.70 ± 0.00* | *99.80 ± 0.00* | *99.80 ± 0.00* | *99.80 ± 0.00* |
| | $S^2GT$-C | *99.80 ± 0.00* | **99.90 ± 0.00** | **99.90 ± 0.00** | **99.90 ± 0.00** | **99.90 ± 0.00** | **99.90 ± 0.00** |
| | $S^2GT_3$-C | **99.90 ± 0.00** | **99.90 ± 0.00** | **99.90 ± 0.00** | **99.90 ± 0.00** | **99.90 ± 0.00** | **99.90 ± 0.00** |
| | BiTAO | 98.60 ± 0.10 | *99.40 ± 0.00* | *99.80 ± 0.00* | *99.80 ± 0.00* | **99.90 ± 0.00** | **99.90 ± 0.00** |
| | $S^2GT$-T | *99.80 ± 0.00* | **99.90 ± 0.00** | **99.90 ± 0.00** | **99.90 ± 0.00** | **99.90 ± 0.00** | **99.90 ± 0.00** |
| | $S^2GT_3$-T | **99.90 ± 0.00** | **99.90 ± 0.00** | **99.90 ± 0.00** | **99.90 ± 0.00** | **99.90 ± 0.00** | **99.90 ± 0.00** |
| wilt | BiCART | **98.50 ± 0.20** | **98.50 ± 0.20** | **98.50 ± 0.20** | 98.50 ± 0.20 | **98.90 ± 0.50** | **98.90 ± 0.50** |
| | $S^2GT$-C | **98.80 ± 0.20** | **98.50 ± 0.10** | *98.20 ± 0.60* | **98.60 ± 0.40** | *98.60 ± 0.40* | *98.80 ± 0.20* |
| | $S^2GT_3$-C | 98.30 ± 0.40 | *98.30 ± 0.20* | **98.50 ± 0.40** | 98.40 ± 0.30 | 98.40 ± 0.30 | 98.30 ± 0.20 |
| | BiTAO | **98.50 ± 0.20** | 98.40 ± 0.10 | **98.50 ± 0.10** | 98.50 ± 0.20 | 98.50 ± 0.20 | 98.50 ± 0.20 |
| | $S^2GT$-T | **98.50 ± 0.10** | *98.50 ± 0.10* | 98.10 ± 0.40 | 98.10 ± 0.40 | 98.10 ± 0.40 | *98.50 ± 0.10* |
| | $S^2GT_3$-T | *98.40 ± 0.50* | **98.70 ± 0.20** | **98.70 ± 0.20** | **98.70 ± 0.20** | **98.70 ± 0.20** | **98.70 ± 0.20** |

## G.2 Additional Empirical Evaluation

### G.2.1 Runtime

Table 6: Runtime (s ± std.) for all axis-aligned approaches separated into TDIDT and Non-Greedy approaches by dashed lines per dataset. Best approach is **bolded**, second-best is *italicized*.

| dataset | model | 2 | 3 | 4 | 5 | 6 | Best |
|---|---|---|---|---|---|---|---|
| adult | CART | **0.036 ± 0.000** | *0.048 ± 0.000* | *0.059 ± 0.001* | *0.071 ± 0.001* | *0.083 ± 0.000* | *0.083 ± 0.000* |
| | HSTree | **0.036 ± 0.000** | **0.046 ± 0.001** | **0.058 ± 0.001** | **0.067 ± 0.000** | **0.077 ± 0.001** | **0.077 ± 0.001** |
| | SERDT | 20.560 ± 0.037 | 30.489 ± 0.040 | 36.904 ± 0.157 | 44.494 ± 0.720 | 52.905 ± 0.963 | 52.905 ± 0.963 |
| | SGT-C | *0.386 ± 0.009* | 0.596 ± 0.009 | 0.894 ± 0.016 | 1.120 ± 0.023 | 1.144 ± 0.033 | 1.144 ± 0.033 |
| | SGT$_3$-C | 0.545 ± 0.005 | 1.069 ± 0.006 | 1.503 ± 0.013 | 2.723 ± 0.029 | 1.729 ± 0.082 | 1.729 ± 0.082 |
| | AxTAO | *1.303 ± 0.004* | *1.708 ± 0.194* | 6.007 ± 0.445 | 7.710 ± 0.562 | 9.812 ± 0.602 | 9.812 ± 0.602 |
| | DPDT | **0.311 ± 0.003** | 3.499 ± 0.187 | 7.731 ± 0.032 | 9.249 ± 0.088 | *9.276 ± 0.667* | *9.276 ± 0.667* |
| | SPLIT | 1.363 ± 0.516 | **1.353 ± 0.511** | 7.286 ± 2.613 | **3.477 ± 0.208** | **7.823 ± 1.706** | **7.823 ± 1.706** |
| | SGT-T | 2.029 ± 0.020 | 2.522 ± 0.879 | **4.038 ± 1.353** | 37.314 ± 1.389 | 41.237 ± 23.798 | 41.237 ± 23.798 |
| | SGT$_3$-T | 2.806 ± 0.123 | 7.232 ± 0.066 | 22.743 ± 19.981 | 50.972 ± 34.416 | 49.766 ± 28.622 | 49.766 ± 28.622 |
| avila | CART | — | — | **0.050 ± 0.000** | **0.061 ± 0.002** | **0.069 ± 0.000** | **0.069 ± 0.000** |

| Dataset | Method | | | | | | |
|---|---|---|---|---|---|---|---|
| | HSTree | — | — | *0.054 ± 0.002* | **0.061 ± 0.000** | *0.072 ± 0.002* | *0.072 ± 0.002* |
| | SERDT | — | — | 2.734 ± 0.010 | 3.344 ± 0.021 | 3.995 ± 0.012 | 3.995 ± 0.012 |
| | SGT-C | — | — | 1.711 ± 0.036 | *2.343 ± 0.007* | 2.733 ± 0.088 | 2.733 ± 0.088 |
| | SGT₃-C | — | — | 6.553 ± 0.210 | 8.060 ± 0.358 | 7.817 ± 0.096 | 7.817 ± 0.096 |
| | AxTAO | — | — | **0.970 ± 0.319** | **1.681 ± 0.033** | **3.038 ± 1.320** | **3.038 ± 1.320** |
| | DPDT | — | — | *3.042 ± 0.201* | *7.466 ± 0.303* | *10.148 ± 0.460* | *10.148 ± 0.460* |
| | SPLIT | — | — | 48.450 ± 9.734 | 104.178 ± 1.324 | 70.139 ± 8.502 | 70.139 ± 8.502 |
| | SGT-T | — | — | 6.837 ± 1.515 | 16.545 ± 3.764 | 31.646 ± 22.865 | 31.646 ± 22.865 |
| | SGT₃-T | — | — | 27.996 ± 7.018 | 28.331 ± 1.424 | 21.534 ± 0.983 | 21.534 ± 0.983 |
| bank | CART | **0.002 ± 0.000** | **0.002 ± 0.000** | **0.002 ± 0.000** | **0.003 ± 0.000** | **0.003 ± 0.000** | **0.003 ± 0.000** |
| | HSTree | **0.002 ± 0.000** | *0.003 ± 0.000* | *0.003 ± 0.000* | **0.003 ± 0.000** | **0.003 ± 0.000** | **0.003 ± 0.000** |
| | SERDT | *0.033 ± 0.000* | 0.050 ± 0.000 | 0.068 ± 0.000 | *0.085 ± 0.003* | *0.093 ± 0.002* | *0.093 ± 0.002* |
| | SGT-C | 0.101 ± 0.001 | 0.138 ± 0.004 | 0.165 ± 0.015 | 0.184 ± 0.008 | 0.179 ± 0.010 | 0.165 ± 0.015 |
| | SGT₃-C | 0.142 ± 0.000 | 0.187 ± 0.002 | 0.217 ± 0.008 | 0.220 ± 0.013 | 0.216 ± 0.016 | 0.217 ± 0.008 |
| | AxTAO | **0.036 ± 0.017** | *0.115 ± 0.025* | **0.147 ± 0.022** | **0.256 ± 0.062** | *0.259 ± 0.083* | *0.256 ± 0.062* |
| | DPDT | *0.040 ± 0.007* | **0.041 ± 0.001** | 0.617 ± 0.027 | *0.308 ± 0.003* | **0.128 ± 0.001** | **0.128 ± 0.001** |
| | SPLIT | 0.653 ± 0.609 | 0.648 ± 0.601 | 0.757 ± 0.538 | 2.495 ± 0.512 | 1.369 ± 0.302 | 2.495 ± 0.512 |
| | SGT-T | 0.726 ± 0.228 | 2.917 ± 0.473 | 3.262 ± 0.610 | 7.584 ± 7.667 | 3.388 ± 1.137 | 7.584 ± 7.667 |
| | SGT₃-T | 1.107 ± 0.339 | 4.162 ± 4.690 | 2.019 ± 0.634 | 2.130 ± 0.663 | 2.101 ± 0.646 | 2.019 ± 0.634 |
| bean | CART | — | *0.107 ± 0.001* | *0.135 ± 0.001* | *0.160 ± 0.001* | *0.186 ± 0.002* | *0.186 ± 0.002* |
| | HSTree | — | **0.093 ± 0.002** | **0.119 ± 0.004** | **0.138 ± 0.001** | **0.161 ± 0.003** | **0.161 ± 0.003** |
| | SERDT | — | 3.861 ± 0.007 | 4.996 ± 0.009 | 6.197 ± 0.009 | 7.345 ± 0.007 | 7.345 ± 0.007 |
| | SGT-C | — | 1.892 ± 0.027 | 2.420 ± 0.071 | 1.797 ± 0.041 | 2.328 ± 0.061 | 2.328 ± 0.061 |
| | SGT₃-C | — | 1.179 ± 0.009 | 2.670 ± 0.076 | 4.684 ± 0.085 | 7.103 ± 0.125 | 7.103 ± 0.125 |
| | AxTAO | — | **1.367 ± 0.054** | **2.008 ± 0.048** | **3.189 ± 0.589** | **4.472 ± 0.106** | **4.472 ± 0.106** |
| | DPDT | — | *3.258 ± 0.174* | *5.503 ± 0.286* | *7.565 ± 0.361* | *5.889 ± 0.337* | *5.889 ± 0.337* |
| | SPLIT | — | 38.276 ± 7.431 | 42.074 ± 18.018 | 55.909 ± 14.860 | 71.596 ± 12.543 | 71.596 ± 12.543 |
| | SGT-T | — | 6.180 ± 0.111 | 19.833 ± 9.656 | 18.729 ± 4.516 | 43.396 ± 24.977 | 43.396 ± 24.977 |
| | SGT₃-T | — | 8.404 ± 0.343 | 14.332 ± 2.636 | 43.177 ± 2.059 | 26.883 ± 1.676 | 43.177 ± 2.059 |
| bidding | CART | **0.005 ± 0.000** | **0.005 ± 0.000** | *0.006 ± 0.000* | **0.005 ± 0.000** | **0.005 ± 0.000** | **0.005 ± 0.000** |
| | HSTree | **0.005 ± 0.000** | **0.005 ± 0.000** | *0.006 ± 0.000* | *0.006 ± 0.000* | *0.006 ± 0.000* | *0.006 ± 0.000* |
| | SERDT | 0.190 ± 0.002 | 0.311 ± 0.001 | 0.334 ± 0.007 | 0.341 ± 0.002 | 0.353 ± 0.003 | 0.353 ± 0.003 |
| | SGT-C | *0.132 ± 0.001* | *0.207 ± 0.002* | 0.208 ± 0.004 | 0.256 ± 0.004 | 0.333 ± 0.011 | 0.333 ± 0.011 |
| | SGT₃-C | 0.276 ± 0.007 | 0.479 ± 0.025 | 0.533 ± 0.040 | 0.534 ± 0.042 | 0.544 ± 0.043 | 0.479 ± 0.025 |
| | AxTAO | *0.070 ± 0.001* | **0.186 ± 0.141** | **0.123 ± 0.032** | **0.234 ± 0.091** | **0.204 ± 0.064** | **0.234 ± 0.091** |
| | DPDT | **0.020 ± 0.001** | 0.426 ± 0.068 | 0.264 ± 0.001 | 0.955 ± 0.143 | 0.243 ± 0.012 | 0.264 ± 0.001 |
| | SPLIT | 0.636 ± 0.107 | 0.647 ± 0.111 | 1.013 ± 0.145 | 1.077 ± 0.176 | 1.031 ± 0.151 | 1.031 ± 0.151 |
| | SGT-T | 0.632 ± 0.175 | 1.212 ± 0.532 | 1.325 ± 0.043 | 1.656 ± 0.494 | 4.025 ± 0.473 | 1.325 ± 0.043 |
| | SGT₃-T | 1.090 ± 0.009 | 2.184 ± 1.188 | 2.432 ± 1.080 | 2.393 ± 0.967 | 2.472 ± 1.033 | 2.432 ± 1.080 |
| covtype | CART | — | *1.008 ± 0.009* | *1.321 ± 0.011* | *1.636 ± 0.022* | *1.977 ± 0.021* | *1.977 ± 0.021* |
| | HSTree | — | **0.913 ± 0.011** | **1.236 ± 0.103** | **1.466 ± 0.025** | **1.723 ± 0.012** | **1.723 ± 0.012** |
| | SERDT | — | 281.991 ± 0.979 | 360.049 ± 1.806 | 447.986 ± 0.312 | 534.572 ± 1.955 | 534.572 ± 1.955 |
| | SGT-C | — | 6.135 ± 0.034 | 11.224 ± 0.014 | 10.618 ± 0.064 | 15.770 ± 0.096 | 15.770 ± 0.096 |
| | SGT₃-C | — | 8.910 ± 0.052 | 14.011 ± 0.071 | 19.099 ± 0.151 | 29.561 ± 0.430 | 29.561 ± 0.430 |
| | AxTAO | — | **30.698 ± 0.183** | **52.465 ± 12.625** | **99.446 ± 16.476** | *170.256 ± 28.126* | *170.256 ± 28.126* |
| | DPDT | — | 124.065 ± 0.671 | 207.358 ± 1.485 | 298.735 ± 1.185 | 415.134 ± 1.714 | 415.134 ± 1.714 |
| | SPLIT | — | 912.950 ± 157.383 | 901.083 ± 169.281 | 877.478 ± 164.931 | 881.974 ± 153.045 | 912.950 ± 157.383 |
| | SGT-T | — | *34.679 ± 7.011* | *56.332 ± 5.875* | *100.787 ± 11.809* | **148.474 ± 9.684** | **148.474 ± 9.684** |
| | SGT₃-T | — | 73.772 ± 15.480 | 122.400 ± 38.379 | 397.392 ± 85.282 | 728.959 ± 118.892 | 728.959 ± 118.892 |
| electricity | CART | *0.031 ± 0.000* | *0.047 ± 0.000* | *0.062 ± 0.000* | *0.077 ± 0.000* | *0.093 ± 0.001* | *0.093 ± 0.001* |
| | HSTree | **0.029 ± 0.000** | **0.043 ± 0.001** | **0.057 ± 0.000** | **0.070 ± 0.001** | **0.085 ± 0.000** | **0.085 ± 0.000** |
| | SERDT | 2.923 ± 0.036 | 4.184 ± 0.010 | 5.262 ± 0.046 | 6.237 ± 0.025 | 7.454 ± 0.093 | 7.454 ± 0.093 |
| | SGT-C | 0.396 ± 0.002 | 0.438 ± 0.006 | 0.880 ± 0.007 | 1.356 ± 0.011 | 2.145 ± 0.177 | 2.145 ± 0.177 |
| | SGT₃-C | 0.646 ± 0.014 | 1.138 ± 0.012 | 2.301 ± 0.081 | 4.603 ± 0.381 | 7.767 ± 0.918 | 7.767 ± 0.918 |
| | AxTAO | **0.438 ± 0.003** | **1.408 ± 0.008** | **2.020 ± 0.007** | **6.282 ± 2.817** | **10.311 ± 0.628** | **10.311 ± 0.628** |
| | DPDT | 1.539 ± 0.008 | *3.158 ± 0.016* | *8.492 ± 0.024* | 13.254 ± 0.028 | *19.481 ± 0.057* | *19.481 ± 0.057* |
| | SPLIT | *0.458 ± 0.115* | 3.509 ± 0.578 | 12.528 ± 2.394 | *11.515 ± 0.450* | 22.478 ± 0.111 | 22.478 ± 0.111 |
| | SGT-T | 1.505 ± 0.374 | 4.662 ± 0.122 | 11.463 ± 1.274 | 31.993 ± 5.000 | 72.724 ± 18.726 | 72.724 ± 18.726 |
| | SGT₃-T | 6.664 ± 1.547 | 20.198 ± 7.617 | 30.156 ± 8.124 | 79.497 ± 26.969 | 92.449 ± 6.446 | 92.449 ± 6.446 |
| eucalyptus | CART | — | **0.006 ± 0.000** | **0.007 ± 0.000** | **0.008 ± 0.000** | **0.009 ± 0.000** | **0.009 ± 0.000** |
| | HSTree | — | *0.007 ± 0.000* | *0.009 ± 0.000* | *0.010 ± 0.000* | *0.011 ± 0.000* | *0.011 ± 0.000* |
| | SERDT | — | 4.464 ± 0.019 | 5.694 ± 0.082 | 7.129 ± 0.144 | 8.441 ± 0.177 | 8.441 ± 0.177 |
| | SGT-C | — | 0.319 ± 0.048 | 0.523 ± 0.063 | 0.799 ± 0.046 | 0.995 ± 0.030 | 0.995 ± 0.030 |
| | SGT₃-C | — | 0.688 ± 0.048 | 2.559 ± 0.275 | 2.658 ± 0.305 | 1.765 ± 0.119 | 2.559 ± 0.275 |
| | AxTAO | — | 4.184 ± 0.028 | 8.124 ± 4.652 | 7.608 ± 0.738 | 33.022 ± 12.345 | 33.022 ± 12.345 |
| | DPDT | — | **0.099 ± 0.006** | **0.096 ± 0.002** | **0.952 ± 0.225** | **3.289 ± 0.128** | *3.289 ± 0.128* |
| | SPLIT | — | *0.267 ± 0.003* | 3.009 ± 0.936 | 2.925 ± 0.568 | 8.046 ± 2.917 | 8.046 ± 2.917 |
| | SGT-T | — | 1.878 ± 0.407 | 7.580 ± 6.658 | 6.044 ± 1.326 | *7.649 ± 1.954* | 7.649 ± 1.954 |
| | SGT₃-T | — | 3.240 ± 1.273 | *2.730 ± 0.543* | 15.882 ± 9.963 | 14.908 ± 8.775 | **3.240 ± 1.273** |
| eye-movements | CART | *0.023 ± 0.000* | *0.034 ± 0.000* | *0.044 ± 0.000* | *0.053 ± 0.000* | *0.062 ± 0.000* | *0.062 ± 0.000* |
| | HSTree | **0.021 ± 0.000** | **0.031 ± 0.001** | **0.040 ± 0.000** | **0.051 ± 0.002** | **0.060 ± 0.003** | **0.060 ± 0.003** |
| | SERDT | 1.427 ± 0.002 | 2.177 ± 0.004 | 2.844 ± 0.008 | 3.491 ± 0.017 | 4.302 ± 0.013 | 4.302 ± 0.013 |
| | SGT-C | 0.464 ± 0.009 | 0.910 ± 0.010 | 1.540 ± 0.020 | 2.501 ± 0.018 | 3.879 ± 0.329 | 3.879 ± 0.329 |

| | | | | | | | |
|---|---|---|---|---|---|---|---|
| | SGT$_3$-C | $0.661 \pm 0.015$ | $2.216 \pm 0.179$ | $4.876 \pm 0.325$ | $6.152 \pm 0.443$ | $12.470 \pm 0.314$ | $12.470 \pm 0.314$ |
| | AxTAO | $\mathbf{0.227 \pm 0.004}$ | $\mathbf{0.748 \pm 0.444}$ | $\mathbf{1.074 \pm 0.358}$ | $\mathbf{2.180 \pm 0.571}$ | $\mathbf{3.849 \pm 0.049}$ | $\mathbf{3.849 \pm 0.049}$ |
| | DPDT | $0.973 \pm 0.018$ | $3.191 \pm 0.031$ | $5.777 \pm 0.023$ | $9.090 \pm 0.044$ | $5.727 \pm 0.943$ | $5.727 \pm 0.943$ |
| | SPLIT | $4.920 \pm 1.249$ | $7.691 \pm 2.536$ | $6.656 \pm 1.058$ | $6.736 \pm 0.976$ | $12.934 \pm 1.627$ | $6.736 \pm 0.976$ |
| | SGT-T | $2.330 \pm 0.574$ | $5.937 \pm 1.198$ | $12.002 \pm 3.378$ | $24.338 \pm 4.959$ | $54.067 \pm 25.371$ | $54.067 \pm 25.371$ |
| | SGT$_3$-T | $4.173 \pm 0.033$ | $12.936 \pm 1.726$ | $16.618 \pm 5.723$ | $32.929 \pm 8.677$ | $62.689 \pm 18.553$ | $62.689 \pm 18.553$ |
| eye-state | CART | $0.015 \pm 0.000$ | $0.022 \pm 0.000$ | $0.028 \pm 0.000$ | $0.036 \pm 0.000$ | $0.044 \pm 0.001$ | $0.044 \pm 0.001$ |
| | HSTree | $\mathbf{0.014 \pm 0.000}$ | $0.020 \pm 0.000$ | $0.026 \pm 0.000$ | $0.033 \pm 0.000$ | $\mathbf{0.040 \pm 0.000}$ | $\mathbf{0.040 \pm 0.000}$ |
| | SERDT | $0.905 \pm 0.036$ | $1.318 \pm 0.029$ | $1.615 \pm 0.033$ | $2.047 \pm 0.032$ | $2.495 \pm 0.071$ | $2.495 \pm 0.071$ |
| | SGT-C | $0.339 \pm 0.003$ | $0.505 \pm 0.002$ | $0.678 \pm 0.005$ | $1.171 \pm 0.005$ | $2.087 \pm 0.100$ | $2.087 \pm 0.100$ |
| | SGT$_3$-C | $0.574 \pm 0.007$ | $1.063 \pm 0.008$ | $2.511 \pm 0.133$ | $4.777 \pm 0.111$ | $6.192 \pm 0.148$ | $6.192 \pm 0.148$ |
| | AxTAO | $\mathbf{0.449 \pm 0.081}$ | $1.122 \pm 0.191$ | $1.808 \pm 0.049$ | $\mathbf{2.954 \pm 0.564}$ | $5.547 \pm 1.728$ | $5.547 \pm 1.728$ |
| | DPDT | $0.773 \pm 0.012$ | $\mathbf{0.485 \pm 0.101}$ | $\mathbf{1.461 \pm 0.005}$ | $3.796 \pm 1.080$ | $\mathbf{3.550 \pm 0.032}$ | $\mathbf{3.550 \pm 0.032}$ |
| | SPLIT | $0.465 \pm 0.251$ | $2.575 \pm 0.490$ | $2.502 \pm 0.443$ | $112.005 \pm 25.574$ | $44.632 \pm 11.038$ | $44.632 \pm 11.038$ |
| | SGT-T | $2.187 \pm 0.956$ | $4.029 \pm 0.029$ | $9.571 \pm 2.318$ | $26.239 \pm 6.816$ | $38.509 \pm 12.263$ | $38.509 \pm 12.263$ |
| | SGT$_3$-T | $4.998 \pm 1.079$ | $14.493 \pm 4.296$ | $23.130 \pm 0.825$ | $37.413 \pm 1.708$ | $84.647 \pm 14.679$ | $84.647 \pm 14.679$ |
| fault | CART | — | $\mathbf{0.012 \pm 0.000}$ | $0.015 \pm 0.000$ | $0.018 \pm 0.000$ | $0.020 \pm 0.000$ | $0.020 \pm 0.000$ |
| | HSTree | — | $\mathbf{0.012 \pm 0.000}$ | $0.016 \pm 0.000$ | $0.019 \pm 0.000$ | $0.021 \pm 0.000$ | $0.021 \pm 0.000$ |
| | SERDT | — | $0.455 \pm 0.003$ | $0.605 \pm 0.004$ | $0.777 \pm 0.004$ | $0.958 \pm 0.006$ | $0.958 \pm 0.006$ |
| | SGT-C | — | $0.641 \pm 0.032$ | $1.044 \pm 0.045$ | $1.945 \pm 0.153$ | $2.599 \pm 0.102$ | $2.599 \pm 0.102$ |
| | SGT$_3$-C | — | $1.928 \pm 0.047$ | $2.356 \pm 0.170$ | $4.842 \pm 0.093$ | $5.925 \pm 0.192$ | $5.925 \pm 0.192$ |
| | AxTAO | — | $\mathbf{0.282 \pm 0.108}$ | $0.461 \pm 0.082$ | $0.624 \pm 0.134$ | $\mathbf{1.128 \pm 0.190}$ | $\mathbf{1.128 \pm 0.190}$ |
| | DPDT | — | $0.609 \pm 0.001$ | $\mathbf{0.332 \pm 0.006}$ | $1.336 \pm 0.010$ | $1.217 \pm 0.008$ | $1.217 \pm 0.008$ |
| | SPLIT | — | $9.265 \pm 5.923$ | $6.138 \pm 0.328$ | $9.549 \pm 5.396$ | $8.209 \pm 0.269$ | $8.209 \pm 0.269$ |
| | SGT-T | — | $4.148 \pm 0.724$ | $7.013 \pm 1.576$ | $12.704 \pm 1.537$ | $64.566 \pm 1.261$ | $64.566 \pm 1.261$ |
| | SGT$_3$-T | — | $5.172 \pm 0.935$ | $12.787 \pm 0.413$ | $10.829 \pm 1.168$ | $42.420 \pm 8.558$ | $42.420 \pm 8.558$ |
| gas-drift | CART | — | $0.891 \pm 0.005$ | $1.149 \pm 0.002$ | $1.387 \pm 0.013$ | $1.569 \pm 0.014$ | $1.569 \pm 0.014$ |
| | HSTree | — | $\mathbf{0.763 \pm 0.011}$ | $\mathbf{0.980 \pm 0.003}$ | $\mathbf{1.217 \pm 0.058}$ | $\mathbf{1.325 \pm 0.004}$ | $\mathbf{1.325 \pm 0.004}$ |
| | SERDT | — | $29.884 \pm 0.096$ | $39.047 \pm 0.190$ | $48.294 \pm 0.116$ | $57.239 \pm 0.289$ | $57.239 \pm 0.289$ |
| | SGT-C | — | $7.714 \pm 0.016$ | $14.076 \pm 0.137$ | $17.205 \pm 0.505$ | $20.722 \pm 0.314$ | $20.722 \pm 0.314$ |
| | SGT$_3$-C | — | $35.316 \pm 0.725$ | $26.835 \pm 0.653$ | $40.630 \pm 1.449$ | $47.566 \pm 0.873$ | $47.566 \pm 0.873$ |
| | AxTAO | — | $8.720 \pm 1.664$ | $13.863 \pm 2.509$ | $25.127 \pm 6.892$ | $34.319 \pm 7.415$ | $34.319 \pm 7.415$ |
| | DPDT | — | $36.866 \pm 0.395$ | $40.200 \pm 0.063$ | $29.836 \pm 0.182$ | $37.349 \pm 0.300$ | $37.349 \pm 0.300$ |
| | SPLIT | — | $135.935 \pm 4.754$ | $120.914 \pm 8.115$ | $371.082 \pm 39.808$ | $141.962 \pm 6.909$ | $141.962 \pm 6.909$ |
| | SGT-T | — | $30.638 \pm 7.954$ | $43.388 \pm 7.772$ | $109.836 \pm 24.944$ | $89.735 \pm 5.680$ | $89.735 \pm 5.680$ |
| | SGT$_3$-T | — | $89.856 \pm 31.211$ | $101.685 \pm 61.669$ | $92.602 \pm 9.560$ | $257.699 \pm 186.309$ | $257.699 \pm 186.309$ |
| higgs | CART | $\mathbf{0.045 \pm 1.535}$ | $\mathbf{0.057 \pm 1.134}$ | $0.076 \pm 2.312$ | $0.112 \pm 2.103$ | $0.127 \pm 1.230$ | $0.127 \pm 1.230$ |
| | HSTree | $37.872 \pm 1.412$ | $51.436 \pm 2.121$ | $71.554 \pm 1.768$ | $87.323 \pm 1.203$ | $111.087 \pm 2.392$ | $111.087 \pm 2.392$ |
| | SERDT | $54.929 \pm 2.123$ | $60.801 \pm 3.237$ | $82.336 \pm 2.899$ | $107.199 \pm 4.434$ | $127.199 \pm 5.187$ | $127.199 \pm 5.187$ |
| | SGT-C | $38.987 \pm 5.122$ | $65.121 \pm 5.102$ | $84.432 \pm 6.939$ | $103.653 \pm 7.729$ | $124.762 \pm 8.092$ | $103.653 \pm 7.729$ |
| | SGT$_3$-C | $67.214 \pm 4.032$ | $75.952 \pm 7.323$ | $83.548 \pm 7.122$ | $111.954 \pm 8.892$ | $141.062 \pm 9.321$ | $187.612 \pm 11.321$ |
| | AxTAO | $174.298 \pm 12.321$ | $296.307 \pm 19.010$ | $361.494 \pm 21.210$ | $448.253 \pm 19.453$ | $587.211 \pm 20.227$ | $587.211 \pm 20.227$ |
| | DPDT | $\mathbf{103.342 \pm 9.101}$ | $\mathbf{270.478 \pm 18.23}$ | $430.524 \pm 24.434$ | $499.492 \pm 22.112$ | $645.241 \pm 29.110$ | $645.241 \pm 29.110$ |
| | SPLIT | $643.232 \pm 29.211$ | $764.445 \pm 22.867$ | $900.000 \pm 0.000$ | $900.000 \pm 0.000$ | $900.000 \pm 0.000$ | $900.000 \pm 0.000$ |
| | SGT-T | $299.308 \pm 7.472$ | $425.915 \pm 12.825$ | $563.486 \pm 32.081$ | $743.802 \pm 24.001$ | $900.000 \pm 0.000$ | $900.000 \pm 0.000$ |
| | SGT$_3$-T | $389.313 \pm 22.072$ | $476.130 \pm 11.432$ | $628.492 \pm 34.403$ | $900.000 \pm 0.000$ | $900.000 \pm 0.000$ | $900.000 \pm 0.000$ |
| htru | CART | $\mathbf{0.026 \pm 0.000}$ | $\mathbf{0.037 \pm 0.001}$ | $\mathbf{0.049 \pm 0.000}$ | $\mathbf{0.064 \pm 0.004}$ | $\mathbf{0.058 \pm 0.002}$ | $\mathbf{0.049 \pm 0.000}$ |
| | HSTree | $0.027 \pm 0.001$ | $0.039 \pm 0.001$ | $0.050 \pm 0.000$ | $0.068 \pm 0.007$ | $0.059 \pm 0.003$ | $0.068 \pm 0.007$ |
| | SERDT | $0.983 \pm 0.030$ | $1.504 \pm 0.005$ | $1.930 \pm 0.006$ | $2.340 \pm 0.008$ | $2.754 \pm 0.103$ | $1.930 \pm 0.006$ |
| | SGT-C | $0.328 \pm 0.002$ | $0.456 \pm 0.004$ | $0.646 \pm 0.020$ | $0.586 \pm 0.065$ | $0.561 \pm 0.020$ | $0.456 \pm 0.004$ |
| | SGT$_3$-C | $0.495 \pm 0.003$ | $0.763 \pm 0.018$ | $0.731 \pm 0.100$ | $0.996 \pm 0.035$ | $0.882 \pm 0.119$ | $0.731 \pm 0.100$ |
| | AxTAO | $\mathbf{0.278 \pm 0.084}$ | $0.430 \pm 0.145$ | $0.842 \pm 0.122$ | $1.275 \pm 0.679$ | $1.164 \pm 0.626$ | $1.164 \pm 0.626$ |
| | DPDT | $0.304 \pm 0.079$ | $\mathbf{0.054 \pm 0.001}$ | $\mathbf{0.483 \pm 0.044}$ | $\mathbf{0.051 \pm 0.001}$ | $\mathbf{0.055 \pm 0.001}$ | $\mathbf{0.483 \pm 0.044}$ |
| | SPLIT | $5.767 \pm 3.391$ | $0.757 \pm 0.085$ | $2.783 \pm 0.315$ | $9.423 \pm 3.014$ | $9.901 \pm 2.383$ | $9.901 \pm 2.383$ |
| | SGT-T | $1.516 \pm 0.019$ | $1.819 \pm 0.601$ | $5.049 \pm 1.393$ | $16.676 \pm 10.774$ | $17.290 \pm 2.296$ | $17.290 \pm 2.296$ |
| | SGT$_3$-T | $1.566 \pm 0.341$ | $1.352 \pm 0.293$ | $1.635 \pm 0.351$ | $1.586 \pm 0.319$ | $1.665 \pm 0.335$ | $1.352 \pm 0.293$ |
| magic | CART | $\mathbf{0.023 \pm 0.000}$ | $\mathbf{0.034 \pm 0.001}$ | $\mathbf{0.044 \pm 0.000}$ | $\mathbf{0.064 \pm 0.001}$ | $0.074 \pm 0.001$ | $0.074 \pm 0.001$ |
| | HSTree | $\mathbf{0.023 \pm 0.001}$ | $\mathbf{0.034 \pm 0.000}$ | $0.045 \pm 0.000$ | $\mathbf{0.064 \pm 0.000}$ | $0.075 \pm 0.001$ | $0.075 \pm 0.001$ |
| | SERDT | $0.824 \pm 0.001$ | $1.239 \pm 0.006$ | $1.605 \pm 0.004$ | $2.078 \pm 0.010$ | $2.447 \pm 0.007$ | $2.447 \pm 0.007$ |
| | SGT-C | $0.312 \pm 0.000$ | $0.477 \pm 0.003$ | $0.689 \pm 0.003$ | $1.071 \pm 0.042$ | $1.876 \pm 0.061$ | $1.876 \pm 0.061$ |
| | SGT$_3$-C | $0.325 \pm 0.006$ | $0.948 \pm 0.028$ | $1.621 \pm 0.135$ | $2.710 \pm 0.089$ | $3.284 \pm 0.353$ | $2.710 \pm 0.089$ |
| | AxTAO | $0.349 \pm 0.081$ | $0.502 \pm 0.002$ | $1.173 \pm 0.036$ | $2.349 \pm 0.095$ | $2.645 \pm 0.151$ | $2.645 \pm 0.151$ |
| | DPDT | $\mathbf{0.231 \pm 0.059}$ | $1.884 \pm 0.007$ | $2.812 \pm 0.004$ | $8.062 \pm 0.050$ | $11.521 \pm 0.121$ | $11.521 \pm 0.121$ |
| | SPLIT | $2.514 \pm 0.667$ | $0.667 \pm 0.047$ | $3.353 \pm 1.291$ | $240.554 \pm 202.408$ | $69.764 \pm 29.671$ | $69.764 \pm 29.671$ |
| | SGT-T | $1.687 \pm 0.336$ | $4.590 \pm 1.273$ | $7.290 \pm 4.003$ | $18.635 \pm 7.555$ | $38.378 \pm 36.280$ | $38.378 \pm 36.280$ |
| | SGT$_3$-T | $3.670 \pm 0.536$ | $9.933 \pm 1.604$ | $20.429 \pm 2.513$ | $27.240 \pm 7.119$ | $52.910 \pm 15.146$ | $52.910 \pm 15.146$ |
| mini-boone | CART | $1.327 \pm 0.003$ | $2.377 \pm 0.006$ | $3.169 \pm 0.001$ | $3.292 \pm 0.013$ | $3.920 \pm 0.012$ | $3.920 \pm 0.012$ |
| | HSTree | $\mathbf{1.234 \pm 0.010}$ | $\mathbf{2.112 \pm 0.006}$ | $\mathbf{2.923 \pm 0.260}$ | $\mathbf{2.987 \pm 0.008}$ | $\mathbf{3.561 \pm 0.015}$ | $\mathbf{3.561 \pm 0.015}$ |
| | SERDT | $59.664 \pm 0.425$ | $94.866 \pm 0.362$ | $121.947 \pm 0.338$ | $148.979 \pm 0.959$ | $146.750 \pm 0.547$ | $146.750 \pm 0.547$ |
| | SGT-C | $11.023 \pm 0.078$ | $12.995 \pm 0.129$ | $16.074 \pm 0.116$ | $20.609 \pm 0.135$ | $25.352 \pm 0.062$ | $25.352 \pm 0.062$ |
| | SGT$_3$-C | $13.670 \pm 0.121$ | $18.370 \pm 0.194$ | $20.669 \pm 0.178$ | $25.379 \pm 1.723$ | $66.695 \pm 4.619$ | $66.695 \pm 4.619$ |
| | AxTAO | $\mathbf{3.156 \pm 0.058}$ | $\mathbf{10.769 \pm 1.910}$ | $19.074 \pm 2.473$ | $28.574 \pm 3.315$ | $36.061 \pm 5.404$ | $36.061 \pm 5.404$ |

| Dataset | Method | | | | | | |
|---|---|---|---|---|---|---|---|
| | DPDT | 9.395 ± 1.738 | 148.946 ± 1.520 | 256.252 ± 1.234 | *47.860 ± 0.151* | 453.668 ± 4.310 | 453.668 ± 4.310 |
| | SPLIT | *8.502 ± 0.052* | 83.926 ± 33.161 | 188.186 ± 19.342 | 212.160 ± 16.792 | 152.526 ± 14.973 | 212.160 ± 16.792 |
| | SGT-T | 26.137 ± 5.566 | *33.979 ± 2.929* | 53.121 ± 7.172 | 66.677 ± 12.084 | *148.090 ± 52.451* | *148.090 ± 52.451* |
| | SGT₃-T | 63.035 ± 8.172 | 60.582 ± 45.599 | 71.228 | 125.311 ± 14.876 | 256.178 ± 7.425 | 256.178 ± 7.425 |
| mushroom | CART | **0.007 ± 0.000** | **0.009 ± 0.000** | **0.010 ± 0.000** | **0.009 ± 0.000** | **0.009 ± 0.000** | **0.009 ± 0.000** |
| | HSTree | **0.007 ± 0.000** | **0.009 ± 0.000** | **0.010 ± 0.001** | **0.009 ± 0.000** | *0.010 ± 0.001* | **0.009 ± 0.000** |
| | SERDT | 2.501 ± 0.030 | 3.648 ± 0.029 | 4.413 ± 0.083 | 5.367 ± 0.089 | 6.145 ± 0.225 | 6.145 ± 0.225 |
| | SGT-C | *0.185 ± 0.003* | 0.331 ± 0.008 | 0.399 ± 0.026 | *0.327 ± 0.004* | 0.330 ± 0.004 | *0.327 ± 0.004* |
| | SGT₃-C | 0.274 ± 0.003 | *0.326 ± 0.005* | 0.392 ± 0.028 | 0.395 ± 0.003 | 0.400 ± 0.002 | 0.392 ± 0.028 |
| | AxTAO | 1.110 ± 0.011 | 1.324 ± 0.549 | 1.471 ± 0.004 | *1.463 ± 0.004* | 3.212 ± 0.009 | 1.463 ± 0.004 |
| | DPDT | *0.264 ± 0.015* | **0.742 ± 0.024** | 1.504 ± 0.085 | 1.718 ± 0.176 | **0.725 ± 0.029** | 1.504 ± 0.085 |
| | SPLIT | **0.123 ± 0.011** | 1.065 ± 0.484 | **1.123 ± 0.470** | **0.865 ± 0.241** | 0.895 ± 0.210 | **0.865 ± 0.241** |
| | SGT-T | 0.557 ± 0.002 | 10.795 ± 0.073 | 1.596 ± 0.020 | 1.981 ± 0.212 | 2.027 ± 0.206 | 1.981 ± 0.212 |
| | SGT₃-T | 0.919 ± 0.007 | *1.017 ± 0.021* | 1.462 ± 0.020 | 1.490 ± 0.012 | 1.535 ± 0.019 | *1.462 ± 0.020* |
| occupancy | CART | **0.012 ± 0.000** | **0.012 ± 0.000** | **0.012 ± 0.000** | **0.012 ± 0.000** | *0.025 ± 0.002* | *0.025 ± 0.002* |
| | HSTree | **0.012 ± 0.001** | *0.016 ± 0.000* | *0.018 ± 0.000* | *0.019 ± 0.001* | **0.017 ± 0.003** | **0.017 ± 0.003** |
| | SERDT | 0.546 ± 0.011 | 0.801 ± 0.004 | 1.113 ± 0.055 | 1.254 ± 0.103 | 1.405 ± 0.154 | 1.113 ± 0.055 |
| | SGT-C | *0.148 ± 0.002* | 0.186 ± 0.003 | 0.176 ± 0.004 | 0.197 ± 0.006 | 0.405 ± 0.024 | 0.405 ± 0.024 |
| | SGT₃-C | 0.216 ± 0.002 | 0.222 ± 0.036 | 0.365 ± 0.015 | 0.511 ± 0.033 | 0.579 ± 0.068 | 0.365 ± 0.015 |
| | AxTAO | *0.183 ± 0.001* | **0.098 ± 0.002** | **0.111 ± 0.009** | 1.085 ± 0.068 | **1.626 ± 0.475** | **1.626 ± 0.475** |
| | DPDT | **0.088 ± 0.001** | *1.757 ± 0.002* | 1.722 ± 0.300 | 4.697 ± 0.181 | 5.932 ± 0.196 | 4.697 ± 0.181 |
| | SPLIT | 0.206 ± 0.019 | 4.228 ± 1.873 | 4.165 ± 1.847 | *4.192 ± 1.823* | *4.248 ± 1.824* | *4.192 ± 1.823* |
| | SGT-T | 0.541 ± 0.013 | 5.539 ± 4.348 | 2.112 ± 0.388 | 20.543 ± 11.216 | 27.321 ± 16.985 | 27.321 ± 16.985 |
| | SGT₃-T | 1.292 ± 0.249 | 2.159 ± 1.011 | 2.262 ± 0.160 | 14.579 ± 7.958 | 7.077 ± 7.276 | 7.077 ± 7.276 |
| page | CART | — | **0.012 ± 0.000** | 0.012 ± 0.000 | 0.012 ± 0.000 | 0.014 ± 0.000 | 0.012 ± 0.000 |
| | HSTree | — | *0.008 ± 0.000* | *0.013 ± 0.000* | *0.013 ± 0.000* | *0.015 ± 0.000* | *0.013 ± 0.000* |
| | SERDT | — | 0.335 ± 0.002 | 0.477 ± 0.018 | 0.589 ± 0.017 | 0.711 ± 0.022 | 0.711 ± 0.022 |
| | SGT-C | — | 0.263 ± 0.001 | 0.437 ± 0.022 | 0.696 ± 0.075 | 0.933 ± 0.015 | 0.696 ± 0.075 |
| | SGT₃-C | — | 0.817 ± 0.063 | 1.053 ± 0.122 | 1.318 ± 0.105 | 1.798 ± 0.142 | 1.318 ± 0.105 |
| | AxTAO | — | **0.399 ± 0.096** | **0.347 ± 0.137** | *0.601 ± 0.143* | *0.976 ± 0.366* | *0.976 ± 0.366* |
| | DPDT | — | 0.905 ± 0.007 | 1.071 ± 0.000 | **0.315 ± 0.010** | **0.423 ± 0.015** | **0.423 ± 0.015** |
| | SPLIT | — | 5.425 ± 2.155 | 8.316 ± 0.927 | 5.511 ± 1.992 | 9.737 ± 1.043 | 5.511 ± 1.992 |
| | SGT-T | — | 3.422 ± 1.302 | 4.174 ± 0.396 | 11.024 ± 9.302 | 24.833 ± 23.466 | 24.833 ± 23.466 |
| | SGT₃-T | — | 4.892 ± 0.064 | 9.753 ± 1.842 | 11.594 ± 8.558 | 23.490 ± 15.246 | 9.753 ± 1.842 |
| pendigits | CART | — | — | *0.021 ± 0.000* | *0.027 ± 0.000* | *0.033 ± 0.000* | *0.033 ± 0.000* |
| | HSTree | — | — | **0.020 ± 0.000** | **0.025 ± 0.000** | **0.030 ± 0.000** | **0.030 ± 0.000** |
| | SERDT | — | — | 2.127 ± 0.017 | 2.628 ± 0.032 | 3.216 ± 0.037 | 3.216 ± 0.037 |
| | SGT-C | — | — | 2.312 ± 0.020 | 3.443 ± 0.053 | 4.788 ± 0.033 | 4.788 ± 0.033 |
| | SGT₃-C | — | — | 4.421 ± 0.009 | 7.037 ± 0.124 | 8.468 ± 0.429 | 8.468 ± 0.429 |
| | AxTAO | — | — | *1.653 ± 0.364* | **3.087 ± 1.128** | **4.157 ± 0.779** | **4.157 ± 0.779** |
| | DPDT | — | — | **0.811 ± 0.004** | *3.948 ± 0.021* | *9.062 ± 0.068* | *9.062 ± 0.068* |
| | SPLIT | — | — | 28.828 ± 1.348 | 35.454 ± 1.105 | 48.427 ± 1.831 | 48.427 ± 1.831 |
| | SGT-T | — | — | 10.774 ± 1.335 | 18.070 ± 0.007 | 48.994 ± 24.878 | 48.994 ± 24.878 |
| | SGT₃-T | — | — | 32.178 ± 5.256 | 180.997 ± 9.845 | 282.237 ± 20.653 | 282.237 ± 20.653 |
| raisin | CART | **0.002 ± 0.000** | **0.002 ± 0.000** | **0.003 ± 0.000** | **0.003 ± 0.000** | **0.003 ± 0.000** | **0.003 ± 0.000** |
| | HSTree | **0.002 ± 0.000** | *0.003 ± 0.000* | **0.003 ± 0.000** | **0.003 ± 0.000** | **0.003 ± 0.000** | **0.003 ± 0.000** |
| | SERDT | *0.038 ± 0.000* | 0.058 ± 0.001 | *0.080 ± 0.000* | 0.096 ± 0.002 | *0.106 ± 0.001* | *0.080 ± 0.000* |
| | SGT-C | 0.137 ± 0.002 | 0.196 ± 0.011 | 0.263 ± 0.029 | 0.249 ± 0.034 | 0.317 ± 0.087 | 0.249 ± 0.034 |
| | SGT₃-C | 0.137 ± 0.001 | 0.241 ± 0.026 | 0.275 ± 0.037 | 0.129 ± 0.003 | 0.136 ± 0.002 | 0.241 ± 0.026 |
| | AxTAO | *0.040 ± 0.018* | **0.071 ± 0.057** | **0.110 ± 0.052** | **0.098 ± 0.025** | **0.081 ± 0.038** | **0.110 ± 0.052** |
| | DPDT | **0.016 ± 0.000** | 0.276 ± 0.001 | 0.542 ± 0.048 | 0.779 ± 0.064 | 1.060 ± 0.112 | 1.060 ± 0.112 |
| | SPLIT | 0.780 ± 0.394 | 0.613 ± 0.353 | 0.741 ± 0.303 | *0.485 ± 0.144* | *0.115 ± 0.024* | *0.741 ± 0.303* |
| | SGT-T | 0.764 ± 0.259 | 1.336 ± 0.303 | 2.336 ± 0.817 | 2.859 ± 0.880 | 2.654 ± 1.199 | 2.336 ± 0.817 |
| | SGT₃-T | 0.650 ± 0.260 | 1.859 ± 0.844 | 2.006 ± 0.631 | 2.078 ± 0.649 | 2.106 ± 0.676 | 1.859 ± 0.844 |
| rice | CART | **0.005 ± 0.000** | **0.007 ± 0.000** | **0.008 ± 0.000** | *0.010 ± 0.000* | *0.012 ± 0.001* | *0.010 ± 0.000* |
| | HSTree | **0.005 ± 0.000** | **0.007 ± 0.000** | *0.009 ± 0.000* | **0.008 ± 0.000** | *0.009 ± 0.001* | *0.009 ± 0.000* |
| | SERDT | 0.176 ± 0.021 | 0.255 ± 0.019 | 0.326 ± 0.002 | 0.414 ± 0.006 | 0.457 ± 0.002 | 0.326 ± 0.002 |
| | SGT-C | *0.134 ± 0.002* | *0.239 ± 0.003* | 0.337 ± 0.029 | 0.441 ± 0.062 | 0.541 ± 0.049 | 0.541 ± 0.049 |
| | SGT₃-C | 0.204 ± 0.002 | 0.311 ± 0.022 | 0.236 ± 0.018 | 0.232 ± 0.013 | 0.232 ± 0.017 | 0.204 ± 0.002 |
| | AxTAO | **0.058 ± 0.023** | **0.065 ± 0.024** | *0.279 ± 0.033* | *0.354 ± 0.117* | *0.527 ± 0.114* | *0.527 ± 0.114* |
| | DPDT | *0.121 ± 0.021* | *0.182 ± 0.006* | **0.176 ± 0.005** | **0.049 ± 0.006** | **0.059 ± 0.006** | **0.182 ± 0.006** |
| | SPLIT | 1.334 ± 0.386 | 0.648 ± 0.177 | 0.645 ± 0.167 | 0.644 ± 0.157 | 0.655 ± 0.148 | 1.334 ± 0.386 |
| | SGT-T | 0.550 ± 0.006 | 2.356 ± 1.251 | 5.808 ± 4.953 | 3.399 ± 0.706 | 7.415 ± 1.940 | 5.808 ± 4.953 |
| | SGT₃-T | 0.933 ± 0.349 | 6.232 ± 5.830 | 1.991 ± 0.849 | 2.316 ± 1.014 | 2.158 ± 0.941 | 1.991 ± 0.849 |
| room | CART | **0.006 ± 0.000** | **0.008 ± 0.000** | **0.010 ± 0.001** | **0.010 ± 0.001** | **0.010 ± 0.001** | **0.010 ± 0.001** |
| | HSTree | *0.007 ± 0.000* | *0.009 ± 0.000* | *0.011 ± 0.001* | *0.011 ± 0.000* | *0.011 ± 0.001* | *0.011 ± 0.000* |
| | SERDT | 0.541 ± 0.019 | 0.795 ± 0.005 | 1.069 ± 0.006 | 1.288 ± 0.009 | 1.546 ± 0.018 | 1.546 ± 0.018 |
| | SGT-C | 0.238 ± 0.004 | 0.535 ± 0.004 | 0.704 ± 0.027 | 0.770 ± 0.014 | 0.773 ± 0.012 | 0.773 ± 0.012 |
| | SGT₃-C | 0.478 ± 0.017 | 0.884 ± 0.053 | 0.938 ± 0.069 | 0.967 ± 0.075 | 0.932 ± 0.085 | 0.938 ± 0.069 |
| | AxTAO | *0.308 ± 0.001* | **0.506 ± 0.002** | *0.871 ± 0.166* | **0.751 ± 0.260** | **0.804 ± 0.272** | **0.804 ± 0.272** |
| | DPDT | **0.084 ± 0.007** | 0.738 ± 0.089 | **0.578 ± 0.006** | 2.410 ± 0.044 | *1.021 ± 0.072* | 2.410 ± 0.044 |
| | SPLIT | 0.327 ± 0.032 | *0.614 ± 0.129* | 5.658 ± 0.178 | 7.797 ± 0.979 | 6.261 ± 0.343 | 6.261 ± 0.343 |
| | SGT-T | 2.344 ± 0.318 | 3.408 ± 0.033 | 9.386 ± 5.627 | 6.525 ± 1.961 | 6.014 ± 2.769 | 6.014 ± 2.769 |

| | | | | | | | |
|---|---|---|---|---|---|---|---|
| | SGT$_3$-T | 1.972 ± 0.101 | 3.035 ± 0.150 | 3.632 ± 0.132 | 8.988 ± 4.116 | 12.089 ± 7.349 | 3.632 ± 0.132 |
| segment | CART | — | **0.008 ± 0.000** | **0.010 ± 0.000** | **0.010 ± 0.000** | **0.011 ± 0.000** | **0.011 ± 0.000** |
| | HSTree | — | *0.009 ± 0.000* | **0.010 ± 0.000** | *0.011 ± 0.000* | *0.012 ± 0.000* | *0.012 ± 0.000* |
| | SERDT | — | 0.346 ± 0.001 | *0.442 ± 0.012* | 0.504 ± 0.010 | 0.580 ± 0.018 | 0.580 ± 0.018 |
| | SGT-C | — | 0.410 ± 0.015 | 0.468 ± 0.016 | 0.844 ± 0.053 | 1.009 ± 0.109 | 1.009 ± 0.109 |
| | SGT$_3$-C | — | 0.545 ± 0.017 | 1.271 ± 0.084 | 1.234 ± 0.038 | 1.908 ± 0.096 | 1.234 ± 0.038 |
| | AxTAO | — | **0.198 ± 0.094** | **0.524 ± 0.158** | *0.790 ± 0.207* | **0.732 ± 0.038** | **0.732 ± 0.038** |
| | DPDT | — | *0.329 ± 0.030* | *1.110 ± 0.011* | **0.516 ± 0.012** | *1.142 ± 0.064* | *1.142 ± 0.064* |
| | SPLIT | — | 4.735 ± 1.463 | 4.778 ± 1.408 | 4.999 ± 1.451 | 5.115 ± 1.427 | 5.115 ± 1.427 |
| | SGT-T | — | 2.355 ± 0.157 | 4.999 ± 1.288 | 7.528 ± 1.169 | 7.641 ± 0.787 | 7.641 ± 0.787 |
| | SGT$_3$-T | — | 2.777 ± 0.930 | 7.510 ± 0.555 | 9.224 ± 2.099 | 19.645 ± 15.345 | 9.224 ± 2.099 |
| skin | CART | **0.043 ± 0.001** | **0.059 ± 0.000** | **0.067 ± 0.001** | **0.081 ± 0.000** | **0.091 ± 0.000** | **0.091 ± 0.000** |
| | HSTree | *0.049 ± 0.001* | *0.064 ± 0.001* | *0.080 ± 0.001* | *0.090 ± 0.002* | *0.099 ± 0.001* | *0.099 ± 0.001* |
| | SERDT | 2.930 ± 0.044 | 4.439 ± 0.101 | 5.699 ± 0.068 | 6.902 ± 0.101 | 8.007 ± 0.374 | 8.007 ± 0.374 |
| | SGT-C | 0.399 ± 0.009 | 0.520 ± 0.010 | 0.645 ± 0.008 | 0.776 ± 0.009 | 0.919 ± 0.018 | 0.919 ± 0.018 |
| | SGT$_3$-C | 0.398 ± 0.014 | 0.537 ± 0.009 | 0.717 ± 0.026 | 0.944 ± 0.013 | 1.117 ± 0.013 | 1.117 ± 0.013 |
| | AxTAO | *3.356 ± 0.022* | 10.456 ± 1.594 | 20.172 ± 8.560 | 21.687 ± 6.169 | 62.218 ± 3.298 | 62.218 ± 3.298 |
| | DPDT | **2.691 ± 0.023** | **1.058 ± 0.011** | **13.309 ± 0.004** | **19.033 ± 0.066** | **20.969 ± 0.209** | **20.969 ± 0.209** |
| | SPLIT | 82.441 ± 25.752 | 77.346 ± 23.421 | 96.253 ± 22.971 | 269.301 ± 47.638 | 86.591 ± 8.351 | 86.591 ± 8.351 |
| | SGT-T | 4.319 ± 1.118 | *9.555 ± 0.206* | 31.501 ± 0.513 | 41.956 ± 8.365 | 84.671 ± 0.898 | 84.671 ± 0.898 |
| | SGT$_3$-T | 9.801 ± 2.874 | 24.299 ± 10.724 | 83.188 ± 1.171 | 78.788 ± 26.254 | 176.502 ± 16.665 | 176.502 ± 16.665 |
| wilt | CART | **0.005 ± 0.000** | **0.007 ± 0.000** | **0.008 ± 0.000** | **0.009 ± 0.001** | **0.008 ± 0.001** | **0.009 ± 0.001** |
| | HSTree | *0.006 ± 0.000* | **0.007 ± 0.000** | *0.009 ± 0.000* | *0.010 ± 0.001* | *0.010 ± 0.001* | *0.010 ± 0.001* |
| | SERDT | 0.159 ± 0.019 | 0.212 ± 0.000 | 0.289 ± 0.002 | 0.368 ± 0.013 | 0.394 ± 0.022 | 0.368 ± 0.013 |
| | SGT-C | 0.127 ± 0.002 | *0.169 ± 0.001* | 0.252 ± 0.009 | 0.292 ± 0.033 | 0.316 ± 0.059 | 0.252 ± 0.009 |
| | SGT$_3$-C | 0.183 ± 0.006 | 0.242 ± 0.021 | 0.352 ± 0.024 | 0.251 ± 0.010 | 0.256 ± 0.009 | 0.242 ± 0.021 |
| | AxTAO | *0.087 ± 0.028* | **0.128 ± 0.040** | **0.421 ± 0.125** | **0.584 ± 0.083** | **0.462 ± 0.117** | **0.584 ± 0.083** |
| | DPDT | **0.078 ± 0.008** | *0.239 ± 0.005* | *1.501 ± 0.011* | **0.134 ± 0.008** | *0.627 ± 0.035* | *0.627 ± 0.035* |
| | SPLIT | 1.612 ± 0.771 | 0.761 ± 0.545 | 1.685 ± 0.718 | 1.898 ± 0.157 | 1.684 ± 0.100 | 1.898 ± 0.157 |
| | SGT-T | 0.894 ± 0.390 | 1.666 ± 0.538 | 2.890 ± 0.713 | 13.621 ± 10.497 | 4.850 ± 2.549 | 4.850 ± 2.549 |
| | SGT$_3$-T | 2.030 ± 0.609 | 2.338 ± 0.916 | 8.042 ± 6.568 | 10.688 ± 10.439 | 10.251 ± 10.054 | 10.688 ± 10.439 |

Table 7: Runtime (s ± std.) for all bivariate approaches separated into TDIDT and Non-Greedy approaches by dashed lines per dataset. Best approach is **bolded**, second-best is *italicized*.

| dataset | model | 2 | 3 | 4 | 5 | 6 | Best |
|---|---|---|---|---|---|---|---|
| adult | BiCART | 161.982 ± 4.528 | 230.208 ± 2.447 | 208.602 ± 2.611 | 164.421 ± 1.459 | 244.966 ± 6.894 | 244.966 ± 6.894 |
| | S$^2$GT-C | **0.914 ± 0.023** | **1.466 ± 0.018** | **1.785 ± 0.013** | **2.138 ± 0.251** | **2.883 ± 0.237** | **2.883 ± 0.237** |
| | S$^2$GT$_3$-C | *1.018 ± 0.011* | *2.766 ± 0.069* | *5.652 ± 0.187* | *9.798 ± 0.353* | *9.137 ± 0.356* | *9.137 ± 0.356* |
| | BiTAO | 519.138 ± 9.067 | 660.103 ± 10.453 | 835.272 ± 438.190 | 676.755 ± 254.419 | 715.373 ± 277.318 | 676.755 ± 254.419 |
| | S$^2$GT-T | **3.048 ± 0.475** | **5.817 ± 0.526** | **10.216 ± 0.996** | **8.596 ± 0.906** | **29.082 ± 19.674** | **8.596 ± 0.906** |
| | S$^2$GT$_3$-T | *8.479 ± 1.166* | *16.804 ± 3.316* | *28.323 ± 7.263* | *58.823 ± 7.955* | *227.868 ± 25.558* | *58.823 ± 7.955* |
| avila | BiCART | — | — | **3.605 ± 0.016** | **2.272 ± 0.004** | **2.646 ± 0.010** | **2.646 ± 0.010** |
| | S$^2$GT-C | — | — | *4.248 ± 0.131* | *5.380 ± 0.214* | *6.402 ± 0.171* | *6.402 ± 0.171* |
| | S$^2$GT$_3$-C | — | — | *9.139 ± 0.178* | *10.774 ± 0.165* | *9.608 ± 0.245* | *9.608 ± 0.245* |
| | BiTAO | — | — | **8.205 ± 0.404** | **13.076 ± 1.953** | *27.271 ± 4.852* | *27.271 ± 4.852* |
| | S$^2$GT-T | — | — | *16.503 ± 11.068* | *18.244 ± 2.260* | **23.580 ± 1.872** | **23.580 ± 1.872** |
| | S$^2$GT$_3$-T | — | — | 19.400 ± 0.481 | 22.246 ± 4.050 | 32.641 ± 0.356 | 32.641 ± 0.356 |
| bank | BiCART | **0.011 ± 0.000** | **0.015 ± 0.001** | **0.010 ± 0.001** | **0.011 ± 0.001** | **0.011 ± 0.001** | **0.010 ± 0.001** |
| | S$^2$GT-C | *0.132 ± 0.001* | *0.144 ± 0.012* | *0.149 ± 0.009* | *0.142 ± 0.011* | *0.141 ± 0.009* | *0.132 ± 0.001* |
| | S$^2$GT$_3$-C | 0.170 ± 0.008 | 0.212 ± 0.017 | 0.209 ± 0.021 | 0.213 ± 0.016 | 0.214 ± 0.020 | 0.170 ± 0.008 |
| | BiTAO | **0.437 ± 0.170** | *0.539 ± 0.056* | 1.560 ± 0.410 | 1.577 ± 0.412 | 1.538 ± 0.364 | 1.560 ± 0.410 |
| | S$^2$GT-T | 0.680 ± 0.034 | *0.546 ± 0.137* | **0.531 ± 0.176** | **0.515 ± 0.152** | **0.476 ± 0.159** | **0.546 ± 0.137** |
| | S$^2$GT$_3$-T | *0.635 ± 0.277* | 0.718 ± 0.202 | 0.860 ± 0.271 | 0.877 ± 0.259 | 1.106 ± 0.353 | *0.635 ± 0.277* |
| bean | BiCART | — | *5.241 ± 0.034* | 4.492 ± 0.016 | **5.400 ± 0.063** | **6.030 ± 0.085** | **6.030 ± 0.085** |
| | S$^2$GT-C | — | **4.264 ± 0.059** | **4.134 ± 0.057** | *5.789 ± 0.114* | 7.849 ± 0.426 | 7.849 ± 0.426 |
| | S$^2$GT$_3$-C | — | 6.099 ± 0.066 | 9.546 ± 0.254 | 9.039 ± 0.777 | *6.448 ± 0.122* | 9.546 ± 0.254 |
| | BiTAO | — | *13.466 ± 0.089* | *17.550 ± 3.502* | **21.262 ± 4.176** | *17.836 ± 5.087* | *17.836 ± 5.087* |
| | S$^2$GT-T | — | **10.078 ± 0.063** | **14.549 ± 2.278** | *35.814 ± 17.026* | 49.716 ± 23.694 | 49.716 ± 23.694 |
| | S$^2$GT$_3$-T | — | 25.607 ± 0.510 | 28.688 ± 5.987 | 45.855 ± 0.793 | *36.229 ± 4.994* | 45.855 ± 0.793 |
| bidding | BiCART | **0.122 ± 0.000** | **0.226 ± 0.004** | **0.087 ± 0.000** | **0.088 ± 0.000** | **0.087 ± 0.001** | **0.087 ± 0.000** |
| | S$^2$GT-C | *0.379 ± 0.062* | *0.420 ± 0.076* | *0.453 ± 0.089* | *0.444 ± 0.084* | *0.444 ± 0.085* | *0.420 ± 0.076* |
| | S$^2$GT$_3$-C | 0.520 ± 0.007 | 0.631 ± 0.043 | 0.643 ± 0.040 | 0.641 ± 0.041 | 0.634 ± 0.045 | 0.520 ± 0.007 |
| | BiTAO | **0.455 ± 0.155** | 2.854 ± 1.296 | **0.881 ± 0.103** | **0.968 ± 0.127** | **0.956 ± 0.122** | **0.881 ± 0.103** |
| | S$^2$GT-T | *3.742 ± 2.479* | *1.552 ± 0.201* | 1.886 ± 0.069 | 2.064 ± 0.305 | 2.151 ± 0.424 | *1.552 ± 0.201* |
| | S$^2$GT$_3$-T | 3.784 ± 2.417 | **1.336 ± 0.026** | *1.449 ± 0.033* | *1.363 ± 0.031* | *1.847 ± 0.070* | *1.336 ± 0.026* |
| electricity | BiCART | **1.710 ± 0.013** | **2.459 ± 0.015** | *4.716 ± 0.011* | **4.399 ± 0.060** | **4.468 ± 0.027** | **4.468 ± 0.027** |

| Dataset | Method | | | | | | |
|---|---|---|---|---|---|---|---|
| | S$^2$GT-C | *1.751 ± 0.020* | *3.068 ± 0.041* | **3.943 ± 0.022** | *6.214 ± 0.068* | *7.709 ± 0.056* | *7.709 ± 0.056* |
| | S$^2$GT$_3$-C | *2.077 ± 0.024* | *4.083 ± 0.066* | *6.865 ± 0.109* | *13.089 ± 0.404* | *16.725 ± 0.356* | *16.725 ± 0.356* |
| | BiTAO | **3.090 ± 0.013** | **5.217 ± 0.982** | **19.134 ± 2.517** | **25.496 ± 15.712** | *23.716 ± 5.873* | *23.716 ± 5.873* |
| | S$^2$GT-T | *4.900 ± 0.901* | *13.187 ± 2.958* | *26.192 ± 8.955* | *35.407 ± 5.905* | *64.266 ± 27.552* | *64.266 ± 27.552* |
| | S$^2$GT$_3$-T | *16.278 ± 1.949* | *29.913 ± 8.363* | *38.181 ± 0.606* | *58.046 ± 15.761* | *186.041 ± 81.833* | *186.041 ± 81.833* |
| eye-movements | BiCART | *2.343 ± 0.006* | *5.074 ± 0.073* | *6.529 ± 0.029* | **5.365 ± 0.003** | *6.213 ± 0.023* | *6.213 ± 0.023* |
| | S$^2$GT-C | **1.882 ± 0.068** | **3.410 ± 0.018** | **5.833 ± 0.091** | *9.208 ± 0.331* | *14.078 ± 0.496* | *14.078 ± 0.496* |
| | S$^2$GT$_3$-C | *2.973 ± 0.024* | *6.749 ± 0.222* | *14.320 ± 0.381* | *23.759 ± 3.123* | *41.232 ± 2.209* | *41.232 ± 2.209* |
| | BiTAO | **5.639 ± 1.477** | **10.443 ± 1.853** | **12.694 ± 3.756** | **25.351 ± 15.615** | **30.017 ± 7.258** | **30.017 ± 7.258** |
| | S$^2$GT-T | *5.723 ± 0.934* | *13.424 ± 2.150* | *20.808 ± 2.432* | *40.531 ± 6.476* | *70.415 ± 28.127* | *70.415 ± 28.127* |
| | S$^2$GT$_3$-T | *13.826 ± 1.436* | *31.005 ± 7.743* | *30.284 ± 4.121* | *55.390 ± 7.432* | *69.564 ± 6.342* | *31.005 ± 7.743* |
| eye-state | BiCART | *1.575 ± 0.018* | **2.290 ± 0.085** | **3.017 ± 0.008** | **2.397 ± 0.016** | **1.431 ± 0.008** | **1.431 ± 0.008** |
| | S$^2$GT-C | **1.428 ± 0.009** | *2.650 ± 0.068* | *4.042 ± 0.019* | *6.192 ± 0.074* | *5.843 ± 0.092* | *5.843 ± 0.092* |
| | S$^2$GT$_3$-C | *1.998 ± 0.009* | *3.895 ± 0.009* | *6.083 ± 0.386* | *7.614 ± 0.383* | *14.603 ± 0.709* | *7.614 ± 0.383* |
| | BiTAO | *5.084 ± 0.763* | **6.086 ± 0.920** | **15.055 ± 3.777** | **16.758 ± 3.696** | **17.236 ± 4.397** | **17.236 ± 4.397** |
| | S$^2$GT-T | **4.785 ± 1.786** | *10.767 ± 5.110* | *20.799 ± 3.613* | *25.682 ± 10.657* | *39.077 ± 5.903* | *39.077 ± 5.903* |
| | S$^2$GT$_3$-T | *10.356 ± 2.611* | *13.256 ± 0.249* | *36.587 ± 3.486* | *84.161 ± 64.111* | *83.416 ± 22.508* | *83.416 ± 22.508* |
| fault | BiCART | — | **0.671 ± 0.004** | **0.858 ± 0.020** | **1.056 ± 0.021** | **2.960 ± 0.041** | **2.960 ± 0.041** |
| | S$^2$GT-C | — | *1.612 ± 0.073* | *3.789 ± 0.147* | *4.264 ± 0.128* | *3.640 ± 0.253* | *3.640 ± 0.253* |
| | S$^2$GT$_3$-C | — | *2.670 ± 0.381* | *5.217 ± 0.624* | *4.927 ± 0.401* | *15.148 ± 1.188* | *4.927 ± 0.401* |
| | BiTAO | — | *27.657 ± 8.662* | *63.598 ± 13.320* | *37.251 ± 2.027* | **9.754 ± 3.397** | **9.754 ± 3.397** |
| | S$^2$GT-T | — | **6.341 ± 1.211** | **11.313 ± 7.714** | **11.403 ± 3.118** | *18.722 ± 1.558* | *11.403 ± 3.118* |
| | S$^2$GT$_3$-T | — | *9.007 ± 1.816* | **7.998 ± 0.319** | *17.878 ± 0.855* | *28.492 ± 5.414* | *17.878 ± 0.855* |
| gas-drift | BiCART | — | *306.141 ± 1.082* | *549.345 ± 1.295* | *595.498 ± 2.316* | *687.982 ± 21.935* | *687.982 ± 21.935* |
| | S$^2$GT-C | — | **63.441 ± 2.416** | **85.418 ± 5.980** | **89.618 ± 1.188** | **107.478 ± 2.900** | **107.478 ± 2.900** |
| | S$^2$GT$_3$-C | — | *69.479 ± 0.213* | *177.539 ± 4.145* | *100.374 ± 2.373* | *110.818 ± 1.290* | *110.818 ± 1.290* |
| | BiTAO | — | *383.562 ± 93.594* | *1327.892 ± 366.219* | *582.853 ± 87.410* | *1090.152 ± 121.334* | *1090.152 ± 121.334* |
| | S$^2$GT-T | — | **202.147 ± 53.643** | **192.221 ± 68.401** | **208.110 ± 44.917** | **186.536 ± 20.301** | **186.536 ± 20.301** |
| | S$^2$GT$_3$-T | — | *212.746 ± 133.915* | *202.622 ± 15.431* | *249.498 ± 18.637* | *253.412 ± 18.672* | *202.622 ± 15.431* |
| htru | BiCART | **0.465 ± 0.000** | **0.691 ± 0.008** | **1.050 ± 0.065** | **1.035 ± 0.115** | **1.175 ± 0.210** | **1.175 ± 0.210** |
| | S$^2$GT-C | *1.375 ± 0.019* | *1.614 ± 0.023* | *1.867 ± 0.009* | *1.838 ± 0.239* | *3.212 ± 0.266* | *1.867 ± 0.009* |
| | S$^2$GT$_3$-C | *1.706 ± 0.039* | *1.496 ± 0.091* | *2.098 ± 0.188* | *2.490 ± 0.329* | *2.681 ± 0.201* | *1.496 ± 0.091* |
| | BiTAO | **0.490 ± 0.004** | *4.274 ± 0.386* | *7.087 ± 2.078* | **3.572 ± 0.883** | *6.305 ± 0.846* | *6.305 ± 0.846* |
| | S$^2$GT-T | *1.559 ± 0.254* | **3.521 ± 0.623** | **4.992 ± 0.509** | *11.568 ± 2.664* | *12.452 ± 3.665* | **3.521 ± 0.623** |
| | S$^2$GT$_3$-T | *3.907 ± 1.259* | *4.842 ± 1.006* | *7.089 ± 1.676* | *9.252 ± 1.575* | *13.567 ± 1.836* | *9.252 ± 1.575* |
| magic | BiCART | **0.532 ± 0.003** | **0.761 ± 0.004** | **0.986 ± 0.004** | **1.214 ± 0.005** | **1.399 ± 0.007** | **1.214 ± 0.005** |
| | S$^2$GT-C | *1.222 ± 0.006* | *2.023 ± 0.016* | *2.579 ± 0.042* | *3.417 ± 0.036* | *5.149 ± 0.293* | *3.417 ± 0.036* |
| | S$^2$GT$_3$-C | *1.853 ± 0.015* | *2.336 ± 0.033* | *3.632 ± 0.243* | *5.462 ± 0.175* | *6.613 ± 0.503* | *3.632 ± 0.243* |
| | BiTAO | **2.312 ± 1.436** | *7.309 ± 1.476* | *5.492 ± 2.836* | *11.394 ± 1.136* | *12.020 ± 6.460* | *12.020 ± 6.460* |
| | S$^2$GT-T | *3.670 ± 0.175* | **6.565 ± 2.845** | *8.800 ± 1.447* | *19.018 ± 4.394* | *22.002 ± 2.200* | *19.018 ± 4.394* |
| | S$^2$GT$_3$-T | *6.552 ± 0.972* | *11.303 ± 3.233* | *28.370 ± 18.524* | *64.631 ± 32.421* | *49.323 ± 4.491* | *64.631 ± 32.421* |
| mini-boone | BiCART | *205.993 ± 1.017* | *283.306 ± 3.623* | *421.122 ± 6.973* | *585.503 ± 2.113* | *685.981 ± 1.666* | *685.981 ± 1.666* |
| | S$^2$GT-C | *87.626 ± 0.976* | **87.751 ± 0.520** | **105.152 ± 0.643** | **135.392 ± 4.667** | **196.851 ± 0.727** | **196.851 ± 0.727** |
| | S$^2$GT$_3$-C | **77.147 ± 0.087** | *131.214 ± 0.526* | *184.391 ± 2.667* | *182.661 ± 4.632* | *275.842 ± 1.822* | *275.842 ± 1.822* |
| | BiTAO | *513.002 ± 218.327* | *695.079 ± 137.696* | *1580.557 ± 159.385* | *746.081 ± 90.647* | *692.239 ± 108.287* | *692.239 ± 108.287* |
| | S$^2$GT-T | **255.640 ± 60.646** | **240.491 ± 36.959** | **249.069 ± 31.002** | **364.934 ± 26.289** | **254.313 ± 11.954** | **254.313 ± 11.954** |
| | S$^2$GT$_3$-T | *547.680 ± 454.771* | *674.206 ± 232.831* | *268.206 ± 40.527* | *375.451 ± 48.913* | *659.697 ± 88.394* | *659.697 ± 88.394* |
| mushroom | BiCART | *9.876 ± 0.035* | *11.360 ± 0.076* | *11.661 ± 0.285* | *11.900 ± 0.104* | *11.424 ± 0.047* | *11.360 ± 0.076* |
| | S$^2$GT-C | **0.450 ± 0.004** | **0.574 ± 0.058** | **0.574 ± 0.060** | **0.573 ± 0.060** | **0.567 ± 0.054** | **0.574 ± 0.058** |
| | S$^2$GT$_3$-C | *0.674 ± 0.002* | *0.677 ± 0.008* | *0.676 ± 0.010* | *0.671 ± 0.003* | *0.664 ± 0.007* | *0.674 ± 0.002* |
| | BiTAO | *23.650 ± 0.082* | *32.016 ± 0.068* | *31.680 ± 0.114* | *31.644 ± 0.217* | *30.900 ± 0.695* | *32.016 ± 0.068* |
| | S$^2$GT-T | **0.903 ± 0.051** | **1.237 ± 0.153** | **1.257 ± 0.165** | **1.254 ± 0.178** | **1.228 ± 0.168** | **1.237 ± 0.153** |
| | S$^2$GT$_3$-T | *1.725 ± 0.026* | *1.708 ± 0.009* | *1.700 ± 0.012* | *1.717 ± 0.039* | *1.708 ± 0.034* | *1.725 ± 0.026* |
| occupancy | BiCART | **0.165 ± 0.002** | **0.236 ± 0.002** | **0.242 ± 0.027** | **0.396 ± 0.011** | **0.181 ± 0.002** | **0.181 ± 0.002** |
| | S$^2$GT-C | *0.653 ± 0.025* | *0.939 ± 0.061* | *0.744 ± 0.021* | *0.590 ± 0.050* | *0.585 ± 0.052* | *0.939 ± 0.061* |
| | S$^2$GT$_3$-C | *0.739 ± 0.014* | *0.680 ± 0.002* | *0.991 ± 0.057* | *1.741 ± 0.049* | *1.879 ± 0.047* | *0.739 ± 0.014* |
| | BiTAO | **0.358 ± 0.007** | **0.592 ± 0.279** | **0.966 ± 0.297** | *2.024 ± 1.027* | *2.756 ± 1.003* | *2.756 ± 1.003* |
| | S$^2$GT-T | *2.334 ± 0.224* | *3.354 ± 1.055* | *10.974 ± 7.008* | *8.160 ± 1.116* | *8.196 ± 1.756* | *8.196 ± 1.756* |
| | S$^2$GT$_3$-T | *3.550 ± 0.639* | *3.333 ± 1.181* | *4.591 ± 1.160* | *3.789 ± 1.850* | *3.924 ± 2.057* | *3.550 ± 0.639* |
| page | BiCART | — | **0.136 ± 0.001** | **0.265 ± 0.002** | **0.568 ± 0.048** | **0.624 ± 0.088** | **0.624 ± 0.088** |
| | S$^2$GT-C | — | *0.647 ± 0.005* | *1.219 ± 0.055* | *1.820 ± 0.025* | *1.951 ± 0.072* | *1.219 ± 0.055* |
| | S$^2$GT$_3$-C | — | *1.553 ± 0.106* | *0.974 ± 0.004* | *1.120 ± 0.110* | *1.441 ± 0.129* | *0.974 ± 0.004* |
| | BiTAO | — | **2.071 ± 0.330** | **2.233 ± 0.756** | **2.252 ± 0.853** | *3.958 ± 2.572* | *3.958 ± 2.572* |
| | S$^2$GT-T | — | *3.085 ± 0.805* | *8.576 ± 8.214* | *13.182 ± 11.022* | *6.672 ± 1.158* | **3.085 ± 0.805** |
| | S$^2$GT$_3$-T | — | *6.055 ± 1.312* | *10.564 ± 2.250* | *10.963 ± 1.690* | *15.098 ± 2.583* | *10.564 ± 2.250* |

| | | | | | | |
|---|---|---|---|---|---|---|
| pendigits | BiCART | — | — | **3.703 ± 0.029** | **4.409 ± 0.038** | **4.979 ± 0.132** | **4.979 ± 0.132** |
| | $S^2$GT-C | — | — | *5.447 ± 0.137* | *5.681 ± 0.077* | *5.999 ± 0.100* | *5.999 ± 0.100* |
| | $S^2$GT$_3$-C | — | — | 13.424 ± 0.194 | 11.251 ± 0.546 | 16.132 ± 0.661 | 11.251 ± 0.546 |
| | BiTAO | — | — | **16.041 ± 1.840** | **27.429 ± 2.592** | **39.368 ± 6.442** | **39.368 ± 6.442** |
| | $S^2$GT-T | — | — | *21.442 ± 11.148* | **20.195 ± 3.336** | *41.947 ± 12.994* | *41.947 ± 12.994* |
| | $S^2$GT$_3$-T | — | — | 31.893 ± 6.406 | 163.337 ± 7.436 | 56.912 ± 9.267 | 56.912 ± 9.267 |
| raisin | BiCART | **0.019 ± 0.000** | **0.037 ± 0.001** | **0.041 ± 0.001** | **0.030 ± 0.001** | **0.041 ± 0.001** | **0.037 ± 0.001** |
| | $S^2$GT-C | *0.180 ± 0.001* | *0.329 ± 0.003* | 0.465 ± 0.014 | 0.520 ± 0.028 | 0.560 ± 0.063 | 0.329 ± 0.003 |
| | $S^2$GT$_3$-C | 0.269 ± 0.001 | 0.388 ± 0.064 | *0.391 ± 0.061* | *0.382 ± 0.063* | *0.381 ± 0.055* | *0.269 ± 0.001* |
| | BiTAO | **0.285 ± 0.219** | **0.635 ± 0.211** | **0.471 ± 0.088** | **0.480 ± 0.092** | 1.322 ± 0.369 | *0.635 ± 0.211* |
| | $S^2$GT-T | *0.509 ± 0.154* | 1.571 ± 0.680 | *0.992 ± 0.376* | 1.216 ± 0.535 | **1.137 ± 0.567** | **0.509 ± 0.154** |
| | $S^2$GT$_3$-T | 0.727 ± 0.285 | *0.932 ± 0.477* | 1.159 ± 0.605 | 0.948 ± 0.491 | *1.317 ± 0.714* | 0.727 ± 0.285 |
| rice | BiCART | **0.117 ± 0.000** | **0.168 ± 0.001** | **0.213 ± 0.006** | **0.236 ± 0.004** | **0.169 ± 0.002** | **0.236 ± 0.004** |
| | $S^2$GT-C | *0.317 ± 0.006* | *0.468 ± 0.008* | *0.715 ± 0.060* | 1.082 ± 0.077 | 1.335 ± 0.025 | *0.317 ± 0.006* |
| | $S^2$GT$_3$-C | 0.359 ± 0.002 | 0.804 ± 0.007 | 0.815 ± 0.074 | *0.839 ± 0.083* | *0.841 ± 0.079* | 0.359 ± 0.002 |
| | BiTAO | **0.391 ± 0.001** | **0.595 ± 0.363** | 1.089 ± 0.348 | **1.103 ± 0.452** | **0.897 ± 0.303** | **1.103 ± 0.452** |
| | $S^2$GT-T | *0.728 ± 0.013* | *1.912 ± 0.823* | 5.103 ± 3.741 | 4.694 ± 1.198 | 2.002 ± 0.473 | *1.912 ± 0.823* |
| | $S^2$GT$_3$-T | 1.425 ± 0.322 | 3.249 ± 3.737 | *3.242 ± 3.695* | *3.186 ± 3.657* | 3.047 ± 3.429 | 3.249 ± 3.737 |
| room | BiCART | **0.245 ± 0.002** | **0.329 ± 0.001** | **0.354 ± 0.007** | **0.371 ± 0.006** | **1.095 ± 0.039** | **0.354 ± 0.007** |
| | $S^2$GT-C | *0.529 ± 0.010* | *1.018 ± 0.131* | *1.145 ± 0.033* | 1.299 ± 0.073 | *1.278 ± 0.065* | *1.145 ± 0.033* |
| | $S^2$GT$_3$-C | 0.802 ± 0.016 | 1.259 ± 0.076 | 1.284 ± 0.063 | *1.274 ± 0.061* | 1.286 ± 0.068 | 1.259 ± 0.076 |
| | BiTAO | 4.212 ± 0.520 | *2.872 ± 0.386* | **1.356 ± 0.090** | *5.522 ± 0.832* | **3.008 ± 1.874** | 5.522 ± 0.832 |
| | $S^2$GT-T | *2.757 ± 0.664* | **2.666 ± 0.581** | *3.444 ± 0.002* | **4.148 ± 1.386** | *4.337 ± 0.795* | *3.444 ± 0.002* |
| | $S^2$GT$_3$-T | **2.726 ± 0.176** | 2.930 ± 0.463 | 6.014 ± 4.492 | 9.987 ± 7.450 | 13.029 ± 9.741 | **2.726 ± 0.176** |
| segment | BiCART | — | **0.713 ± 0.018** | **0.856 ± 0.008** | **0.928 ± 0.034** | **0.986 ± 0.050** | **0.986 ± 0.050** |
| | $S^2$GT-C | — | *1.404 ± 0.020* | *1.555 ± 0.106* | 2.007 ± 0.044 | 2.454 ± 0.174 | 2.454 ± 0.174 |
| | $S^2$GT$_3$-C | — | 2.722 ± 0.065 | 3.512 ± 0.235 | 3.526 ± 0.332 | 3.515 ± 0.317 | 3.512 ± 0.235 |
| | BiTAO | — | *3.508 ± 1.662* | **2.871 ± 1.091** | **4.486 ± 1.403** | **5.786 ± 0.956** | **5.786 ± 0.956** |
| | $S^2$GT-T | — | **3.402 ± 0.607** | *4.658 ± 0.903* | 6.702 ± 1.683 | 6.846 ± 0.676 | 6.702 ± 1.683 |
| | $S^2$GT$_3$-T | — | 6.858 ± 1.575 | 5.228 ± 0.796 | *6.383 ± 1.162* | 8.472 ± 1.177 | *6.383 ± 1.162* |
| skin | BiCART | **0.550 ± 0.004** | **0.778 ± 0.010** | **0.837 ± 0.012** | **0.300 ± 0.005** | **0.840 ± 0.010** | **0.840 ± 0.010** |
| | $S^2$GT-C | 1.529 ± 0.086 | *1.572 ± 0.126* | 2.164 ± 0.047 | 2.281 ± 0.059 | 2.371 ± 0.097 | 2.371 ± 0.097 |
| | $S^2$GT$_3$-C | *1.244 ± 0.173* | 1.668 ± 0.109 | *1.843 ± 0.028* | *1.975 ± 0.152* | *1.716 ± 0.089* | *1.716 ± 0.089* |
| | BiTAO | **1.343 ± 0.007** | **2.537 ± 0.034** | **4.490 ± 0.230** | **4.175 ± 1.884** | *24.620 ± 4.906* | 24.620 ± 4.906 |
| | $S^2$GT-T | *3.948 ± 0.217* | 10.320 ± 4.136 | 6.232 ± 0.635 | *7.273 ± 0.468* | **6.485 ± 1.392** | *7.273 ± 0.468* |
| | $S^2$GT$_3$-T | 4.443 ± 0.053 | *6.694 ± 0.465* | *5.812 ± 1.213* | 39.704 ± 31.111 | 54.114 ± 41.059 | **5.812 ± 1.213** |
| wilt | BiCART | **0.061 ± 0.001** | **0.064 ± 0.001** | **0.066 ± 0.001** | **0.066 ± 0.001** | **0.079 ± 0.002** | **0.079 ± 0.002** |
| | $S^2$GT-C | *0.280 ± 0.003* | *0.366 ± 0.019* | *0.479 ± 0.028* | *0.401 ± 0.027* | *0.396 ± 0.034* | *0.280 ± 0.003* |
| | $S^2$GT$_3$-C | 0.313 ± 0.007 | 0.480 ± 0.020 | 0.595 ± 0.057 | 0.672 ± 0.038 | 0.680 ± 0.035 | 0.480 ± 0.020 |
| | BiTAO | **0.413 ± 0.122** | **0.432 ± 0.164** | **0.376 ± 0.109** | *0.764 ± 0.213* | **0.768 ± 0.209** | **0.413 ± 0.122** |
| | $S^2$GT-T | *0.781 ± 0.193* | 0.867 ± 0.197 | 1.424 ± 0.627 | 1.501 ± 0.732 | 1.373 ± 0.615 | 0.781 ± 0.193 |
| | $S^2$GT$_3$-T | 0.825 ± 0.293 | *0.678 ± 0.174* | 0.940 ± 0.358 | **0.762 ± 0.272** | *1.056 ± 0.434* | *0.678 ± 0.174* |

### G.2.2 Optimization Procedure Ablation

In this section, we examine our bin-to-branch optimization procedure. Here, we provide two ablations: removing the coordinate descent procedure and only replying on Weighted K-Means to assign bins to branches (denoted as "No CD") and replacing the Weighted K-Means initialization with a random initialization (denoted as "No WKM"). For these experiments, we work with a subset of our datasets - Electricity, Eye-Movements, and Eye-State. We run hyperparameter tuning for both ablations and choose the model with the lowest validation loss for each depth. We report the average test accuracy for the ablated TDIDT approaches in Table 8 and provide the runtimes for these approaches in Table 9. In these tables, "All" denotes using the full optimization procedure. In general, we observe that both ablations consistently achieve lower test accuracy than the full models across most depths and tested datasets, especially at depths above two, with a minimal sacrifice in runtime.

Table 8: Test Accuracy (% ± std.) results for the optimization ablation analysis. Best approach for each model and dataset is **bolded**.

| | Model | 2 | 3 | 4 | 5 | 6 | Overall |
|---|---|---|---|---|---|---|---|
| | | | | **ShapeCART** | | | |
| electricity | All | **75.5±0.1** | **78.5±0.4** | **81.3±0.8** | **84.4±0.1** | **84.8±0.6** | **84.8±0.6** |
| | No CD | 74.9±0.2 | 76.5±0.3 | 79.6±0.5 | 83.7±1.2 | 83.4±0.2 | 83.4±0.2 |
| | No WKM | 74.9±0.2 | 76.5±0.3 | 79.4±0.1 | 83.2±0.4 | 83.3±0.5 | 83.3±0.5 |
| eye-movements | All | **48.1±2.1** | **53±0.7** | **54.9±1.2** | 53.9±0.7 | **57.4±1.2** | **57.4±1.2** |
| | No CD | 47.1±3.3 | 52.5±1.3 | 52.4±2.7 | 47.4±1.6 | 51±0.5 | 51±0.5 |
| | No WKM | 47.3±1.2 | 51.8±1 | 51.5±0.7 | 47.2±0.8 | 50.1±1.3 | 50.1±1.3 |
| eye-state | All | 64.3±0.9 | **69±0.2** | 70.5±1.2 | 72.4±1.3 | 74.5±1.1 | 74.5±1.1 |
| | No CD | 62.5±0.3 | 68.6±0.9 | 70.3±1 | 71.9±0.9 | 73.7±0.7 | 73.7±0.7 |
| | No WKM | **64.4±1.2** | 68.5±1 | **70.5±1.2** | **72.4±1.3** | **74.5±1.1** | **74.5±1.1** |
| | | | | **ShapeCART$_3$** | | | |
| electricity | All | **76.5±0.3** | **80.6±0.6** | **84±1.2** | **86±0.4** | 87.8±1 | 87.8±1 |
| | No CD | 73.5±1.2 | 78.5±0.3 | 82.3±0.1 | 85.7±0.3 | 88±0.5 | 88±0.5 |
| | No WKM | 73±0.4 | 78.1±0.3 | 82±0.4 | 84.8±0.5 | **88.3±0.3** | **88.3±0.3** |
| eye-movements | All | **47.1±0.5** | 54.4±0.8 | 56±0.3 | **57.5±1.6** | **60.9±2.6** | **60.9±2.6** |
| | No CD | 46±0.5 | 54.8±1.4 | 56.1±1.4 | 54.4±1.7 | 56.5±3 | 56.5±3 |
| | No WKM | 45.6±0.4 | **55.3±1.9** | **56.2±2.3** | 56±1.2 | 57.2±1.9 | 57.2±1.9 |
| eye-state | All | **66.1±0.7** | **70.8±1.4** | 73.3±1 | 76.8±1.3 | 77.5±0.5 | 77.5±0.5 |
| | No CD | 64.6±0.3 | 69.3±1 | **73.7±1.5** | **78.4±0.1** | 76.7±0.5 | 76.7±0.5 |
| | No WKM | 65.1±0.8 | 70.7±1.1 | 73±1 | 77.5±1.3 | **77.7±0.2** | **77.7±0.2** |
| | | | | **Shape$^2$CART** | | | |
| electricity | All | **80.6±0.6** | **82±0.6** | **83.9±0.3** | **86.5±1.2** | **87.5±0.5** | **87.5±0.5** |
| | No CD | 78.3±0.4 | 77.4±0.7 | 79.8±0.5 | 84.8±0.5 | 86.3±0.3 | 86.3±0.3 |
| | No WKM | 78.5±0.6 | 77.4±0.2 | 79.8±1 | 84.2±1 | 85.7±0.4 | 85.7±0.4 |
| eye-movements | All | **57.8±1.4** | **60±2** | **61.7±3.2** | **62.6±3.3** | **64.8±3.5** | **64.8±3.5** |
| | No CD | 57.7±3.2 | 59±1.5 | 58.2±3.1 | 58.4±2.9 | 60.5±0.5 | 60.5±0.5 |
| | No WKM | 56.4±2 | 57.5±1.6 | 58.7±0.7 | 57.1±0.2 | 61.1±1 | 61.1±1 |
| eye-state | All | **71.7±1.1** | **72.8±0.7** | **74.4±2.1** | **76.1±2** | 80.2±0.3 | 80.2±0.3 |
| | No CD | 71.5±0.5 | 72.1±1.3 | 71.9±0.9 | 74.5±0.9 | **80.3±0.4** | **80.3±0.4** |
| | No WKM | 71.2±0.3 | 71.8±1.4 | 71.7±1.2 | 73.4±1.8 | 80.2±1.2 | 80.2±1.2 |
| | | | | **Shape$^2$CART$_3$** | | | |
| electricity | All | **81.8±0.2** | **84.5±0.3** | **86.6±0.8** | **87.3±1** | 88.8±0.2 | 88.8±0.2 |
| | No CD | 79.8±0.5 | 81.8±0.3 | 85±0.6 | 86±0.3 | **89±0.3** | 89±0.3 |
| | No WKM | 80±0.4 | 81.2±0.3 | 85.1±0.6 | 85.8±0.9 | 89.1±0.5 | **89.1±0.5** |
| eye-movements | All | **59.6±2.4** | **63.6±1.4** | **66.2±1.9** | **67.5±3** | **63.9±2.4** | **63.9±2.4** |
| | No CD | 55.1±1 | 58±0 | 58±1.7 | 57.4±3.4 | 51.9±1.8 | 51.9±1.8 |
| | No WKM | 56.6±1.1 | 58±2 | 60.3±3 | 62.7±6.3 | 55.4±3.6 | 55.4±3.6 |
| eye-state | All | 73.1±1.1 | **75.9±0.4** | 79.3±1.1 | 81.1±0.6 | **80.6±1.9** | 81.1±0.6 |
| | No CD | **74±0.8** | 75.4±1.8 | **80.2±0.2** | 81.5±0.4 | 80.3±1.1 | **81.5±0.4** |
| | No WKM | 73.3±1.1 | 75.3±2 | 79±3 | **81.7±6.3** | 78.8±3.6 | 78.8±3.6 |

Table 9: Runtime (s. $\pm$ std.) results for the optimization ablation analysis. Best approach for each model and dataset is **bolded**.

| | Model | 2 | 3 | 4 | 5 | 6 | Overall |
|---|---|---|---|---|---|---|---|
| | | | | ShapeCART | | | |
| electricity | All | 0.396±0.002 | **0.438±0.006** | 0.88±0.007 | 1.356±0.011 | 2.145±0.177 | 2.145±0.177 |
| | No CD | **0.283±0.018** | 0.439±0.017 | **0.692±0.06** | 0.883±0.018 | **1.385±0.097** | **1.385±0.097** |
| | No WKM | 0.294±0.011 | 0.481±0.009 | 0.722±0.016 | **0.809±0.012** | 1.463±0.043 | 1.463±0.043 |
| eye-movements | All | 0.464±0.009 | 0.91±0.01 | 1.54±0.02 | 2.501±0.018 | 3.879±0.329 | 3.879±0.329 |
| | No CD | **0.277±0.011** | **0.48±0.018** | **0.846±0.033** | **1.421±0.046** | **2.382±0.473** | **2.382±0.473** |
| | No WKM | 0.31±0.012 | 0.622±0.022 | 1.576±0.159 | 1.847±0.039 | 2.693±0.2 | 2.693±0.2 |
| eye-state | All | 0.339±0.003 | 0.505±0.002 | 0.678±0.005 | 1.171±0.005 | 2.087±0.1 | 2.087±0.1 |
| | No CD | **0.165±0.011** | 0.292±0.007 | 0.457±0.029 | 0.88±0.094 | 1.306±0.089 | 1.306±0.089 |
| | No WKM | 0.172±0.006 | **0.238±0.034** | **0.283±0.008** | **0.501±0.038** | **0.834±0.039** | **0.834±0.039** |
| | | | | ShapeCART$_3$ | | | |
| electricity | All | 0.646±0.014 | 1.138±0.012 | 2.301±0.081 | 4.603±0.381 | 7.767±0.918 | 7.767±0.918 |
| | No CD | **0.318±0.014** | **0.546±0.014** | **1.087±0.054** | **1.992±0.045** | 3.01±0.35 | 3.01±0.35 |
| | No WKM | 0.438±0.004 | 1.118±0.052 | 1.903±0.068 | 2.098±0.12 | **2.688±0.02** | **2.688±0.02** |
| eye-movements | All | 0.661±0.015 | 2.216±0.179 | 4.876±0.325 | 6.152±0.443 | 12.47±0.314 | 12.47±0.314 |
| | No CD | **0.352±0.027** | **0.656±0.105** | **1.201±0.03** | **1.749±0.193** | **2.762±0.078** | **2.762±0.078** |
| | No WKM | 0.629±0.07 | 1.564±0.264 | 3.672±0.31 | 3.811±0.331 | 4.829±0.155 | 4.829±0.155 |
| eye-state | All | 0.574±0.007 | 1.063±0.008 | 2.511±0.133 | 4.777±0.111 | 6.192±0.148 | 6.192±0.148 |
| | No CD | 0.192±0.008 | 0.48±0.044 | 1.071±0.144 | 1.975±0.048 | 4.265±0.132 | 4.265±0.132 |
| | No WKM | **0.136±0.007** | **0.478±0.014** | **0.59±0.029** | **1.167±0.073** | **1.86±0.154** | **1.86±0.154** |
| | | | | Shape$^2$CART | | | |
| electricity | All | 1.751±0.02 | 3.068±0.041 | 3.943±0.022 | 6.214±0.068 | 7.709±0.056 | 7.709±0.056 |
| | No CD | **1.585±0.005** | 2.759±0.052 | **3.713±0.17** | 4.962±0.154 | **5.619±0.056** | **5.619±0.056** |
| | No WKM | 1.885±0.007 | **2.36±0.004** | 3.816±0.123 | **4.508±0.047** | 5.876±0.025 | 5.876±0.025 |
| eye-movements | All | 1.882±0.068 | 3.41±0.018 | 5.833±0.091 | 9.208±0.331 | 14.078±0.496 | 14.078±0.496 |
| | No CD | **1.553±0.124** | 2.845±0.055 | 5.288±0.141 | **7.391±0.063** | 9.512±0.076 | 9.512±0.076 |
| | No WKM | 1.698±0.046 | **3.278±0.165** | **4.849±0.063** | 9.436±0.206 | **9.116±0.577** | **9.116±0.577** |
| eye-state | All | 1.428±0.009 | 2.65±0.068 | 4.042±0.019 | 6.192±0.074 | 5.843±0.092 | 5.843±0.092 |
| | No CD | 1.301±0.05 | **1.454±0.014** | 2.261±0.017 | 3.397±0.191 | 4.196±0.163 | 4.196±0.163 |
| | No WKM | **1.048±0.022** | 1.762±0.029 | **2.223±0.002** | **2.62±0.08** | **3.191±0.099** | **3.191±0.099** |
| | | | | Shape$^2$CART$_3$ | | | |
| electricity | All | **2.077±0.024** | 4.083±0.066 | 6.865±0.109 | 13.089±0.404 | 16.725±0.356 | 16.725±0.356 |
| | No CD | 2.274±0.035 | **3.113±0.104** | **4.469±0.124** | **5.756±0.742** | **8.358±0.733** | **5.756±0.742** |
| | No WKM | 2.253±0.035 | 4.182±0.025 | 5.976±0.013 | 6.594±0.101 | 8.824±0.615 | 8.824±0.615 |
| eye-movements | All | 2.973±0.024 | 6.749±0.222 | 14.32±0.381 | 23.759±3.123 | 41.232±2.209 | 41.232±2.209 |
| | No CD | 2.533±0.071 | **4.393±0.277** | **7.732±0.387** | **9.818±0.516** | **13.094±0.479** | **9.818±0.516** |
| | No WKM | **2.344±0.071** | 6.604±0.165 | 16.5±1.009 | 23.284±0.671 | 21.673±1.054 | 21.673±1.054 |
| eye-state | All | 1.998±0.009 | 3.895±0.009 | 6.083±0.386 | 7.614±0.383 | 14.603±0.709 | 7.614±0.383 |
| | No CD | **1.077±0.008** | **1.994±0.008** | 3.078±0.041 | 4.739±0.317 | 9.667±0.879 | **4.739±0.317** |
| | No WKM | 1.386±0.004 | 2.247±0.017 | **2.959±0.05** | **3.863±0.527** | **6.192±0.195** | 6.192±0.195 |

### G.2.3 Pairwise Selection Heuristic Ablation

In this section, we examine the benefits of utilizing our pairwise selection heuristic. To do this, we do not score pairs of features and instead allow Shape$^2$CART and Shape$^2$CART$_3$ to construct all bivariate shape functions at each node. Like in the previous ablation, we work with a subset of our datasets and run hyperparameter tuning to choose the model with the highest validation accuracy at each depth. We report the average test accuracy in 10 and training runtime in 11.

We observe that utilizing our heuristic has significant benefits in improving the training runtime for both Shape$^2$CART and Shape$^2$CART$_3$ across all datasets and depth. This improvement is very apparent on the eye-movements dataset where Shape$^2$CART and Shape$^2$CART$_3$ have an $\sim 5\times$ and $\sim 6\times$ reduction, respectively, in training runtime at depth 6. When looking at test accuracy, we observe that allowing for all pairs does benefit test accuracy. Interestingly, the accuracy of Shape$^2$CART$_3$ benefits from limiting pairs on most depths on the eye-state dataset.

Table 10: Test accuracy results (% ± std.) for the pairwise heuristic ablation.

| Dataset | Model | 2 | 3 | 4 | 5 | 6 | Overall |
|---|---|---|---|---|---|---|---|
| | | **Shape$^2$CART** | | | | | |
| electricity | Heur. | $80.6 \pm 0.6$ | $82 \pm 0.6$ | $83.9 \pm 0.3$ | $86.5 \pm 1.2$ | $87.5 \pm 0.5$ | $87.5 \pm 0.5$ |
| | No Heur. | $\mathbf{83.1 \pm 0.8}$ | $\mathbf{85.8 \pm 0.2}$ | $\mathbf{86.3 \pm 0.3}$ | $\mathbf{87 \pm 0.1}$ | $\mathbf{88.1 \pm 1}$ | $\mathbf{88.1 \pm 1}$ |
| eye-movements | Heur. | $57.8 \pm 1.4$ | $60 \pm 2$ | $61.7 \pm 3.2$ | $62.6 \pm 3.3$ | $64.8 \pm 3.5$ | $64.8 \pm 3.5$ |
| | No Heur. | $\mathbf{59.3 \pm 1.2}$ | $\mathbf{63.6 \pm 2.5}$ | $\mathbf{68.4 \pm 1.6}$ | $\mathbf{75.2 \pm 3}$ | $\mathbf{77.8 \pm 2.7}$ | $\mathbf{77.8 \pm 2.7}$ |
| eye-state | Heur. | $\mathbf{71.7 \pm 1.1}$ | $72.8 \pm 0.7$ | $74.4 \pm 2.1$ | $76.1 \pm 2$ | $80.2 \pm 0.3$ | $80.2 \pm 0.3$ |
| | No Heur. | $70.8 \pm 0.8$ | $\mathbf{73.7 \pm 0.7}$ | $\mathbf{76.1 \pm 1}$ | $\mathbf{77 \pm 2.1}$ | $\mathbf{81 \pm 1}$ | $\mathbf{81 \pm 1}$ |
| | | **Shape$^2$CART$_3$** | | | | | |
| electricity | Heur. | $81.8 \pm 0.2$ | $84.5 \pm 0.3$ | $86.6 \pm 0.8$ | $87.3 \pm 1$ | $\mathbf{88.8 \pm 20}$ | $\mathbf{88.8 \pm 20}$ |
| | No Heur. | $\mathbf{84.7 \pm 0.8}$ | $\mathbf{88.9 \pm 0.2}$ | $\mathbf{89 \pm 1}$ | $\mathbf{89 \pm 0.1}$ | $88.3 \pm 0.1$ | $88.9 \pm 0.2$ |
| eye-movements | Heur. | $59.6 \pm 2.4$ | $63.6 \pm 1.4$ | $66.2 \pm 1.9$ | $67.5 \pm 3$ | $63.9 \pm 2.4$ | $63.9 \pm 2.4$ |
| | No Heur. | $\mathbf{62 \pm 1.2}$ | $\mathbf{69.9 \pm 2.5}$ | $\mathbf{78.6 \pm 1.6}$ | $\mathbf{79.7 \pm 3}$ | $\mathbf{74.9 \pm 2.7}$ | $\mathbf{74.9 \pm 2.7}$ |
| eye-state | Heur. | $\mathbf{73.1 \pm 1.1}$ | $75.9 \pm 0.4$ | $\mathbf{79.3 \pm 1.1}$ | $\mathbf{81.1 \pm 0.6}$ | $80.6 \pm 1.9$ | $\mathbf{81.1 \pm 0.6}$ |
| | No Heur. | $73 \pm 0.8$ | $\mathbf{76.3 \pm 0.7}$ | $77.2 \pm 0.1$ | $78.9 \pm 2.1$ | $76.1 \pm 1$ | $78.9 \pm 2.1$ |

Table 11: Runtime results (s. ± std.) for the pairwise heuristic ablation.

| Dataset | Model | 2 | 3 | 4 | 5 | 6 | Overall |
|---|---|---|---|---|---|---|---|
| | | **Shape$^2$CART** | | | | | |
| electricity | Heur. | $\mathbf{1.751 \pm 0.02}$ | $\mathbf{3.068 \pm 0.041}$ | $\mathbf{3.943 \pm 0.022}$ | $\mathbf{6.214 \pm 0.068}$ | $\mathbf{7.709 \pm 0.056}$ | $\mathbf{7.709 \pm 0.056}$ |
| | No Heur. | $3.989 \pm 0.037$ | $7.202 \pm 0.123$ | $9.121 \pm 0.142$ | $13.878 \pm 0.319$ | $16.101 \pm 0.187$ | $16.101 \pm 0.187$ |
| eye-movements | Heur. | $\mathbf{1.882 \pm 0.068}$ | $\mathbf{3.41 \pm 0.018}$ | $\mathbf{5.833 \pm 0.091}$ | $\mathbf{9.208 \pm 0.331}$ | $\mathbf{14.078 \pm 0.496}$ | $\mathbf{14.078 \pm 0.496}$ |
| | No Heur. | $16.716 \pm 0.204$ | $28.674 \pm 0.132$ | $41.5 \pm 0.361$ | $75.599 \pm 4.085$ | $77.686 \pm 4.074$ | $77.686 \pm 4.074$ |
| eye-state | Heur. | $\mathbf{1.428 \pm 0.009}$ | $\mathbf{2.65 \pm 0.068}$ | $\mathbf{4.042 \pm 0.019}$ | $\mathbf{6.192 \pm 0.074}$ | $\mathbf{5.843 \pm 0.092}$ | $\mathbf{5.843 \pm 0.092}$ |
| | No Heur. | $8.321 \pm 0.127$ | $9.455 \pm 0.201$ | $11.765 \pm 0.312$ | $13.482 \pm 0.483$ | $14.123 \pm 0.336$ | $14.123 \pm 0.336$ |
| | | **Shape$^2$CART$_3$** | | | | | |
| electricity | Heur. | $\mathbf{2.077 \pm 0.024}$ | $\mathbf{4.083 \pm 0.066}$ | $\mathbf{6.865 \pm 0.109}$ | $\mathbf{13.089 \pm 0.404}$ | $\mathbf{16.725 \pm 0.356}$ | $16.725 \pm 0.356$ |
| | No Heur. | $5.084 \pm 0.037$ | $11.164 \pm 0.123$ | $18.321 \pm 0.101$ | $26.528 \pm 0.319$ | $42.123 \pm 0.331$ | $\mathbf{11.164 \pm 0.123}$ |
| eye-movements | Heur. | $\mathbf{2.973 \pm 0.024}$ | $\mathbf{6.749 \pm 0.222}$ | $\mathbf{14.32 \pm 0.381}$ | $\mathbf{23.759 \pm 3.123}$ | $\mathbf{41.232 \pm 2.209}$ | $\mathbf{41.232 \pm 2.209}$ |
| | No Heur. | $30.616 \pm 0.204$ | $56.75 \pm 0.132$ | $155.597 \pm 0.361$ | $201.283 \pm 4.085$ | $224.403 \pm 4.074$ | $224.403 \pm 4.074$ |
| eye-state | Heur. | $\mathbf{1.998 \pm 0.009}$ | $\mathbf{3.895 \pm 0.009}$ | $\mathbf{6.083 \pm 0.386}$ | $\mathbf{7.614 \pm 0.383}$ | $\mathbf{14.603 \pm 0.709}$ | $\mathbf{7.614 \pm 0.383}$ |
| | No Heur. | $7.594 \pm 0.127$ | $9.84 \pm 0.201$ | $17.761 \pm 0.413$ | $21.993 \pm 0.483$ | $33.826 \pm 0.651$ | $21.993 \pm 0.483$ |

# H Visualized Trees

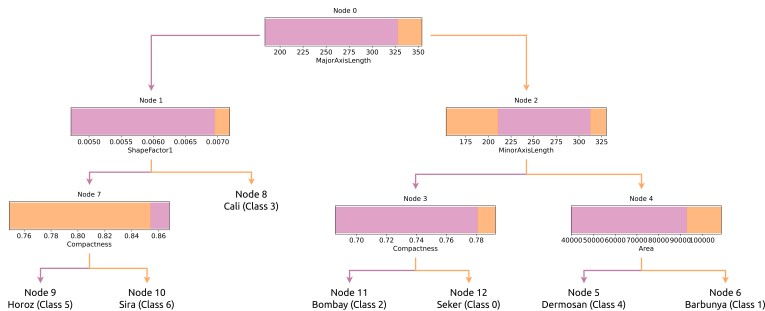

Figure H.5: ShapeCART (Depth 3) trained on Bean

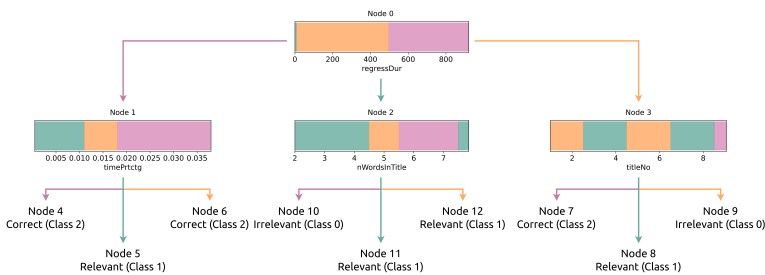

Figure H.6: ShapeCART$_3$ (Depth 2) trained on Eye-Movements

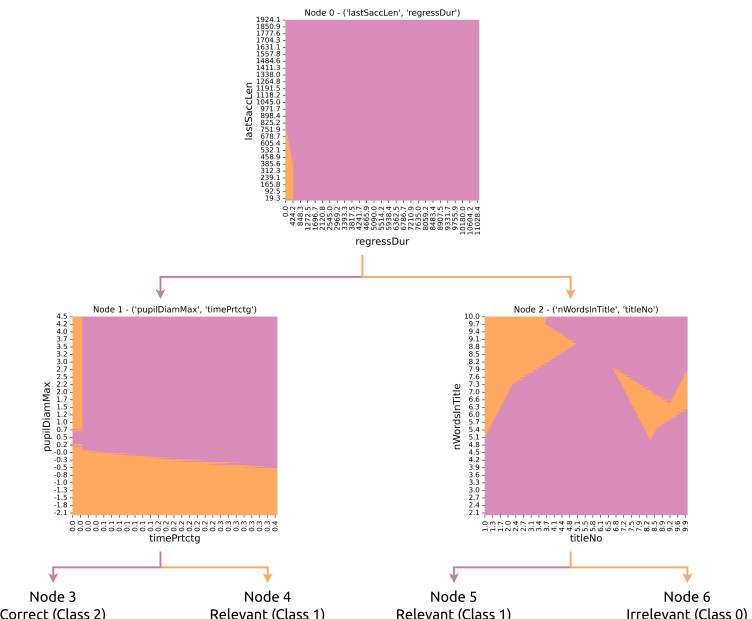

Figure H.7: Shape$^2$CART (Depth 2) Trained on the Eye-Movements dataset

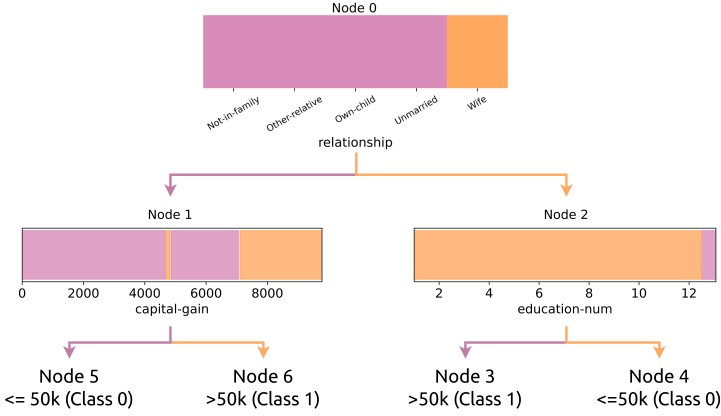

Figure H.8: ShapeCART (Depth 2) Trained on the Adult dataset

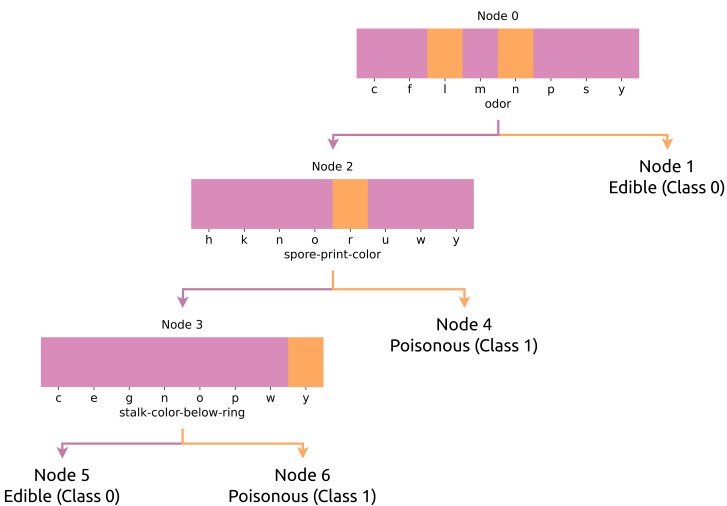

Figure H.9: ShapeCART (Depth 3) Trained on the Mushroom dataset

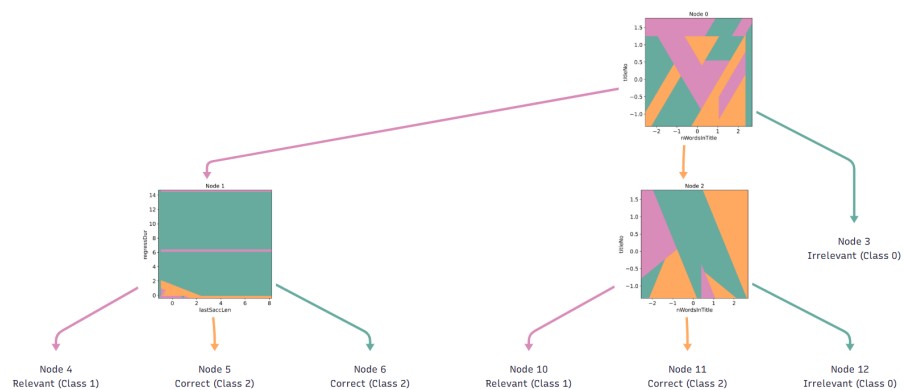

Figure H.10: Shape$^2$CART$_3$ (Depth 2) Trained on Eye-Movements

## I Other Shape Function Candidates

While our approach uses CART to construct the internal tree, our two-stage approach can be applied to any algorithm that can group samples, including other trees. To demonstrate this, we present a version of the ShapeCART algorithm that constructs the internal tree via DPDT [29] (denoted as SGT-C (D)). Note that we exclude Higgs from this evaluation as the DPDT version did not finish on most depths. Test accuracy results for this algorithm per dataset can be seen in Table 12 and runtime results in Table 13. Overall, we observe minor accuracy differences between constructing the internal tree with CART and DPDT. However, we observe that the runtime of the DPDT variant is much higher across all depths and datasets.

## J Regression

To adapt ShapeCART to the regression setting ($y_n \in \mathbb{R} \quad \forall n = 1, \dots, N$), we simply need to modify the superadditive impurity metric $\mathcal{H}$ from Gini/Entropy (class-based impurity measures), to one adapted for a regression setting, such as squared-error or absolute error. To accommodate this, we must modify the calculation of the empirical distribution of a set of samples (Equation (7)) to instead calculate the mean value of the target in the set of samples.

Table 12: Test Accuracy comparison (% ± std.) between ShapeCART and ShapeCART with DPDT internal tree.

| Dataset | Model | 2 | 3 | 4 | 5 | 6 | Overall |
|---|---|---|---|---|---|---|---|
| adult | SGT-C | **82.80 ± 0.20** | **84.10 ± 0.30** | 84.00 ± 0.30 | **84.30 ± 0.40** | **84.60 ± 0.60** | **84.60 ± 0.60** |
| | SGT-C (D) | 81.30 ± 1.90 | 83.90 ± 0.20 | **84.10 ± 0.30** | 84.20 ± 0.60 | 84.50 ± 0.60 | 84.50 ± 0.60 |
| avila | SGT-C | — | — | 64.50 ± 2.90 | 71.70 ± 1.80 | 82.00 ± 2.10 | 82.00 ± 2.10 |
| | SGT-C (D) | — | — | **67.30 ± 2.80** | **75.50 ± 2.80** | **83.80 ± 3.30** | **83.80 ± 3.30** |
| bank | SGT-C | **88.50 ± 2.30** | **89.50 ± 3.20** | **97.00 ± 0.90** | **96.70 ± 0.00** | **96.70 ± 0.00** | **97.00 ± 0.90** |
| | SGT-C (D) | 88.00 ± 2.00 | 89.00 ± 2.80 | 90.90 ± 2.20 | 92.40 ± 1.60 | 89.30 ± 4.10 | 92.40 ± 1.60 |
| bean | SGT-C | — | 85.10 ± 1.20 | **88.80 ± 0.40** | 89.70 ± 0.40 | 89.80 ± 0.10 | 89.80 ± 0.10 |
| | SGT-C (D) | — | **85.20 ± 0.90** | 88.70 ± 0.30 | **90.40 ± 0.90** | **90.90 ± 0.60** | **90.90 ± 0.60** |
| bidding | SGT-C | **98.10 ± 0.70** | **98.40 ± 0.60** | **99.70 ± 0.20** | **99.60 ± 0.30** | **99.60 ± 0.10** | **99.60 ± 0.10** |
| | SGT-C (D) | **98.10 ± 0.70** | 98.20 ± 0.70 | 99.50 ± 0.20 | 99.20 ± 0.20 | **99.60 ± 0.20** | **99.60 ± 0.20** |
| covtype | SGT-C | — | **68.90 ± 0.20** | 70.40 ± 0.10 | 72.10 ± 0.50 | 73.70 ± 0.20 | 73.70 ± 0.20 |
| | SGT-C (D) | — | 68.80 ± 0.70 | **70.50 ± 0.20** | **72.40 ± 0.10** | **73.80 ± 0.20** | **73.80 ± 0.20** |
| electricity | SGT-C | 75.50 ± 0.10 | 78.50 ± 0.40 | 81.30 ± 0.80 | 84.40 ± 0.10 | 84.80 ± 0.60 | 84.80 ± 0.60 |
| | SGT-C (D) | **75.60 ± 0.20** | **79.30 ± 0.60** | **82.50 ± 0.40** | **85.10 ± 0.60** | **86.60 ± 0.40** | **86.60 ± 0.40** |
| eucalyptus | SGT-C | — | 53.60 ± 3.30 | 55.90 ± 4.00 | 56.30 ± 4.60 | **58.80 ± 7.20** | **58.80 ± 7.20** |
| | SGT-C (D) | — | **54.10 ± 3.50** | **56.30 ± 4.80** | **57.20 ± 7.20** | 56.10 ± 1.20 | 57.20 ± 7.20 |
| eye-movements | SGT-C | **48.10 ± 2.10** | 53.00 ± 0.70 | 54.90 ± 1.20 | 53.90 ± 0.70 | 57.40 ± 1.20 | 57.40 ± 1.20 |
| | SGT-C (D) | 47.30 ± 0.30 | **53.20 ± 0.40** | **55.00 ± 1.20** | **57.00 ± 0.10** | **60.40 ± 0.20** | **60.40 ± 0.20** |
| eye-state | SGT-C | 64.30 ± 0.90 | 69.00 ± 0.20 | 70.50 ± 1.20 | 72.40 ± 1.30 | 74.50 ± 1.10 | 74.50 ± 1.10 |
| | SGT-C (D) | **66.60 ± 0.60** | **70.20 ± 0.50** | **72.50 ± 0.90** | **74.10 ± 0.10** | **75.30 ± 0.40** | **75.30 ± 0.40** |
| fault | SGT-C | — | 59.70 ± 3.90 | **65.30 ± 1.80** | 68.00 ± 0.40 | 69.70 ± 3.60 | 69.70 ± 3.60 |
| | SGT-C (D) | — | **60.20 ± 5.70** | **65.30 ± 1.40** | **70.10 ± 1.70** | 69.90 ± 2.50 | 69.90 ± 2.50 |
| gas-drift | SGT-C | — | **66.30 ± 1.30** | 76.00 ± 1.00 | 83.10 ± 2.00 | **89.00 ± 0.90** | **89.00 ± 0.90** |
| | SGT-C (D) | — | 66.00 ± 2.10 | **76.80 ± 1.20** | **83.80 ± 0.90** | 87.70 ± 2.70 | 87.70 ± 2.70 |
| htru | SGT-C | **97.80 ± 0.10** | 97.80 ± 0.10 | 97.80 ± 0.20 | **97.90 ± 0.20** | 97.60 ± 0.10 | **97.80 ± 0.10** |
| | SGT-C (D) | **97.80 ± 0.10** | 97.90 ± 0.30 | 97.90 ± 0.20 | 97.70 ± 0.10 | **97.80 ± 0.10** | **97.80 ± 0.10** |
| magic | SGT-C | 73.90 ± 1.00 | 77.10 ± 0.10 | **79.60 ± 1.10** | 80.10 ± 1.70 | **81.00 ± 0.50** | **81.00 ± 0.50** |
| | SGT-C (D) | **74.90 ± 1.50** | **77.60 ± 1.00** | 78.90 ± 1.00 | **80.20 ± 0.30** | 80.70 ± 0.70 | 80.70 ± 0.70 |
| mini-boone | SGT-C | **84.80 ± 0.20** | 87.10 ± 0.50 | 88.30 ± 0.30 | **89.20 ± 0.20** | **89.70 ± 0.10** | **89.70 ± 0.10** |
| | SGT-C (D) | 84.70 ± 0.20 | **87.40 ± 0.50** | **88.40 ± 0.30** | 89.10 ± 0.30 | 89.60 ± 0.30 | 89.60 ± 0.30 |
| mushroom | SGT-C | 98.60 ± 0.30 | 99.30 ± 0.10 | 99.80 ± 0.20 | **99.90 ± 0.10** | 99.90 ± 0.10 | 99.90 ± 0.10 |
| | SGT-C (D) | **98.90 ± 0.30** | **99.60 ± 0.20** | **99.90 ± 0.10** | **99.90 ± 0.10** | **100.00 ± 0.00** | **100.00 ± 0.00** |
| occupancy | SGT-C | **99.00 ± 0.10** | 98.90 ± 0.10 | 99.00 ± 0.10 | **99.00 ± 0.20** | **99.20 ± 0.20** | **99.20 ± 0.20** |
| | SGT-C (D) | **99.00 ± 0.10** | 98.90 ± 0.10 | **99.10 ± 0.20** | **99.00 ± 0.20** | 99.00 ± 0.20 | 99.00 ± 0.20 |
| page | SGT-C | — | 95.20 ± 0.30 | 96.40 ± 0.20 | **96.30 ± 1.20** | 96.20 ± 0.40 | 96.30 ± 1.20 |
| | SGT-C (D) | — | **95.60 ± 0.20** | **96.70 ± 0.20** | 96.00 ± 0.20 | **96.40 ± 0.20** | **96.70 ± 0.20** |
| pendigits | SGT-C | — | — | **79.30 ± 3.90** | 86.40 ± 0.70 | 88.30 ± 1.20 | 88.30 ± 1.20 |
| | SGT-C (D) | — | — | 78.10 ± 1.20 | **86.70 ± 0.10** | **90.10 ± 0.30** | **90.10 ± 0.30** |
| raisin | SGT-C | 85.70 ± 1.40 | **85.90 ± 2.30** | 84.30 ± 3.90 | 84.30 ± 2.90 | 84.40 ± 2.90 | 84.30 ± 2.90 |
| | SGT-C (D) | **85.90 ± 1.40** | 85.60 ± 2.20 | **85.90 ± 1.40** | **85.70 ± 2.20** | **85.70 ± 2.20** | **85.60 ± 2.20** |
| rice | SGT-C | **93.10 ± 0.40** | **92.70 ± 0.80** | **92.50 ± 0.90** | 92.10 ± 0.90 | 92.50 ± 0.50 | 92.50 ± 0.50 |
| | SGT-C (D) | **93.10 ± 0.40** | **92.70 ± 0.40** | 92.20 ± 0.90 | **93.00 ± 0.60** | **93.00 ± 0.60** | **93.00 ± 0.60** |
| room | SGT-C | 93.00 ± 0.40 | **98.30 ± 0.20** | **99.40 ± 0.10** | **99.40 ± 0.00** | **99.60 ± 0.00** | **99.60 ± 0.00** |
| | SGT-C (D) | **93.20 ± 0.60** | 98.20 ± 0.30 | 99.20 ± 0.20 | 99.20 ± 0.20 | 99.50 ± 0.30 | 99.50 ± 0.30 |
| segment | SGT-C | — | 83.30 ± 3.60 | **88.90 ± 0.80** | **89.80 ± 2.50** | **91.50 ± 1.90** | **91.50 ± 1.90** |
| | SGT-C (D) | — | **83.30 ± 3.70** | 87.80 ± 1.00 | 89.60 ± 2.50 | 91.10 ± 0.60 | 91.10 ± 0.60 |
| skin | SGT-C | 93.20 ± 0.20 | **97.70 ± 0.10** | **98.50 ± 0.00** | **98.90 ± 0.10** | **99.20 ± 0.20** | **99.20 ± 0.20** |
| | SGT-C (D) | **93.40 ± 0.30** | 97.60 ± 0.40 | 98.30 ± 0.20 | **98.90 ± 0.00** | **99.20 ± 0.30** | **99.20 ± 0.30** |
| wilt | SGT-C | 97.10 ± 0.60 | 97.10 ± 0.10 | **97.90 ± 0.50** | 97.60 ± 0.20 | 97.60 ± 0.20 | **97.90 ± 0.50** |
| | SGT-C (D) | **97.30 ± 0.40** | **97.20 ± 0.40** | 97.80 ± 0.50 | **97.90 ± 0.60** | **97.90 ± 0.60** | **97.90 ± 0.60** |

Table 13: Runtime comparison (s. ± std.) between SGT-C and SGT-C with DPDT internal tree.

| Dataset | Model | 2 | 3 | 4 | 5 | 6 | Overall |
|---|---|---|---|---|---|---|---|
| adult | SGT-C | **0.386 ± 0.009** | **0.596 ± 0.009** | **0.894 ± 0.016** | **1.120 ± 0.023** | **1.144 ± 0.033** | **1.144 ± 0.033** |
| | SGT-C (D) | 12.129 ± 0.425 | 21.990 ± 0.318 | 33.145 ± 2.011 | 42.382 ± 5.490 | 47.805 ± 7.702 | 47.805 ± 7.702 |
| avila | SGT-C | — | — | **1.711 ± 0.036** | **2.343 ± 0.007** | **2.733 ± 0.088** | **2.733 ± 0.088** |
| | SGT-C (D) | — | — | 79.209 ± 1.388 | 117.866 ± 2.197 | 147.429 ± 3.796 | 147.429 ± 3.796 |
| bank | SGT-C | **0.101 ± 0.001** | **0.138 ± 0.004** | **0.165 ± 0.015** | **0.184 ± 0.008** | **0.179 ± 0.010** | **0.165 ± 0.015** |
| | SGT-C (D) | 1.878 ± 0.051 | 3.073 ± 0.234 | 1.203 ± 0.145 | 1.334 ± 0.202 | 3.492 ± 0.586 | 1.334 ± 0.202 |
| bean | SGT-C | — | **1.892 ± 0.027** | **2.420 ± 0.071** | **1.797 ± 0.041** | **2.328 ± 0.061** | **2.328 ± 0.061** |
| | SGT-C (D) | — | 35.693 ± 0.138 | 14.431 ± 0.125 | 22.438 ± 0.111 | 32.694 ± 0.614 | 32.694 ± 0.614 |
| bidding | SGT-C | **0.132 ± 0.001** | **0.207 ± 0.002** | **0.208 ± 0.004** | **0.256 ± 0.004** | **0.333 ± 0.011** | **0.333 ± 0.011** |
| | SGT-C (D) | 0.998 ± 0.116 | 1.516 ± 0.195 | 2.172 ± 0.249 | 2.395 ± 0.414 | 2.660 ± 0.374 | 2.660 ± 0.374 |
| covtype | SGT-C | — | **6.135 ± 0.034** | **11.224 ± 0.014** | **10.618 ± 0.064** | **15.770 ± 0.096** | **15.770 ± 0.096** |
| | SGT-C (D) | — | 439.259 ± 2.396 | 227.655 ± 2.720 | 496.196 ± 4.219 | 603.667 ± 7.049 | 603.667 ± 7.049 |
| electricity | SGT-C | **0.396 ± 0.002** | **0.438 ± 0.006** | **0.880 ± 0.007** | **1.356 ± 0.011** | **2.145 ± 0.177** | **2.145 ± 0.177** |
| | SGT-C (D) | 5.202 ± 0.177 | 55.811 ± 0.318 | 89.526 ± 0.634 | 142.068 ± 0.423 | 197.176 ± 10.258 | 197.176 ± 10.258 |
| eucalyptus | SGT-C | — | **0.319 ± 0.048** | **0.523 ± 0.063** | **0.799 ± 0.046** | **0.995 ± 0.030** | **0.995 ± 0.030** |
| | SGT-C (D) | — | 9.549 ± 1.072 | 12.593 ± 1.672 | 21.382 ± 2.201 | 8.036 ± 0.352 | 21.382 ± 2.201 |
| eye-movements | SGT-C | **0.464 ± 0.009** | **0.910 ± 0.010** | **1.540 ± 0.020** | **2.501 ± 0.018** | **3.879 ± 0.329** | **3.879 ± 0.329** |
| | SGT-C (D) | 39.528 ± 0.188 | 74.187 ± 0.853 | 125.956 ± 2.508 | 193.079 ± 8.774 | 272.248 ± 22.320 | 272.248 ± 22.320 |
| eye-state | SGT-C | **0.339 ± 0.003** | **0.505 ± 0.002** | **0.678 ± 0.005** | **1.171 ± 0.005** | **2.087 ± 0.100** | **2.087 ± 0.100** |
| | SGT-C (D) | 7.504 ± 0.050 | 6.531 ± 0.032 | 10.843 ± 0.071 | 17.596 ± 0.487 | 27.433 ± 1.789 | 27.433 ± 1.789 |
| fault | SGT-C | — | **0.641 ± 0.032** | **1.044 ± 0.045** | **1.945 ± 0.153** | **2.599 ± 0.102** | **2.599 ± 0.102** |
| | SGT-C (D) | — | 12.921 ± 1.536 | 22.719 ± 2.346 | 15.749 ± 1.534 | 21.990 ± 1.183 | 21.990 ± 1.183 |
| gas-drift | SGT-C | — | **7.714 ± 0.016** | **14.076 ± 0.137** | **17.205 ± 0.505** | **20.722 ± 0.314** | **20.722 ± 0.314** |
| | SGT-C (D) | — | 167.784 ± 2.558 | 490.627 ± 2.307 | 763.837 ± 24.354 | 552.024 ± 10.415 | 552.024 ± 10.415 |
| htru | SGT-C | **0.328 ± 0.002** | **0.456 ± 0.004** | **0.646 ± 0.020** | **0.586 ± 0.065** | **0.561 ± 0.020** | **0.456 ± 0.004** |
| | SGT-C (D) | 9.063 ± 0.021 | 4.563 ± 0.040 | 6.069 ± 0.382 | 24.929 ± 4.588 | 7.213 ± 0.460 | 7.213 ± 0.460 |
| magic | SGT-C | **0.312 ± 0.000** | **0.477 ± 0.003** | **0.689 ± 0.003** | **1.071 ± 0.042** | **1.876 ± 0.061** | **1.876 ± 0.061** |
| | SGT-C (D) | 2.937 ± 0.019 | 5.079 ± 0.021 | 50.830 ± 0.901 | 28.735 ± 0.621 | 17.214 ± 1.284 | 17.214 ± 1.284 |
| mini-boone | SGT-C | **11.023 ± 0.078** | **12.995 ± 0.129** | **16.074 ± 0.116** | **20.609 ± 0.135** | **25.352 ± 0.062** | **25.352 ± 0.062** |
| | SGT-C (D) | 297.692 ± 1.729 | 153.416 ± 2.880 | 236.032 ± 2.896 | 284.030 ± 2.886 | 738.861 ± 24.597 | 738.861 ± 24.597 |
| mushroom | SGT-C | **0.185 ± 0.003** | **0.331 ± 0.008** | **0.399 ± 0.026** | **0.327 ± 0.004** | **0.330 ± 0.004** | **0.327 ± 0.004** |
| | SGT-C (D) | 3.644 ± 0.108 | 4.357 ± 0.128 | 4.824 ± 0.229 | 3.400 ± 0.045 | 4.881 ± 0.258 | 4.881 ± 0.258 |
| occupancy | SGT-C | **0.148 ± 0.002** | **0.186 ± 0.003** | **0.176 ± 0.004** | **0.197 ± 0.006** | **0.405 ± 0.024** | **0.405 ± 0.024** |
| | SGT-C (D) | 1.821 ± 0.010 | 7.952 ± 0.093 | 7.419 ± 0.673 | 9.229 ± 1.337 | 7.998 ± 0.908 | 7.998 ± 0.908 |
| page | SGT-C | — | **0.263 ± 0.001** | **0.437 ± 0.022** | **0.696 ± 0.075** | **0.933 ± 0.015** | **0.696 ± 0.075** |
| | SGT-C (D) | — | 6.266 ± 0.109 | 5.173 ± 0.086 | 13.694 ± 0.449 | 6.238 ± 0.361 | 5.173 ± 0.086 |
| pendigits | SGT-C | — | — | **2.312 ± 0.020** | **3.443 ± 0.053** | **4.788 ± 0.033** | **4.788 ± 0.033** |
| | SGT-C (D) | — | — | 128.053 ± 2.424 | 203.633 ± 0.778 | 70.396 ± 0.761 | 70.396 ± 0.761 |
| raisin | SGT-C | **0.137 ± 0.002** | **0.196 ± 0.011** | **0.263 ± 0.029** | **0.249 ± 0.034** | **0.317 ± 0.087** | **0.249 ± 0.034** |
| | SGT-C (D) | 1.620 ± 0.017 | 1.262 ± 0.275 | 2.892 ± 0.513 | 1.762 ± 0.414 | 1.832 ± 0.348 | 1.262 ± 0.275 |
| rice | SGT-C | **0.134 ± 0.002** | **0.239 ± 0.003** | **0.337 ± 0.029** | **0.441 ± 0.062** | **0.541 ± 0.049** | **0.541 ± 0.049** |
| | SGT-C (D) | 1.073 ± 0.015 | 1.987 ± 0.017 | 3.106 ± 0.356 | 3.663 ± 0.810 | 3.592 ± 0.679 | 3.663 ± 0.810 |
| room | SGT-C | **0.238 ± 0.004** | **0.535 ± 0.004** | **0.704 ± 0.027** | **0.770 ± 0.014** | **0.773 ± 0.012** | **0.773 ± 0.012** |
| | SGT-C (D) | 20.538 ± 0.232 | 9.141 ± 0.236 | 19.394 ± 0.328 | 22.729 ± 1.789 | 7.997 ± 0.356 | 7.997 ± 0.356 |
| segment | SGT-C | — | **0.410 ± 0.015** | **0.468 ± 0.016** | **0.844 ± 0.053** | **1.009 ± 0.109** | **1.009 ± 0.109** |
| | SGT-C (D) | — | 7.757 ± 0.310 | 20.347 ± 0.481 | 12.982 ± 0.307 | 16.016 ± 0.189 | 16.016 ± 0.189 |
| skin | SGT-C | **0.399 ± 0.009** | **0.520 ± 0.010** | **0.645 ± 0.008** | **0.776 ± 0.009** | **0.919 ± 0.018** | **0.919 ± 0.018** |
| | SGT-C (D) | 9.559 ± 0.388 | 21.933 ± 0.846 | 29.639 ± 0.523 | 35.859 ± 0.545 | 23.851 ± 2.256 | 23.851 ± 2.256 |
| wilt | SGT-C | **0.127 ± 0.002** | **0.169 ± 0.001** | **0.252 ± 0.009** | **0.292 ± 0.033** | **0.316 ± 0.059** | **0.252 ± 0.009** |
| | SGT-C (D) | 1.772 ± 0.048 | 1.539 ± 0.027 | 2.165 ± 0.260 | 2.814 ± 0.336 | 3.213 ± 0.159 | 3.213 ± 0.159 |

We provide regression results for ShapeCART, ShapeCART$_3$, Shape$^2$CART, and Shape$^2$CART$_3$. We benchmark our axis-aligned approaches against CART and our Bivariate approaches against BiCART. Our hyperparameter tuning approach and hyperparameter search in this setting are similar to the classification setting. One key modification is that we only use the MSE criterion (rather than tuning for a splitting criterion like in the classification setting). We report the mean squared error (MSE) of each model on five open source regression datasets - Ailerons (OpenML ID: 44137), Black Friday (OpenML ID: 41540), California Housing (OpenML ID: 44025), and Diamonds (OpenML ID: 42225) - broken down by depth and overall best model, chosen by best validation MSE, in Table 14. We scale the target to zero mean and one standard deviation for all datasets to ensure stability during training. We observe that ShapeCART consistently outperforms CART on all datasets except Ailerons, while ShapeCART$_3$ outperforms CART on all datasets. A similar trend can be seen in the Bivariate approaches, where Shape$^2$CART outperforms BiCART on all datasets except Ailerons. Interestingly, Shape$^2$CART also outperforms Shape$^2$CART$_3$ on California Housing and Ailerons, while achieving similar performance on Black-Friday and Diamonds.

Table 14: Test MSE Results (MSE $\pm$ std.) on regression datasets. Approaches are split into axis-aligned and bivariate by the dotted line. Best approach is **bolded**, second best is *italicized*.

| dataset | Model | 2 | 3 | 4 | 5 | 6 | Overall |
|---|---|---|---|---|---|---|---|
| ailerons | CART | $0.453 \pm 0.011$ | $0.368 \pm 0$ | $0.302 \pm 0.001$ | $0.257 \pm 0.006$ | $0.235 \pm 0.004$ | $0.235 \pm 0.004$ |
| | SGT-C | $0.452 \pm 0.012$ | $0.364 \pm 0.004$ | $0.312 \pm 0.003$ | $0.258 \pm 0.007$ | $0.238 \pm 0.005$ | $0.238 \pm 0.005$ |
| | SGT$_3$-C | $\mathbf{0.375 \pm 0.002}$ | $\mathbf{0.321 \pm 0.004}$ | $\mathbf{0.263 \pm 0.004}$ | $\mathbf{0.23 \pm 0.004}$ | $\mathbf{0.229 \pm 0.006}$ | $\mathbf{0.229 \pm 0.006}$ |
| | BiCART | *$0.363 \pm 0.004$* | $\mathbf{0.28 \pm 0.006}$ | $\mathbf{0.231 \pm 0.016}$ | $\mathbf{0.206 \pm 0.008}$ | $\mathbf{0.197 \pm 0.008}$ | $\mathbf{0.197 \pm 0.008}$ |
| | S$^2$GT-C | $\mathbf{0.358 \pm 0.008}$ | *$0.296 \pm 0.005$* | *$0.256 \pm 0$* | *$0.233 \pm 0.001$* | *$0.231 \pm 0.009$* | *$0.231 \pm 0.009$* |
| | S$^2$GT$_3$-C | $0.37 \pm 0.002$ | $0.311 \pm 0.005$ | $0.265 \pm 0.001$ | $0.237 \pm 0.008$ | $0.236 \pm 0.003$ | $0.236 \pm 0.001$ |
| black-friday | CART | $0.728 \pm 0.012$ | $0.608 \pm 0.008$ | $0.567 \pm 0.018$ | $0.513 \pm 0.011$ | $0.502 \pm 0.009$ | $0.502 \pm 0.009$ |
| | SGT-C | *$0.548 \pm 0.019$* | *$0.528 \pm 0.002$* | *$0.512 \pm 0.001$* | *$0.504 \pm 0.016$* | *$0.496 \pm 0.016$* | *$0.496 \pm 0.016$* |
| | SGT$_3$-C | $\mathbf{0.528 \pm 0.002}$ | $\mathbf{0.507 \pm 0.008}$ | $\mathbf{0.492 \pm 0.008}$ | $\mathbf{0.487 \pm 0.017}$ | $\mathbf{0.484 \pm 0.022}$ | $\mathbf{0.484 \pm 0.022}$ |
| | BiCART | $0.728 \pm 0.011$ | $0.606 \pm 0.012$ | $0.561 \pm 0.008$ | $0.508 \pm 0.011$ | $0.499 \pm 0.013$ | $0.499 \pm 0.013$ |
| | S$^2$GT-C | *$0.546 \pm 0.001$* | *$0.511 \pm 0.002$* | *$0.504 \pm 0.012$* | $\mathbf{0.491 \pm 0.011}$ | *$0.487 \pm 0.013$* | *$0.487 \pm 0.013$* |
| | S$^2$GT$_3$-C | $\mathbf{0.519 \pm 0.002}$ | $\mathbf{0.502 \pm 0.003}$ | $\mathbf{0.49 \pm 0.008}$ | *$0.492 \pm 0.007$* | $\mathbf{0.485 \pm 0.021}$ | $\mathbf{0.485 \pm 0.021}$ |
| california | CART | $0.54 \pm 0.015$ | $0.459 \pm 0.008$ | $0.418 \pm 0.006$ | $0.377 \pm 0.001$ | $0.338 \pm 0.004$ | $0.338 \pm 0.004$ |
| | SGT-C | *$0.508 \pm 0.005$* | *$0.43 \pm 0.007$* | *$0.354 \pm 0.002$* | *$0.305 \pm 0.001$* | *$0.272 \pm 0.008$* | *$0.272 \pm 0.008$* |
| | SGT$_3$-C | $\mathbf{0.426 \pm 0.007}$ | $\mathbf{0.33 \pm 0.001}$ | $\mathbf{0.271 \pm 0.002}$ | $\mathbf{0.254 \pm 0.006}$ | $\mathbf{0.245 \pm 0.001}$ | $\mathbf{0.245 \pm 0.001}$ |
| | BiCART | $0.442 \pm 0.001$ | $0.345 \pm 0.003$ | $0.297 \pm 0.002$ | $0.267 \pm 0.007$ | $0.243 \pm 0.009$ | $0.243 \pm 0.009$ |
| | S$^2$GT-C | $\mathbf{0.358 \pm 0.008}$ | $\mathbf{0.272 \pm 0.001}$ | $\mathbf{0.228 \pm 0.001}$ | $\mathbf{0.224 \pm 0.008}$ | $\mathbf{0.213 \pm 0.012}$ | $\mathbf{0.213 \pm 0.012}$ |
| | S$^2$GT$_3$-C | *$0.415 \pm 0.002$* | *$0.332 \pm 0$* | $0.276 \pm 0.007$ | $0.243 \pm 0.008$ | $0.236 \pm 0.001$ | $0.236 \pm 0.001$ |
| diamonds | CART | $0.169 \pm 0.001$ | $0.125 \pm 0.001$ | $0.105 \pm 0$ | $0.092 \pm 0$ | $0.081 \pm 0$ | $0.081 \pm 0$ |
| | SGT-C | *$0.169 \pm 0.001$* | *$0.11 \pm 0$* | *$0.084 \pm 0.002$* | *$0.062 \pm 0.001$* | *$0.047 \pm 0.001$* | *$0.047 \pm 0.001$* |
| | SGT$_3$-C | $\mathbf{0.116 \pm 0}$ | $\mathbf{0.079 \pm 0.001}$ | $\mathbf{0.048 \pm 0.001}$ | $\mathbf{0.038 \pm 0.003}$ | $\mathbf{0.031 \pm 0.001}$ | $\mathbf{0.031 \pm 0.001}$ |
| | BiCART | $0.163 \pm 0.001$ | $0.113 \pm 0.002$ | $0.084 \pm 0.001$ | $0.066 \pm 0.001$ | $0.051 \pm 0.001$ | $0.051 \pm 0.001$ |
| | S$^2$GT-C | *$0.128 \pm 0.002$* | $\mathbf{0.064 \pm 0.001}$ | $0.044 \pm 0.003$ | $0.042 \pm 0.001$ | *$0.036 \pm 0$* | *$0.036 \pm 0$* |
| | S$^2$GT$_3$-C | $\mathbf{0.106 \pm 0.001}$ | *$0.07 \pm 0.001$* | $\mathbf{0.041 \pm 0.002}$ | $\mathbf{0.034 \pm 0.002}$ | $\mathbf{0.031 \pm 0.002}$ | $\mathbf{0.031 \pm 0.002}$ |

# K Extra Pseudocode

In this section, we provide pseudocode for the TDIDT induction of a tree (Algorithm 3) and of our coordinate descent procedure (Algorithm 4).

---

**Algorithm 3:** ShapeCART (TDIDT Pseudocode)

---

**Input:** Features $\mathcal{X}$, Target $\mathcal{Y}$, Branching Factor $K$, Pairwise Candidate Limit $P$, Coordinate Descent Iterations $R$, Pairwise Reg. $\gamma$, Branching Reg. $\lambda$, Impurity $\mathcal{H}(\cdot)$, Weighted Impurity $\mathcal{L}(\cdot)$, Max Depth $M_{\max}$, Minimum Impurity Improvement $\mathcal{L}_{\min}$, Min. Samples Leaf $S_{\text{leaf}}$, Min. Samples Split $S_{\text{split}}$

**Output:** $\mathcal{I}, \mathcal{T}$

$\mathcal{I} \leftarrow \{\}; \quad \mathcal{T} \leftarrow \{\}$

$Q \leftarrow$ Initialize Empty Priority Queue

$\boldsymbol{L}^{\text{init}} \leftarrow \mathcal{L}(\mathcal{Y})$

$(\boldsymbol{D}^0, \boldsymbol{L}^0, f^0(\cdot)) \leftarrow$ SelectShapeFunc( $\mathcal{X}, \mathcal{Y}, K, P, R, \gamma, \lambda, \mathcal{H}(\cdot), \mathcal{L}(\cdot)$ )

$Q.\text{add}(\{0 : (\boldsymbol{D}^0, \boldsymbol{L}^{\text{init}} - \boldsymbol{L}^0, f^0(\cdot), 0, \mathcal{X} \times \mathcal{Y})\})$

**while** $|Q| > 0$ **do**

    $i, (\boldsymbol{D}^i, \tilde{\boldsymbol{L}}^i, f^i(\cdot), M^i, \mathcal{D}) \leftarrow \text{Pop}(Q)$

    **if** $\tilde{\boldsymbol{L}}^i < \mathcal{L}_{min}$ **then**

        | Add node $i$ to $\mathcal{T}$, assign the label as the dominant class in $\mathcal{D}$, continue to next node

    **end**

    **if** $M = M_{max}$ **then**

        | Add node $i$ to $\mathcal{T}$, assign the label as the dominant class in $\mathcal{D}$, continue to next node

    **end**

    **if** $|D| < S_{split}$ **then**

        | Add node $i$ to $\mathcal{T}$, assign the label as the dominant class in $\mathcal{D}$, continue to next node

    **end**

    **if** $\exists \mathcal{D}^j \in \boldsymbol{D} : |\mathcal{D}^j| < S_{leaf}$ **then**

        | Add node $i$ to $\mathcal{T}$, assign the label as the dominant class in $\mathcal{D}$, continue to next node

    **end**

    Add node $i$ and corresponding shape function $f^i(\cdot)$ to $\mathcal{I}$

    **foreach** $\mathcal{D}^j \in \boldsymbol{D}$ **do**

        Mark $j$ as child of $i$

        $\mathcal{X}^j, \mathcal{Y}^j \leftarrow \mathcal{D}^j$

        $(\boldsymbol{D}^j, \boldsymbol{L}^j, f^j(\cdot)) \leftarrow$ SelectShapeFunc( $\mathcal{X}^j, \mathcal{Y}^j, K, P, R, \gamma, \lambda, \mathcal{H}(\cdot), \mathcal{L}(\cdot)$ )

        $Q.\text{add}(j : \{\boldsymbol{D}^j, \boldsymbol{L}^j - \boldsymbol{L}^i, f^j(\cdot), M^i + 1, \mathcal{D}^j\})$

    **end**

**end**

**return** $\mathcal{I}, \mathcal{T}$

---

**Algorithm 4:** Coordinate Descent

---

**Input:** $\mathbf{a}, [\pi_1, \ldots, \pi_L], [W_1, \ldots, W_L], K, \mathcal{H}(\cdot), R$

**Output:** $\mathbf{a}, \boldsymbol{L}$

**for** $k \leftarrow 1$ **to** $K$ **do**

    $\boldsymbol{W}_k \leftarrow \sum_{\ell=1}^{L} W_\ell \, \mathbf{1}(a_\ell = k)$

    $\boldsymbol{P}_k \leftarrow \sum_{\ell=1}^{k} \mathbf{1}(a_\ell = k) (W_\ell \, \pi_\ell)$

**end**

$L^* \leftarrow \infty$ **for** $r \leftarrow 1$ **to** $R$ **do**

    **foreach** $\ell$ *in RandomizedOrder(1,...,L)* **do**

        $\boldsymbol{P}_{a_\ell} \leftarrow \boldsymbol{P}_{a_\ell} - W_\ell \, \pi_\ell; \boldsymbol{W}_{a_\ell} \leftarrow \boldsymbol{W}_{a_\ell} - W_\ell$

        $L \leftarrow \sum_{j=1}^{k} \boldsymbol{W}_k \mathcal{H}(\boldsymbol{P}_k / \boldsymbol{W}_k)$

        Store $\widetilde{\boldsymbol{P}}_k \leftarrow \boldsymbol{P}_k, \widetilde{\boldsymbol{W}}_k \leftarrow \boldsymbol{W}_k \quad \forall k = 1, \ldots, K$

        **for** $k \leftarrow 1$ **to** $K$ **do**         // Iterate through all Branches

            $\boldsymbol{W}_k \leftarrow \widetilde{\boldsymbol{W}}_k + W_\ell; \boldsymbol{P}_k \leftarrow \widetilde{\boldsymbol{P}}_k + W_\ell \, \pi_\ell$

            $L_k \leftarrow L - \widetilde{\boldsymbol{W}}_k \mathcal{H}(\widetilde{\boldsymbol{P}}_k / \widetilde{\boldsymbol{W}}_k) + \boldsymbol{W}_k \mathcal{H}(\boldsymbol{P}_k / \boldsymbol{W}_k)$

            **if** $L_k \leq L^*$ **then**         // Commit if weighted impurity improves

                $L^* \leftarrow L_k; a_\ell \leftarrow i$

                StoreState$(\boldsymbol{P}_1, \boldsymbol{W}_1, \ldots, \boldsymbol{P}_K, \boldsymbol{W}_K)$

            **end**

        **end**

        $(\boldsymbol{P}_1, \boldsymbol{W}_1, \ldots, \boldsymbol{P}_K, \boldsymbol{W}_K) \leftarrow$ RetrieveState()

    **end**

**end**

**return** $\mathbf{a}, \boldsymbol{L}$

---

