# OpenReview forum: "Empowering Decision Trees via Shape Function Branching"
_NeurIPS.cc/2025/Conference — NeurIPS 2025 poster_

### Official Review · Reviewer_85Ma · 2025-06-08

**Clarity:** 3
**Significance:** 2
**Originality:** 3
**Rating:** 4
**Confidence:** 4

**Summary:**

The authors introduce the Shape Generalized Tree, which learns a GAM with 1 or 2 features at each tree node (using a decision tree) rather than an axis-aligned split or a linear split. They learn these GAM splits in a recursive CART-style agorithm, and additionally allow for multi-way splits rather than binary splits. Experiments on 12 tabular datasets show that the resulting models can outperform traditional baselines for small model sizes while preserving a form of interpretability that allows for manual inspection of decision paths.

**Questions:**

It might be interesting to see more explorations of univariate shape function beyond using CART.

**Ethical Concerns:**

["NO or VERY MINOR ethics concerns only"]

**Final Justification:**

I appreciate the authors' response and have increased my score to 4. I maintain my concerns about comparisons to GAMs -- in realistic scenarios they seem to be much easier to globally interpret and often easier to locally interpret than the method proposed by authors, and I urge the authors to engage more seriously with this in the manuscript.

**Limitations:**

Besides the weaknesses listed above, the paper should include a more thorough discussion of the interpretability limitations introduced by the SGT. The nonlinearity introduced by the tree structure may make the SGT more difficult to interpret than a GAM, even one that has many more features (esp. if it makes the individual component functions for different features less understandable). Moreover, interpreting individual shape functions adds complexity beyond a typical binary axis-aligned split.

**Quality:**

2

**Strengths And Weaknesses:**

- S1: The authors tackle a well-motivated problem
- S2: The authors propose an intuitive method and explore its natural variants
- S3: The paper's writing is clear and the authors choices' for variations of the method seem reasonable
- W1: The authors are missing comparisons to relevant baselines for compact interpretable models such as GAMs (e.g. the [explainable boosting boosting machine](https://github.com/interpretml/interpret)), optimal trees (e.g. [GOSDT](https://arxiv.org/abs/2006.08690)), regularized trees (e.g. [HSTree](https://arxiv.org/abs/2202.00858)), and tree sums with limited feature interactions per tree.
- W2: The experimental results should be more thorough. The 12 selected datasets are somewhat arbitrary and could include a much wider set of a more standardized benchmark (e.g. [PMLB](https://github.com/EpistasisLab/pmlb)). For the depths that the authors evalute, all methods seem to underfit the data (atleast the performance continues to increase through depth 6, the highest depth evaluated). This suggests that in a realistic setting, users may require larger depths, especially after cross-validation. Perhaps the authors may want to evaluate on commonly used high-stakes datasets which are known to admit interpretable classification models (e.g. the classification datasets in [this work](https://arxiv.org/pdf/2201.11931))
- W3: It is also unclear whether models of different depths are comparable when shape functions include more parameters and are more difficult to interpret.

---

> ### Author Rebuttal · Authors · 2025-07-31
>
> We thank the reviewer and are happy to hear that they found our approach well-motivated and reasonable. In the following, we will address all the reviewer's suggestions and concerns.
>
> ### Clarification on GAMs
> We want to clarify that we are not fitting an GAM in each node. Rather, we are learning a piece-wise constant function with a bounded number of pieces ($L$) that best splits the data into two (or $K$) subsets. This is an important distinction as SGTs overcome many interpretability issues GAMs possess. Specifically,  In GAMs, a shape function is learned for each feature ($C$ shape function per feature in multi-class datasets with $C$ classes). The local explanation size in GAMs is identical to its global explanation size (all shape functions). In contrast, in decision trees the local explanation size is bound by the tree depth and the global explanation size is bound by the number of nodes in the tree (at most $2^\texttt{depth}$). Due to these reasons, GAMs are not comparable to trees and we do not include them in our evaluation.
>
> ### Baselines (W1)
>
> We agree that our work could benefit from comparisons to optimal methods. However, due to their lack of scalability on both tree depth and dataset size, we instead opted for two recent SOTA **near-optimal models**: DPDT (KDD 2025) [1], a non-greedy induction method based on MDPs that was found to be within 0.35% of optimal trees on average [1], SPLIT (ICML 2025) [2], a recent method for training near-optimal trees efficiently via lookahead methods that has been shown to effectively approximate the optimal tree in many cases (see Tables 7-9 in [2]). Additionally, per the reviewer recommendation, we include HSTrees (ICML 2022) [3]. An updated table of results is provided in the section below.  Note that we do not provide results for FIGS [6] as like GAMs, their local explanation sizes are not comparable.
>
> ### Experimental Evaluation (W2)
>
> **Benchmark Datasets:** We would like to emphasize that all datasets used in our study are well-established and widely adopted in the literature. They were chosen to ensure diversity in terms of sample size, number of classes, and the inclusion of both numerical and categorical features. However, to further improve the robustness of our empirical evaluations, we have expanded our evaluation to include the Quant-BnB benchmark [5] also used in DPDT [1], bringing the total number of datasets in our empirical analysis to 26. We note that our evaluation spans more datasets than any of the baselines considered. Average results across all datasets are provided below.
>
> **Concerns on Max Depth:** We appreciate the reviewer’s concern regarding the choice of maximum tree depth. Our decision to cap depth at 6 is driven by interpretability considerations and aligns with prior work on compact decision trees ([2,4] explore depths up to 6; [1] focuses primarily on depths 3–5). Since our goal is to improve predictive performance while preserving model interpretability, we believe that focusing on trees of reasonable depths is appropriate and effectively showcases the strengths of our approach.
>
> Note that based on the other reviews, we have improved the robustness and fairness of our hyperparameter tuning setting. Please see our response to Reviewer Vngp for details on these changes.
>
> |model|2|3|4|5|6|Best|
> |---|---|---|---|---|---|---|
> |CART|83.9%|81.6%|82.3%|84.0%|85.1%|85.2%|
> |SERDT|83.7%|80.9%|81.7%|83.7%|85.1%|85.1%|
> |HSTree|83.9%|81.6%|82.4%|84.0%|85.1%|85.1%|
> |ShapeCART|_84.5%_|_82.5%_|_83.7%_|_84.9%_|_86.2%_|_86.3%_|
> |ShapeCART_3|**85.7%**|**84.6%**|**85.9%**|**87.3%**|**87.8%**|**87.8%**|
> |AxTAO|83.9%|82.1%|82.8%|84.5%|85.5%|85.4%|
> |DPDT|84.9%|83.4%|83.8%|85.2%|86.2%|86.2%|
> |SPLIT|81.7%|76.6%|77.9%|79.5%|80.2%|80.3%|
> |ShapeTAO|_85.0%_|_83.5%_|_84.6%_|_85.9%_|_86.8%_|_86.8%_|
> |ShapeTAO_3|**86.4%**|**85.1%**|**86.2%**|**87.5%**|**88.0%**|**88.0%**|
>
> \* To accommodate slower non-greedy baselines, we downsample our largest dataset (Higgs) from 11m samples to 1m samples to ensure all experiments are completed in time. The final manuscript will include the results for the full dataset.
>
> We observe that ShapeCART$_3$ and ShapeCART consistently outperform the other TDIDT approaches, while ShapeTAO and ShapeTAO$_3$ outperform the non-greedy approaches.  In addition to overall average, we provide results on only the Quant-BnB benchmark below and observe a similar trend.
>
> | model | 2 | 3 | 4 | 5 | 6 | Best |
> |---|---|---|---|---|---|---|
> | CART | 89.1% | 88.1% | 87.8% | 89.1% | 90.0% | 90.0% |
> | SERDT | 89.1% | 87.0% | 87.3% | 88.9% | 90.1% | 90.1% |
> | HSTree | 89.1% | 88.1% | 87.8% | 89.2% | 90.1% | 90.0% |
> | ShapeCART | _89.4%_ | _88.4%_ | _88.8%_ | _89.6%_ | _90.7%_ | _90.7%_ |
> | ShapeCART_3 | **91.1%** | **90.3%** | **90.9%** | **92.1%** | **92.2%** | **92.2%** |
> | AxTAO | 89.1% | 88.5% | 88.3% | 89.7% | 90.3% | 90.3% |
> | DPDT | _89.9%_ | _89.9%_ | 89.2% | 90.0% | 90.7% | 90.7% |
> | SPLIT | 87.1% | 82.6% | 84.0% | 85.5% | 86.2% | 86.3% |
> | ShapeTAO | _89.9%_ | 89.3% | _89.6%_ | _90.5%_ | _91.2%_ | _91.2%_ |
> | ShapeTAO_3 | **91.4%** | **90.7%** | **91.4%** | **92.1%** | **92.1%** | **92.2%** |
>
> ### SGT Interpretability (W3)
>
> In contrast to GAMs, the local interpretability of an SGT is inherently bounded: a user need only examine at most one piecewise-constant shape function per decision node along a path, resulting in a local explanation length no greater than the tree’s maximum depth. While we acknowledge that interpreting the shape functions at each node of an SGT may be more complex than interpreting axis-aligned linear splits, our approach nonetheless preserves key interpretability properties such as simulability and modularity (Section 2.2). For example, take the tree trained on the `eye-movements` dataset (Figure 2). Given a sample with `regressDur` = 100, `lastSaccLen` = 0, and `nWordsInTitle` =4, we can read the tree and extract a local explanation of "The model predicted (Relevant, Class 1) because `regressDur` < 220 & `lastSaccLen` < 5 & `nWordsInTitle` $\in [4,5]$." Moreover, the enhanced expressiveness of SGTs can allow for smaller trees while retaining performance, thereby improving the sparsity of local explanations, satisfying the desiderata proposed by [7].
>
> ### Other Shape Function Representations (Q)
> We are happy to hear that the reviewer is interested to see more potential shape function representations. While we use CART to construct the internal tree, our two-stage approach can be applied to any algorithm that can group samples, including other trees. To demonstrate this, we present a version of the ShapeCART algorithm that constructs the internal tree via DPDT [1] (ShapeCART-DPDT). Results for this algorithm on three datasets can be seen below:
>
> | dataset |  | 2 | 3 | 4 | 5 | 6 | Overall |
> |---|---|---|---|---|---|---|---|
> | electricity | ShapeCART-DPDT | **75.6%** | **78.9%** | **82.5%** | 84.2% | **86.8%** | **86.8%** |
> |  | ShapeCART | 75.5% | 78.5% | 81.3% | **84.4%** | 84.8% | 84.8% |
> | eye-movements | ShapeCART-DPDT | 47.3% | 51.9% | 54.8% | **54.5%** | **59.2%** | **59.2%** |
> |  | ShapeCART | **48.1%** | **53.0%** | **54.9%** | 53.9% | 57.4% | 57.4% |
> | eye-state | ShapeCART-DPDT | **66.6%** | **70.1%** | **72.5%** | **73.9%** | **75.3%** | **75.3%** |
> |  | ShapeCART | 64.3% | 69.0% | 70.5% | 72.4% | 74.5% | 74.5% |
>
> We observe that the DPDT internal tree variant outperforms the CART variant across most depths. This is an interesting result and we will perform a full evaluation of this method to add to the revised manuscript.
>
> ### References
>
> [1] Kohler, H. et al. Breiman meets Bellman: Non-Greedy Decision Trees with MDPs.  KDD 2025.
>
> [2] Babbar, V. et al. Near-Optimal Decision Trees in a SPLIT Second. ICML 2025
>
> [3] Agarwal, A. et al. Hierarchical Shrinkage: Improving the accuracy and interpretability of tree-based models. ICML 2022
>
> [4] Souza, V. et al. Decision trees with short explainable rules. NeurIPS 2022.
>
> [5] Mazumder, R. et al. Quant-BnB: A scalable branch-and-bound method for optimal decision trees with continuous features ICML 2022.
>
> [6] Tan, Y. et al.(2025). Fast interpretable greedy-tree sums. PNAS, 122_(7), e2310151122.
>
> [7] Murdoch, W. J. et. al. (2019). Definitions, methods, and applications in interpretable machine learning. PNAS, 116(44), 22071-22080.

---

> > ### Comment · Reviewer_85Ma · 2025-08-01
> >
> > I appreciate the authors' response and have increased my score to 4. I maintain my concerns about comparisons to GAMs -- in realistic scenarios they seem to be much easier to globally interpret and often easier to locally interpret than the method proposed by authors, and I urge the authors to engage more seriously with this in the manuscript.

---

> > > ### Author Response · Authors · 2025-08-01
> > >
> > > Thank you for your consideration and we appreciate you increasing your score.
> > >
> > > We will incorporate a deeper discussion in the paper about the comparison with GAMs and the need for further human studies to evaluate the interpretability of SGTs in various real-world applications.

---

### Official Review · Reviewer_NeDQ · 2025-06-14

**Clarity:** 4
**Significance:** 3
**Originality:** 3
**Rating:** 5
**Confidence:** 4

**Summary:**

This work introduces SGTs, a novel type of decision trees that supports non-linear branching functions at each internal node. In doing so, these models are able to represent complex decision functions more concisely than traditional linear splitting decision trees. Additionally, algorithms for fitting SGTs (ShapeCART) , a generalization of SGTs that supports multi-way splitting (ShapeCART_k), and a generalization that supports bivariate splitting functions are introduced. These trees are shown to improve accuracy relative to baseline greedy decision tree methods across several datasets.

**Questions:**

I have no major questions, but one minor copyediting note:
- Should w_d be unbolded in Equation 6? My understanding is that it would be a scalar.

**Ethical Concerns:**

["NO or VERY MINOR ethics concerns only"]

**Final Justification:**

This paper represents a clear, solid improvement on existing decision tree methods. The evaluation includes many benchmark datasets, forming a solid evaluation of the method. My primary concerns have been addressed during the rebuttal, so I am happy to recommend acceptance for this paper.

**Limitations:**

Yes

**Quality:**

3

**Strengths And Weaknesses:**

Strengths
- This work is well motivated, and identifies a nuanced but clear problem in existing decision trees (inefficiency in representing some kinds of splitting functions).
- The exposition is very clear, and the figures are helpful. In particular, the visualization technique used for SGTs is very clear.
- Fitting SGTs is a somewhat difficult multipart optimization problem. This work presents reasonable simplifications to make this a feasible problem to solve; in particular, the approach of first learning a binning function, then learning bin assignments to learn each splitting function is quite elegant.
- The evaluation is thorough and well described, considering multiple datasets of varying scale.

Weaknesses
- My primary concerns with this work have to do with the presentation of the experiments. If these can be addressed, I think this paper is a solid accept.
  - The Z-score presentation of results is quite strange, and hard to interpret. Moreover, I see no reason to think that the aggregated test accuracies are normally distributed or support a CLT — they would be drawn from some kind of mixture distribution. As such, Z-scores are not an applicable tool. Additionally, it does not make sense to aggregate Z-scores over depth bounds. It may be more appropriate to present accuracy numbers from a reasonable subset of datasets in the main paper for comparison. I believe tables 1 and 2 could be merged if accuracy is reported directly, allowing space for 3-4 datasets to be included.
  - In the presentation of results in the appendix, grouping tables by method makes comparisons quite difficult. I would recommend grouping them by dataset, making head to head comparisons easier.
  - While greedy methods are the most direct competitors, globally optimal trees could be run for the shallower depth bounds, and should be included as competing methods. It does not necessarily take away from ShapeCART if they happen to find superior performance, since they likely scale less well.

---

> ### Author Rebuttal · Authors · 2025-07-31
>
> We thank the reviewer and are happy to hear that they found our approach well-motivated and elegant. We are happy to hear that the reviewer thinks our paper is a solid accept if the presentation of the experiments can be improved. In the following, we will address all the reviewer's suggestions and concerns.
>
> ### Z-Scores
> Thanks for pointing out the limitations associated with presenting Z-scores. Our intention was to enable a fairer comparison across methods by standardizing performance, following the approach in [1]. However, following the reviewer's suggestion, we have revised our presentation to report raw accuracy values, averaged per depth and across datasets, to ensure clarity and transparency in our results. Revised tables can be found in the "Merging Tables and Organization" section below.
>
> ### Extra Baselines
>
> We agree that our work could benefit from comparisons to optimal methods. However, due to their lack of scalability on both tree depth and dataset size, we instead opted for two recent SOTA **near-optimal models**: DPDT (KDD 2025) [2], a non-greedy induction method based on MDPs that was found to be within 0.35% of optimal trees on average [2]., SPLIT (ICML 2025) [3], a recent method for training near-optimal trees efficiently via lookahead methods that has been shown to effectively approximate the optimal tree in many cases (see Tables 7-9 in [3]).
>
> In addition to incorporating additional baselines, we introduce TAO-refined variants of our multi-way branching models (ShapeCART$_3$ and Shape$^2$CART$_3$ ) by extending the original TAO algorithm from the binary to the ternary setting. This extension is achieved by modifying the node-level optimization problem to handle multi-way splits, specifically by upsampling samples that can be routed to multiple branches and appropriately adjusting their weights during the fitting of the internal tree. Additional implementation details will be provided in the appendix of the revised manuscript and are available upon request.
>
> Results for the newly considered approaches can be found in the section below.
>
> ### Merging Tables and Organization
> Thanks for your suggestion to merge tables and increase the available space for content. We would like to clarify that the values reported in Tables 1 and 2 are not directly comparable, as the evaluation of bivariate approaches excludes three datasets (Covertype, Higgs, and Eucalyptus) due to out-of-memory issues with the baselines. Specifically, both BiCART and BiTAO exceed the 32GB memory threshold when training on these datasets, primarily due to their feature augmentation mechanism. That said, we agree that the current table layout could be improved to conserve space and enhance readability. While we are unable to upload a revised version of the paper as part of the rebuttal, we will reorganize the tables in the revised manuscript by displaying them side-by-side and minimizing unnecessary white space in the final version. For reference, we include below the aggregated test accuracy results across datasets and depths.
>
> Note that based on the other reviews, we have improved the robustness and fairness of our hyperparameter tuning setting. Please see our response to Reviewer Vngp for details on these changes. Additionally, we have added additional datasets at the request of Reviewer 85ma, bringing our total dataset count to 26.
>
> Aggregated results for all evaluated approaches can be seen in the table below. We observe that ShapeCART$_3$ and ShapeCART consistently outperform the other TDIDT approaches, while ShapeTAO and ShapeTAO$_3$ outperform the non-greedy approaches.
>
> |model|2|3|4|5|6|Best|
> |---|---|---|---|---|---|---|
> |CART|83.9%|81.6%|82.3%|84.0%|85.1%|85.2%|
> |SERDT|83.7%|80.9%|81.7%|83.7%|85.1%|85.1%|
> |HSTree|83.9%|81.6%|82.4%|84.0%|85.1%|85.1%|
> |ShapeCART|_84.5%_|_82.5%_|_83.7%_|_84.9%_|_86.2%_|_86.3%_|
> |ShapeCART_3|**85.7%**|**84.6%**|**85.9%**|**87.3%**|**87.8%**|**87.8%**|
> |AxTAO|83.9%|82.1%|82.8%|84.5%|85.5%|85.4%|
> |DPDT|84.9%|83.4%|83.8%|85.2%|86.2%|86.2%|
> |SPLIT|81.7%|76.6%|77.9%|79.5%|80.2%|80.3%|
> |ShapeTAO|_85.0%_|_83.5%_|_84.6%_|_85.9%_|_86.8%_|_86.8%_|
> |ShapeTAO_3|**86.4%**|**85.1%**|**86.2%**|**87.5%**|**88.0%**|**88.0%**|
>
> \* To accommodate slower non-greedy baselines, we downsample our largest dataset (Higgs) from 11m samples to 1m samples to ensure all experiments are completed in time. The final manuscript will include the results for the full dataset.
>
> We appreciate the suggestion to present dataset-level results for a small subset of datasets across depths in the main paper and agree that this will help illustrate that SGTs can achieve high predictive performance with smaller tree sizes. Below we provide test accuracy values on three datasets `eye-movements, electricity, & eye-state` across all depths:
>
> | Dataset | Model | 2 | 3 | 4 | 5 | 6 | Best |
> |---|---|---|---|---|---|---|---|
> | eye-movements | CART | 45.9% | 46.3% | 51.3% | 52.4% | 53.2% | 53.2% |
> |  | SERDT | 43.8% | 49.2% | 49.1% | 52.7% | 53.5% | 53.5% |
> |  | HSTree | 45.9% | 46.3% | 51.4% | 52.4% | 53.1% | 53.1% |
> |  | ShapeCART | **48.1%** | _53.0%_ | _54.9%_ | _53.9%_ | _57.4%_ | _57.4%_ |
> |  | ShapeCART_3 | _47.1%_ | **54.4%** | **56.0%** | **57.5%** | **60.9%** | **60.9%** |
> |  | AxTAO | 45.9% | 48.6% | 51.4% | 52.3% | 52.6% | 52.6% |
> |  | DPDT | _48.7%_ | 51.2% | 52.4% | 54.4% | 56.7% | 56.7% |
> |  | SPLIT | 44.0% | 48.4% | 50.2% | 51.9% | 52.0% | 51.9% |
> |  | ShapeTAO | 48.6% | _53.1%_ | _54.8%_ | _56.8%_ | _58.0%_ | _58.0%_ |
> |  | ShapeTAO_3 | **51.4%** | **54.8%** | **55.4%** | **56.9%** | **60.3%** | **60.3%** |
> | electricity | CART | _75.5%_ | 75.8% | 76.0% | 77.6% | 78.0% | 78.0% |
> |  | SERDT | _75.5%_ | 75.8% | 76.2% | 78.6% | 79.4% | 79.4% |
> |  | HSTree | _75.5%_ | 75.8% | 76.0% | 77.7% | 78.1% | 78.1% |
> |  | ShapeCART | _75.5%_ | _78.5%_ | _81.3%_ | _84.4%_ | _84.8%_ | _84.8%_ |
> |  | ShapeCART_3 | **76.5%** | **80.6%** | **84.0%** | **86.0%** | **87.8%** | **87.8%** |
> |  | AxTAO | 75.5% | 75.9% | 75.9% | 78.5% | 79.9% | 79.9% |
> |  | DPDT | _76.1%_ | 78.3% | 79.1% | 80.0% | 80.8% | 80.8% |
> |  | SPLIT | 75.4% | 76.2% | 77.7% | 78.4% | 78.7% | 78.7% |
> |  | ShapeTAO | 75.5% | _79.5%_ | _83.0%_ | _86.0%_ | _87.6%_ | _87.6%_ |
> |  | ShapeTAO_3 | **79.3%** | **82.6%** | **84.9%** | **86.7%** | **88.2%** | **88.2%** |
> | eye-state | CART | 62.7% | 65.8% | 69.1% | 70.9% | 73.2% | 73.2% |
> |  | SERDT | 62.7% | 66.4% | 69.4% | 72.3% | 75.3% | 75.3% |
> |  | HSTree | 62.7% | 65.8% | 69.1% | 70.9% | 73.2% | 73.2% |
> |  | ShapeCART | _64.3%_ | _69.0%_ | _70.5%_ | _72.4%_ | _74.5%_ | _74.5%_ |
> |  | ShapeCART_3 | **66.1%** | **70.8%** | **73.3%** | **76.8%** | **77.5%** | **77.5%** |
> |  | AxTAO | 62.1% | 67.0% | 69.7% | 72.8% | 74.3% | 74.3% |
> |  | DPDT | _65.9%_ | 69.1% | 71.4% | 73.4% | 75.6% | 75.6% |
> |  | SPLIT | 58.5% | 66.4% | 69.0% | 71.8% | 73.6% | 73.6% |
> |  | ShapeTAO | 65.6% | _70.5%_ | _72.7%_ | _73.6%_ | _75.8%_ | _75.8%_ |
> |  | ShapeTAO_3 | **67.1%** | **71.9%** | **75.8%** | **75.8%** | **76.3%** | **76.3%** |
>
> We observe that in many cases, ShapeCART can achieve test accuracy with a shallow tree exceeding that of other TDIDT approaches on the maximum depth. While not as significant, we observe that ShapeTAO can also achieve higher test accuracy than deeper non-greedy approaches.
>
> Thank you for the advice regarding the tables in the Appendix. Due to character limitations, we are unable to provide tables for all datasets in this rebuttal, but in the revised manuscript, we will re-organize the results in the appendix similar to the per-dataset comparison table above.
>
> **Equation 6:** Thank you for pointing out the typo in Equation 6. This is a scalar and should be unbolded.
>
> ### References
>
> [1] Feuer, B. et. al. Tunetables: Context optimization for scalable prior-data fitted networks.  NeurIPS 2024
>
> [2] Kohler, H., et. al. Breiman meets Bellman: Non-Greedy Decision Trees with MDPs.  KDD 2025.
>
> [3] Babbar, V., et. al. Near-Optimal Decision Trees in a SPLIT Second. ICML 2025
>
> [4] Agarwal, A. et. al. Hierarchical Shrinkage: Improving the accuracy and interpretability of tree-based models. ICML 2022

---

> > ### Comment · Reviewer_NeDQ · 2025-07-31
> >
> > Thank you for your thorough response! My primary concerns have been addressed, so I am happy to raise my score to "accept". I think this will be a strong paper.
> >
> > On a minor note that is not effecting my score/overall recommendation, I would suggest that the authors double check the results from SPLIT. My understanding is that SPLIT is a compromise between optimal and greedy trees, using the CART splitting criteria for the greedy component. As such, I am somewhat surprised to see CART consistently outperform it -- it seems like they should be equivalent in the worst case?

---

> > > ### Author Response · Authors · 2025-08-01
> > >
> > > We thank the reviewer for their vote of confidence on our work!
> > >
> > > **Regarding SPLIT:** We have observed the degraded performance of SPLIT and have looked into it.
> > > * As noted in Section 2.1, some optimal methods, including GOSDT [5] and SPLIT [3] (which builds on GOSDT) require pre-binarization of continuous features. We use the official implementation of SPLIT from the authors' GitHub repository and set  `binarize=True` that applies Threshold Guessing for binarization (as used in the paper), producing a sparse set of binary features for splitting.  While full binarization over all thresholds may improve performance, they are orders-of-magnitude more computationally expensive (see Figure 13 in [3]).
> > > * We believe the binarization is responsible for the observed degraded performance. To validate this, we trained CART on the binarized data produced by SPLIT's official implementation and observed similar performance degradation.
> > > * It is worth noting that in our work CART runs on the original continuous feature (while the comparison to CART in [3] assumes CART is using the same binarized features as SPLIT). That is why in our experiments CART is outperforming SPLIT on average.
> > >
> > > We will clarify these details in the final manuscript and include dataset-level results for a more fine-grained comparison.
> > >
> > > [3] Babbar, V., et. al. Near-Optimal Decision Trees in a SPLIT Second. ICML 2025
> > > [5] Lin, Jimmy, et al. "Generalized and scalable optimal sparse decision trees." ICML 2020.

---

> > > > ### Comment · Reviewer_NeDQ · 2025-08-01
> > > >
> > > > This makes sense -- thank you for following up!

---

### Official Review · Reviewer_Vngp · 2025-06-30

**Clarity:** 3
**Significance:** 3
**Originality:** 3
**Rating:** 5
**Confidence:** 4

**Summary:**

This paper proposes to extend the top-down induction tree learning paradigm from only considering linear axis-aligned splits to both (1) non-linear axis-aligned splits on a single feature by using shape functions, (2) non-linear splits on a pair of two features, and (3) multi-way splits (although in practice limited to three-way splits). In comparison with linear axis-aligned splits, the new splits can compactly capture more patterns, whereas in comparison to oblique splits, these shape functions remain more interpretable.

**Questions:**

1. Categorical variables are transformed using one-hot encoding. This means the only possible shape function is the standard binary split, right? Are other encodings of categorical variables possible to exploit the value of shape functions?

**Ethical Concerns:**

["NO or VERY MINOR ethics concerns only"]

**Final Justification:**

The proposed method is simple (in the good sense), useful, and well-explained. Their rebuttal shows that my understanding of their binarization of categorical variables was based on a misunderstanding. My main concern was regards to reproducibility because of lack of details in the hyperparameter tuning of the method. This concern is answered well in the rebuttal and should be expanded on in the updated paper. Therefore, I recommend to accept this paper.

**Limitations:**

yes

**Quality:**

3

**Strengths And Weaknesses:**

Strengths:
- The proposed method is simple and explained well
- The paper examines a variety of novel split types (shape functions, shape functions on pairs of features, multi-way splits)
- The formalism is useful and appropriate
- The related work contains a reflection on the different types of interpretability and the applicability to trees

Weaknesses:
- The experiment hyperparameter tuning needs more details. There are several hyperparameters that are tuned, but how? (1) L, Number of bins per shape function, (2) min. samples per bin, (3) gamma, regularization hyperparameter, (4) lambda, penalization of multi-way branching

Small comments
- “For bivariate shape functions, we instead train a binary bivariate linear tree using BiCART”. More details is needed. Also, what is a linear tree?
- Eq. 11 needs a comma between the two constraints for readability
- In Def. 5, k is not defined (only later). Also k>=2 in the text, but in Eq. 5, k >= 1

---

> ### Author Rebuttal · Authors · 2025-07-31
>
> We thank the reviewer and are happy to hear that they found our approach well-explained, novel, and useful. We hope to answer any remaining concerns and questions below.
>
> ### Categorical Variables:
>
> ShapeCART and its variants support superset branching on categorical variables without requiring any ordinal assumptions among the categories, owing to the flexibility of their shape functions. This is achieved by passing all relevant columns of the one-hot encoded categorical variable into the internal tree, as described in Section 5 (lines 275–277).  An example illustrating SGT branching over multiple levels of a categorical feature is provided in Figure 5 of the appendix, which visualizes an SGT trained on the mushroom dataset (a fully categorical dataset). We are happy to clarify any remaining questions regarding this implementation detail.
>
> ### Tuning and Evaluation:
>
> Thanks for highlighting the need for additional detail regarding hyperparameter tuning. In response, we will expand the description of our tuning protocol in Section 5 and Appendix E.1 based on the description below. In addition, since submission we have improved our hyperparameter tuning procedure to ensure fair comparison across models and datasets by allocating fixed tuning budget per model and per dataset.
>
> * We standardize the evaluation by creating three folds for each dataset using a 70/30 train/test-validation split. The non-training portion is then further divided into validation and test sets using the same 70/30 ratio.
> * For hyperparameter optimization, we perform a Bayesian search with a fixed budget of 50 trials for each model and depth via the Optuna Framework [1] to avoid allocating larger budget to model with more hyperparameters. We believe this adjustment strengthens the methodological rigor of our comparisons.
> * We evaluate each configuration by its average validation accuracy across the folds, and select the hyperparameter combination that results in the best average validation accuracy across all folds per explored depth, as well as the single best overall model.
> * Each model’s training time was limited to 15 minutes for axis-aligned (univariate) approaches and 30 minutes for (bivariate) approaches.
> * All experiments are run on GCP N2 Compute Engines (Intel Cascade Lake) with 8vCPUs and 32 GB Memory
>
> ### Hyperparameter Importance
>
> Given the reviewer's interest in the hyper-parameter tuning, we would also like to share that we have conducted an analysis of hyper-parameter importance for our approaches. Following [2] and [5], we normalize the test accuracy within each model and dataset to zero mean and one std. and train a Random Forest regressor to predict test Z-scores from the model hyperparameters and provide the feature importance for our TDIDT approaches in the table below. We will include this analysis in the final manuscript:
> |  | ShapeCART | ShapeCART$_3$ | Shape$^2$CART | Shape$^2$CART$_3$ |
> |---|---|---|---|---|
> | max_depth | 0.35 | 0.24 | 0.148 | 0.099 |
> | criterion | 0.038 | 0.035 | 0.035 | 0.034 |
> | min_samples_split | 0.079 | 0.082 | 0.082 | 0.085 |
> | min_samples_leaf | 0.182 | 0.178 | 0.193 | 0.191 |
> | min_impurity_decrease | 0.094 | 0.08 | 0.099 | 0.074 |
> | inner_max_leaf_nodes | 0.169 | 0.192 | 0.177 | 0.199 |
> | inner_min_samples_leaf | 0.088 | 0.098 | 0.098 | 0.083 |
> | branching_penalty | --- | 0.095 | --- | 0.085 |
> | H | --- | --- | 0.069 | 0.062 |
> | pairwise_penalty | --- | --- | 0.099 | 0.088 |
>
> We observe that `max_depth, inner_max_leaf_nodes` (aka $L$), and `min_samples_leaf` are among the top-3 most important HPs for all models evaluated.
>
> ### Typos
>
> Thank you for catching some of our typos and readability issues. Here are the provided fixes:
> * Definition 5: We modify "...partition the data into $K \geq k \geq 2$ distinct subsets.." to "...partition the data into at most $K$ distinct subsets.."
> * Equation 11: We will add a comma between the constraints.
> * Linear Tree: This should read "bivariate oblique tree." This refers to a type of tree where the splitting function is a linear combination of 2 features. Formal definitions of these tree types are provided in Definitions 2 (Page 3) ($M$=2). BiCART [6] is a newly proposed TDIDT method for training bivariate oblique trees.
>
> ### References
> [1] Akiba, T. et. al. Optuna: A next-generation hyperparameter optimization framework. (KDD 2019)
>
> [2] Kohler, H., et. al. Breiman meets Bellman: Non-Greedy Decision Trees with MDPs.  KDD 2025.
>
> [3] Babbar, V., et. al. Near-Optimal Decision Trees in a SPLIT Second. ICML 2025
>
> [4] Agarwal, A. et. al. Hierarchical Shrinkage: Improving the accuracy and interpretability of tree-based models. ICML 2022
>
> [5] Grinsztajn, Léo, et. al. "Why do tree-based models still outperform deep learning on typical tabular data?." NeurIPS 2022.
>
> [6] Kairgeldin, R., and Carreira-Perpiñán, M. Á.. Bivariate decision trees: Smaller, interpretable, more accurate. KDD 2024

---

> > ### Comment · Reviewer_Vngp · 2025-08-01
> >
> > Thanks for expanding on the hyper-parameter tuning, and for clarifying splitting on categorical variables. I think this paper is a valuable contribution.

---

> > > ### Author Response · Authors · 2025-08-01
> > >
> > > Thank you for your vote of confidence in our work!

---

### Official Review · Reviewer_w6vq · 2025-07-05

**Clarity:** 4
**Significance:** 2
**Originality:** 3
**Rating:** 4
**Confidence:** 4

**Summary:**

This paper introduces ​Shape Generalized Trees (SGTs)​, a novel class of interpretable decision trees that replace traditional axis-aligned linear splits with ​learnable shape functions​ applied to individual features in each node. These non-linear functions capture complex feature-target relationships within single nodes, significantly reducing tree depth and size while maintaining model transparency. The authors develop ​ShapeCART, an efficient induction algorithm that constructs univariate SGTs using an interpretable binning-and-assignment approach, alongside extensions for bivariate splits (Shape²CART) and multi-way branching (ShapeCARTₖ). Experiments demonstrate that SGT variants consistently outperform conventional trees (CART, SERDT) and bivariate methods (BiCART) across tabular datasets, achieving ​higher accuracy with more compact architectures. Crucially, SGTs retain explainability through ​visualizable node-level decisions, as illustrated in the eye-movements dataset example where non-linear splits replace multiple linear thresholds. This work bridges the expressivity of generalized additive models with the structural clarity of decision trees.

**Questions:**

See weaknessses

**Ethical Concerns:**

["NO or VERY MINOR ethics concerns only"]

**Final Justification:**

Thank you for your thorough response! My primary concerns have been addressed, so I am happy to raise my score.

**Limitations:**

Nan

**Paper Formatting Concerns:**

Nan

**Quality:**

3

**Strengths And Weaknesses:**

**Related Work and Theory**: The paper establishes ​empirical superiority​ of SGTs over CART but omits ​theoretical guarantees​ comparable to recent CART analyses (Klusowski 2020, 2024; Zheng 2023). While ShapeCART's greedy optimization shares CART's structure, its ​non-linear shape functions​ fundamentally alter splitting mechanics. Future work should address: (1) ​Approximation bounds​ for function classes captured by piecewise-constant shape functions; (2) ​Consistency rates​ under the binning-and-assignment paradigm; (3) ​Sparsity guarantees​ given the coordinate descent assignment. Connecting to Klusowski's minimax optimality for additive functions or Zheng's consistency analysis would ground SGTs' advantages in modern learning theory.
* Klusowski, Jason M., and Peter M. Tian. "Large scale prediction with decision trees." Journal of the American Statistical Association 119.545 (2024): 525-537.
* Klusowski, Jason. "Sparse learning with CART." Advances in Neural Information Processing Systems 33 (2020): 11612-11622.
* Zheng,  et al. "On the consistency rate of decision tree learning algorithms." International Conference on Artificial Intelligence and Statistics. PMLR, 2023.

**More Experiments**:
1) Hyperparameter sensitivity analysis​ lacks visualizations of key parameters' impact (e.g., bin count L, pairwise regularization γ) across tasks—critical given their effect on model compactness; ​
2) Visualization limitations—while Figure 2 illustrates a univariate SGT node, the paper doesn't showcase ​bivariate/multi-way shape functions​ (promised in Sec. 4) or ​comparative architecture diagrams​ against linear trees (as implied in Fig. 1c);
3) Baseline contemporaneity gaps—recent advances like NODE-GAM and interpretable oblique trees are absent despite relevance to SGT's "interpretable non-linearity" claims. While depth-controlled comparisons are rigorous (Tables 1-2), adding ​parameter sensitivity curves, ​bivariate shape visualizations, and ​modern baseline comparisons​ would better substantiate the novel structure-performance trade-offs central to SGTs' contribution.

---

> ### Author Rebuttal · Authors · 2025-07-31
>
> We thank the reviewer for their detailed review and suggestions for improvement. In the following, we will address all the reviewer's suggestions and concerns.
>
> ## Theoretical Results
>
> We agree that incorporating theoretical analysis of SGTs and ShapeCART will enhance the work and ground it in a solid theoretical foundation. We are happy to share that we have obtained new theoretical results on: (1) the expressivity of SGTs compared to binary axis-aligned trees; and (2) extending the empirical risk bound for CART in Klusowski and Tian [1] to the proposed ShapeCART algorithm.
>
> ### Expressiveness Guarantees for SGTs
>
> We formally establish that SGTs are more expressive than binary axis-aligned linear trees by showing that: (1) SGTs are at least as expressive as binary axis-aligned linear trees with a similar number of decision nodes; (2) SGTs can be strictly more expressive than axis-aligned linear trees with a similar number of decision nodes.
>
> **Theorem 1**: Every function that can be represented by a binary axis-aligned linear tree (Def. 1), can be represented by an SGT (Def. 3) with the same number of decision nodes.
>
> _Proof:_ Given a binary axis-aligned linear tree, we construct an SGT with similar structure where the shape function in each decision node takes the form $f_\Theta(\mathbf{w}^\top\mathbf{x}) = \mathbf{w}^\top\mathbf{x}_n -\Theta$ where $\mathbf{w}$ and $\Theta$ match the values in the corresponding decision node in the axis-aligned linear tree.
>
> **Theorem 2**: $\forall B\in\mathbb{N}$, there exists a function for which a binary axis-aligned tree requires at least $B$ additional decision nodes to represent, compared to an SGT.
>
> To prove Theorem 2, we construct a family of two-dimensional labeling functions with periodic and rectangular boundaries:
>
> **Definition**Let $\omega \in \mathbb{N}$ be a frequency parameter. We define the Extended-Plus Labeling function $g:[0,1]^2 \to \{0,1\}$ as follows:
> $$g(x_1, x_2) =
> \begin{cases}
> 1 &\text{if } x_2 \in [0.25, 0.75]\;\;\text{or} \;\; \cos(2\pi\omega x_1)\leq 0, \\ 0 &\text{otherwise}.\end{cases}$$
>
> We show that an SGT can represent any function in this family using a fixed number of nodes, while an axis-aligned linear tree requires a number of nodes that grows linearly with the number of rectangles:
>
> **Lemma 1** The Extended-Plus labelling function can be represented exactly by an SGT with two internal nodes for any value of $\omega$.
>
> **Lemma 2** A binary axis-aligned linear tree requires $4\omega +2$ internal nodes to represent the Extended-Plus labelling function.
>
>
> ### Empirical Risk Bound of ShapeCART
>
> In addition, we extend the empirical risk bounds for CART in [1] to ShapeCART. To do so, we introduce a small modification to our coordinate descent initialization by choosing the better of (1) Weighted K-Means assignments or (2) bin assignments based on the left/right split of the root node in the shape function's inner CART. With this change, we can show that the information gain for a given node in ShapeCART is at least as high as the information gain for CART.
>
> **Lemma 3.**  Let $t$ be an individual node in $T$, let $\mathcal{IG}(t)$ denote the information gain obtained by an axis-aligned (threshold) split on the samples in $t$ from CART, and let $\mathcal{IG}_f(t)$ denote the information gain obtained by a shape function split from ShapeCART on the samples in $t$. We show that $\mathcal{IG}_f(t)\geq\mathcal{IG}(t)$
>
> _Proof Sketch_: To construct the shape function $f_j$ for each feature $j$, we first fit a univariate CART tree that partitions the data along $x_j$. The root node of this tree yields the best axis-aligned split, with information gain $\mathcal{IG}(t, j)$. As we select the initial assignment of bins to branches based on the best between: (1) the right/left assignment based on the root node; and (2) a clustering-based assignment from weighted K-means, we are guaranteed to initialize the coord. decent with an assignment that achieves at least $\mathcal{IG}(t, j)$. Since coord. descent monotonically improves the objective, the final shape function satisfies $\mathcal{IG}_f(f_j, t)\geq\mathcal{IG}(t, j),\forall j$. Given that ShapeCART utilizes the same line search mechanism to select the feature shape function, we can state $\mathcal{IG}_f(t) = \max_{j}\mathcal{IG}_f(f_j, t)\geq\mathcal{IG}(t)=\max_j \mathcal{IG}(t,j)$.
>
> Given Lemma 3, ShapeCART naturally satisfies Lemma 4.1 in [1], and consequently the main result of Theorem 4.2 in [1]. We empirically tested this modification and it has a negligible impact on performance. We will add these results to the revised manuscript.
>
> **Future Work on Theoretical Results:** We agree that our work opens additional interesting directions for future work to further theoretically characterize SGTs and ShapeCART (e.g., developing a GridCART variant for ShapeCART following [9]). Naturally this is beyond the scope of this work, but we will extend our discussion on future work to acknowledge these interesting directions, including the references provided for similar works on CART.
>
> ## Hyperparameters Sensitivity Analysis
>
> Thanks for your suggestion regarding the HP analysis. Following [2] and [3], we normalize the test accuracy within each model and dataset and train a Random Forest regressor to predict test Z-scores from the model hyperparameters and provide the feature importance for our TDIDT approaches in the table below. We will include this analysis in the final manuscript.
>
> | |ShapeCART|ShapeCART$_3$|Shape$^2$CART|Shape$^2$CART$_3$|
> |---|---|---|---|---|
> |max_depth|0.35|0.24|0.148|0.099|
> |criterion |0.038|0.035|0.035 |0.034 |
> |min_samples_split |0.079|0.082 |0.082|0.085 |
> |min_samples_leaf | 0.182 |0.178|0.193|0.191|
> |min_impurity_decrease |0.094|0.08|0.099|0.074|
> |inner_max_leaf_nodes |0.169 |0.192|0.177|0.199|
> |inner_min_samples_leaf|0.088|0.098|0.098|0.083|
> |branching_penalty|---|0.095|---|0.085|
> |H |---|---|0.069|0.062|
> |pairwise_penalty|---|---|0.099|0.088|
>
> We observe that `max_depth, inner_max_leaf_nodes` ($L$), and `min_samples_leaf` are among the top-3 most important HPs for all models evaluated.
>
> ## Visualization
>
> Visualizations of SGT variants can be found the appendix (SGT$_3$: Figure 2, Appendix G; S$^2$GT: Figure 3, Appendix G). In the revised manuscript, we will provide additional examples and visualized S$^2$GT$_3$ models.
>
> ## Contemporary Baselines
>
> Thanks for your suggestion to add contemporary baselines. We have extended our evaluation to include three additional baselines. Two non-greedy SOTA models from the recent year: DPDT (KDD 2025) [2] and SPLIT (ICML 2025) [4], as well as HSTree (ICML 2022) [6]. Sparse bivariate oblique trees, as well as their non-greedy variant, are already included in our experiments (BiCART and BiTAO, respectively). Note that higher-order oblique trees ($M>2$) are not naturally visualizable and are considered less interpretable [5]. We similarly excluded SGTs with higher-order shape functions beyond bivariate for the same reason.
>
> Consistent with prior works on trees, we did not include GAMs in our evaluation as they are not naturally comparable to trees. GAMs learn a shape function for each feature ($C$ shape functions per feature in multi-class datasets). Further, the local explanation size in GAMs is identical to its global explanation size (all shape functions). In contrast, in trees the local explanation size is bound by the tree depth and the global explanation size is bound by the number of nodes in the tree (at most $2^\texttt{depth}$).
>
> Note that based on the other reviews, we have improved the robustness and fairness of our hyperparameter tuning setting. Please see our response to Reviewer Vngp for details on these changes. Additionally, we have added additional datasets at the request of Reviewer 85ma bringing our total dataset count to 26.
>
> Aggregated test accuracy results can be seen in the table below. We observe that ShapeCART$_3$ and ShapeCART consistently outperform the other TDIDT approaches, while ShapeTAO and ShapeTAO$_3$ outperform the non-greedy approaches.
>
> |model|2|3|4|5|6|Best|
> |---|---|---|---|---|---|---|
> |CART|83.9%|81.6%|82.3%|84.0%|85.1%|85.2%|
> |SERDT|83.7%|80.9%|81.7%|83.7%|85.1%|85.1%|
> |HSTree|83.9%|81.6%|82.4%|84.0%|85.1%|85.1%|
> |ShapeCART|_84.5%_|_82.5%_|_83.7%_|_84.9%_|_86.2%_|_86.3%_|
> |ShapeCART_3|**85.7%**|**84.6%**|**85.9%**|**87.3%**|**87.8%**|**87.8%**|
> |AxTAO|83.9%|82.1%|82.8%|84.5%|85.5%|85.4%|
> |DPDT|84.9%|83.4%|83.8%|85.2%|86.2%|86.2%|
> |SPLIT|81.7%|76.6%|77.9%|79.5%|80.2%|80.3%|
> |ShapeTAO|_85.0%_|_83.5%_|_84.6%_|_85.9%_|_86.8%_|_86.8%_|
> |ShapeTAO_3|**86.4%**|**85.1%**|**86.2%**|**87.5%**|**88.0%**|**88.0%**|
>
> \* To accommodate slower non-greedy baselines, we downsample our largest dataset (Higgs) from 11m samples to 1m samples to ensure all experiments are completed in time. The final manuscript will include the results for the full dataset.
>
> ## References
>
> [1] Klusowski, J. M., and Tian, P. M. (2024). Large scale prediction with decision trees. J. Am. Stat. Assoc, 119(545), 525-537.
>
> [2] Kohler, H., et. al. Breiman meets Bellman: Non-Greedy Decision Trees with MDPs.  KDD 2025.
>
> [3] Grinsztajn, Léo, et. al. "Why do tree-based models still outperform deep learning on typical tabular data?." NeurIPS 2022.
>
> [4] Babbar, V., et. al. Near-Optimal Decision Trees in a SPLIT Second. ICML 2025
>
> [5] Kairgeldin, R., and Carreira-Perpiñán, M. Á.. Bivariate decision trees: Smaller, interpretable, more accurate. KDD 2024
>
> [6] Agarwal, A. et. al. Hierarchical Shrinkage: Improving the accuracy and interpretability of tree-based models. ICML 2022
>
> [7] Zhang, X. et. al. Axiomatic interpretability for multiclass additive models. KDD 2019
>
> [8] Murdoch, W. J. et. al. (2019). Definitions, methods, and applications in interpretable machine learning. PNAS, 116(44), 22071-22080.
>
> [9] Zheng, et al. "On the consistency rate of decision tree learning algorithms." Int. Conf. on AI & Stats. PMLR, 2023.

---

> > ### Comment · Reviewer_w6vq · 2025-08-01
> >
> > Thank you for your thorough response! My primary concerns have been addressed, so I am happy to raise my score.

---

> > > ### Author Response · Authors · 2025-08-01
> > >
> > > Thank you for your vote of confidence in our work!

---

### Decision · Program_Chairs · 2025-09-17

**Decision:**

Accept (poster)

**Comment:**

The paper introduces Shape Generalized Trees, which are regression trees that partition the input space along non-linear decision boundaries, and a CART-like algorithm for training such trees.
Although learning optimal non-linear decision rules is quite difficult, the authors propose a clever simplification, first learning a binning function and then learning bin assignments.
Using 12 benchmark datasets, the authors demonstrate that the proposed algorithms outperform methods that estimate axis-aligned tres in terms of out-of-sample accuracy.

Initial feedback from reviewers was very positive, with most reviewers requesting relatively minor clarifications and for the authors to perform additional experiments and include more competing methods.
Of course, a revised version of the manuscript **must** include results from the 26 benchmark datasets and all competing methods.

In response to w6vq's remarks about the lack of theoretical results, the authors sketched how several of Klusowski's results about axis-aligned trees can be extended to the more general setting considered here.
These are promising preliminary results and I encourage the authors to develop them further and revise the manuscript to include the precise statements and full proofs.
In their reply to w6vq, the authors promise to include a short discussion about these promising directions and defer theoretical analysis to future work.
Importantly, the authors should **not** include the unchecked theoretical results in the camera-ready manuscript; I am recommending only that that the paper reporting the methodological development and empirical results be accepted.